# Targeting of the m⁶A eraser ALKBH5 suppresses stemness and chemoresistance of colorectal cancer

Heming Zhou [1], Huarong Chen [1,2], Weixin Liu [1], Cong Liang[3], Shiyan Wang[4], Kai Yuan[1], Alvin Ho Kwan Cheung[5], Yuet Wu[1], Wei Kang [6], Henley Cheung [1], Yanqiang Ding [1], Qinyao Wei[1], Hao Su[1,2], Tianhui Li[1], Weimei Luo[7], Sujun Chen [7], Chi Chun Wong [1] ✉ & Jun Yu [1] ✉

The role of RNA N⁶-methyladenoine (m⁶A) eraser AlkB homologue 5 (ALKBH5) in colorectal cancer (CRC) stem cells (CSCs) is unclear. Here, we find that ALKBH5 expression positively correlates with CSC markers in CRC patients. ALKBH5 induces self-renewal and stemness markers in colorectal CSCs and patient-derived organoids (PDOs). Colon-stem cell specific *Alkbh5* knockin accelerates carcinogen-induced CRC, while tumorigenesis is attenuated in colon-stem cell specific *Alkbh5* knockout mice. Integrated RNA-seq, MeRIP-seq and Ribo-seq reveal FAM84A as an ALKBH5 target. ALKBH5 demethylates m⁶A-modified FAM84A mRNA, causing mRNA decay and reduced expression. Mechanistically, we show that FAM84A represses CSCs by interacting with β-catenin and promoting β-catenin ubiquitination and degradation. By boosting CSCs, ALKBH5 overexpression elicit chemoresistance in CSCs, PDOs and transgenic mice. Targeting of ALKBH5 by knockout or VNPs-siALKBH5 synergizes with chemotherapy to trigger tumor regression in CSCs-/PDOs-derived xenografts and ALKBH5 knockout mice. Together, we reveal that ALKBH5 is essential for colorectal CSCs and is a therapeutic target for overcoming CRC chemoresistance.

Colorectal cancer (CRC) is the third most common cancer and the third leading cause of cancer-related deaths in the United States. Despite advances in early diagnosis and therapy over the past decade, the prognosis of advanced, metastatic CRC remains poor[1,2]. Tumorigenesis in the colon is initiated by a subpopulation of Lgr5⁺ (leucine-rich-repeat-containing G-protein-coupled receptor 5) stem cells, normally restricted to the base of the colonic crypts[3]. Sequential genetic and epigenetic alterations can give rise to cancer stem cells (CSCs) that not only sustain carcinogenesis[4,5], but also a root cause with drug resistance, relapse, and metastasis[6]. Targeting of CSCs is thus a promising strategy to inhibit tumorigenesis and reverse therapy resistance. Nevertheless, progress in selective targeting CSCs remains limited, highlighting an unmet need in the therapeutic management of CRC.

[1]Institute of Digestive Disease and Department of Medicine and Therapeutics, State Key Laboratory of Digestive Disease, Li Ka Shing Institute of Health Sciences, The Chinese University of Hong Kong, Hong Kong SAR, China. [2]Department of Anaesthesia and Intensive Care, Faculty of Medicine, The Chinese University of Hong Kong, Hong Kong SAR, China. [3]Departement of Oncology, the First Affiliated Hospital, Sun Yat-sen University, Guangzhou, China. [4]Institute of Precision Medicine, The First Affiliated Hospital, Sun Yat-sen University, Guangzhou, China. [5]Department of Anatomical and Cellular Pathology, The Chinese University of Hong Kong, Hong Kong SAR, China. [6]Department of Anatomical and Cellular Pathology, State Key Laboratory of Oncology in South China, Prince of Wales Hospital, The Chinese University of Hong Kong, Hong Kong SAR, China. [7]West China School of Public Health and West China Fourth Hospital, and State Key Laboratory of Biotherapy, Sichuan University, Chengdu, Sichuan, China. ✉e-mail: chichun.wong@cuhk.edu.hk; junyu@cuhk.edu.hk

Emerging studies suggest the potential roles of m⁶A regulators in modulating stemness and tumorigenesis in colon. M⁶A writers METTL3 and METTL14 have been reported to sustain self-renewal capacity in the normal colon[7,8]. METTL3 and m⁶A readers including YTHDF1/2 and IGF2BP1/2 have been implicated in colorectal tumorigenesis, chemoresistance[9–14], and metastasis[14–17] through modulating the translation of m⁶A downstream targets.

ALKBH5 has been reported to exert oncogenic[18–21] or tumor suppressive function[22–26] in a context-dependent manner. On the other hand, several studies have suggested that ALKBH5 selectively enhances self-renewal capacity of cancer stem cells (CSCs) in solid tumors and leukemia[27–30] and also boosts stemness properties in several tumor types[20,31–34] through activation of WNT/β-catenin signaling. Nevertheless, the role of ALKBH5 in CRC stem cells remains incompletely understood.

Here, we show that ALKBH5 is positively correlated with colorectal CSCs markers in CRC patients. In this study, we comprehensively demonstrate that ALKBH5 is essential for maintaining stemness in colorectal CSCs, patient-derived organoids (PDOs) and colon stem cell-specific (Lgr5⁺) Alkbh5 knockin (Rosa26^{lsl-Alkbh5}LGR5-Cre^{ERT2}, Alkbh5-cKI) and knockout (Alkbh5^{flox/flox} LGR5-Cre^{ERT2}, Alkbh5-cKO) mice. Integrated MeRIP-seq, RNA-seq and Ribo-seq unveil a FAM84A-β-catenin axis driving CRC stemness downstream of ALKBH5. Furthermore, ALKBH5 promotes chemoresistance in CSCs and PDOs. Finally, the targeting of ALKBH5 using knockout or vesicle-like nanoparticles (VNP)-encapsulated siRNA together with chemotherapy synergistically inhibits the growth of colorectal CSCs and PDOs and Alkbh5 knockout mice, suggesting that ALKBH5 is a potential therapeutic target in CRC.

## Results

### ALKBH5 expression correlates with LGR5 and other stemness markers in human CRC

To ask if m⁶A modification plays a role in CRC stemness, we examined the correlation between m⁶A regulators with stemness markers in CPTAC cohort (protein) and TCGA-CRC dataset (mRNA). At protein level, ALKBH5 shows highest positive correlation with LGR5 Index (Fig. S1A)[35,36], whereas at mRNA level m⁶A regulators showed either weak or no correlation with LGR5 and CD133 (Fig. S1B). In TCGA cohort, ALBKH5 mRNA negatively correlates with m⁶A readers, but had no correlation with m⁶A writers and erasers (Fig. S1C).

To ask if ALKBH5 correlates with stemness markers among different cell types in CRC tumor microenvironment (TME), we analyzed a single-cell RNA sequencing (scRNA-seq) dataset (GSE132465)[37]. A higher proportion of epithelial/tumor cells expressed ALKBH5 (>30%) compared to stromal cells, myeloid cells, T cells, and B cells (Fig. S1D). Furthermore, within tumor cell subsets, co-expression of ALKBH5 with LGR5 ($R = 0.39$, $P < 0.0001$; $n = 1295$ cells) and CD133 ($R = 0.25$, $P < 0.0001$; $n = 2867$ cells) were observed (Fig. S1E). In contrast, few cells showed co-expression of ALKBH5 with stemness markers in myeloid, stromal, T, or B cells (Fig. S1E). This suggests that tumor cells are the predominant source of ALKBH5 in the CRC TME that corresponds to stemness properties.

Then we separated tumor cells into LGR5⁺ and LGR5⁻ clusters (Fig. S1F), revealing significantly higher ALKBH5 expression in LGR5⁺ cells compared to LGR5⁻ cells (Fig. S1F). To confirm that ALKBH5 is associated with stemness traits in human CRC, we determined mRNA expression of ALKBH5 and CSC markers ($N = 151$). ALKBH5 mRNA is positively correlated to LGR5 ($P < 0.0001$, $R = 0.429$), CD133 ($P < 0.0001$, $R = 0.413$) and CD44 ($P = 0.0065$, $R = 0.224$) (Fig. S1G). In a second cohort ($N = 12$), ALKBH5 protein was up-regulated in CRC (Fig. S1H and S1I), and positively correlated with CD133 ($P = 0.0002$, $R = 0.721$) and LGR5 ($P < 0.0001$, $R = 0.797$) (Fig. S1J). In addition, we determined ALKBH5 protein levels by immunohistochemistry on tissue microarrays (TMA) in an independent CRC cohort. Consistently, ALKBH5 protein expression also showed positive correlation with

LGR5 ($N = 163$, $P = 0.00037$, $X^2 = 17.0$) and CD133 ($N = 188$, $P = 0.0003$, $X^2 = 12.9$) (Fig. S1K). Moreover, co-staining of ALKBH5 and LGR5 (Fig. S1L) and ALKBH5 and CD133 (Fig. S1M) demonstrated their co-localized expression in murine CRC tumors. Our multicohort analysis thus demonstrated that ALKBH5 expression is correlated with stemness markers in CRC.

### Colon stem cell-specific ALKBH5 knockin promotes intestinal stem cell (ISC) proliferation and restrains differentiation in mice

To determine the impact of colon stem cell-specific ALKBH5 expression on intestinal homeostasis, we generated conditional colon stem cell-specific Alkbh5 knockin mice (Rosa26^{lsl-Alkbh5}LGR5-Cre^{ERT2}, Alkbh5-cKI) (Fig. 1A). To test the role of ALKBH5 (short-term: 2 weeks after Tamoxifen), ALKBH5 knockin was validated in colon crypts (Fig. 1B). Stem-like proliferating cells in colon crypts in ALKBH5 cKI mice was significantly elevated, as indicated by Ki67 staining (Fig. 1C). On the contrary, IF staining of differentiated enterocyte markers, such as MUC2 for goblet cells[38] (Fig. 1D) and Chromogranin A for endocrine cells (Fig. 1E) were down-regulated in ALKBH5 cKI mouse colon, confirming suppressed cell differentiation. TUNEL staining showed reduced apoptosis in ALKBH5 cKI mice upper colon layer (Fig. 1F). Nevertheless, long-term (4 months) ALKBH5 knockin (Fig. S2A, B) had no effect on intestinal homeostasis, as evidenced by histological analysis of small intestine and colon (Fig. S2C), and measurement of colon length and body weight (Fig. S2D). Together, short-term LGR5⁺ cell-specific ALKBH5 knockin promotes ISC proliferation and restrains differentiation and apoptosis in mice.

### Colon stem cell-specific ALKBH5 knockin accelerates CRC initiation in mice

To determine whether high ALKBH5 predisposes colorectal tumorigenesis by inducing CSCs, ALKBH5 cKI mice and wildtype (WT) littermates were administrated AOM-DSS treatment to induce CRC (Fig. 1G). At sacrifice, tumor number ($P = 0.02$) and burden ($P = 0.049$) were both significantly higher in ALKBH5 cKI mice compared to WT mice (Figs. 1H, I). ALKBH5 knockin was confirmed in tumor tissues (Fig. 1J). Up-regulation of ALKBH5 and stemness markers in ALKBH5 cKI CRC tissues was further confirmed (Fig. 1K). Histological evaluation demonstrated that ALKBH5 cKI mice had a higher proportion of high-grade dysplasia (HGD) ($P < 0.0001$) (Fig. 1L), increased cell proliferation (Ki67 staining, $P = 0.02$) (Fig. 1M) and reduced apoptosis (TUNEL staining, $P = 0.0005$) (Fig. 1N) compared to WT mice. Increased LGR5⁺ (Fig. 1O) and CD133⁺ (Fig. 1P) cells in the tumors of ALKBH5 cKI mice were demonstrated by IF. To validate that ALKBH5 cKI induced stemness phenotypes, we isolated primary CRC organoids from ALKBH5 cKI mice and WT mice. Primary CRC organoids from cKI mice had higher ALKBH5 expression (Fig. S2E). As shown in Fig. 1Q, tumor organoids from ALKBH5 cKI mice showed superior self-renewal capacity. Primary CRC organoids (passage 2) from ALKBH5-cKI mice demonstrated increased proliferation (Fig. 1R), suppressed apoptosis (Fig. 1S) and elevated proportion of CD133⁺ cell population (Fig. 1T) as compared to WT mice. Together, LGR5⁺ cell-specific ALKBH5 knockin in mice accelerates CRC initiation.

### Colon stem cell-specific knockout of ALKBH5 impairs colorectal tumorigenesis in mice

We next established conditional and colon stem cell-specific ALKBH5 knockout mice (Alkbh5^{flox/flox}LGR5-Cre^{ERT2}, Alkbh5-cKO) (Fig. 2A). In the absence of AOM/DSS, long-term (4 months) ALKBH5 knockout (Fig. S3A, B) had no effect on intestinal homeostasis, as evidenced by histological analysis of small intestine and colon (Fig. S3C), and measurement of colon length and body weight (Fig. S3D). Meanwhile, ALKBH5 cKO mice demonstrated attenuated AOM/DSS-induced CRC formation compared to WT mice (Fig. 2B), as evidenced by significant reductions in tumor multiplicity ($P = 0.03$) and load ($P = 0.047$)

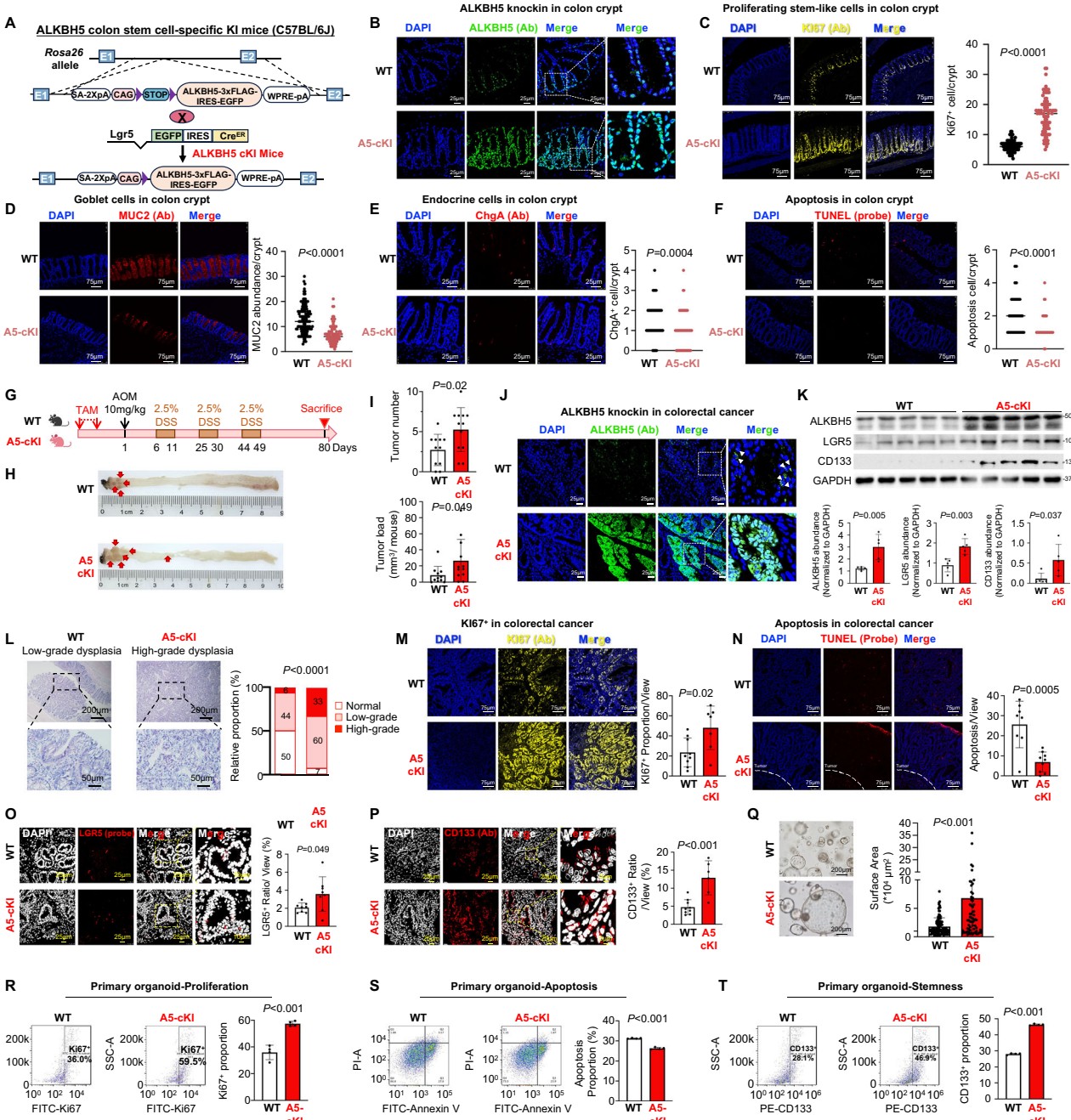

(Fig. 2C, D). ALKBH5 knockout was confirmed (Fig. S3E and Fig. 2E) along with reduced protein expression of LGR5, CD133, CD44 and EpCAM (Fig. 2E). Moreover, ALKBH5 cKO mice had reduced the incidence of adenocarcinoma and HGD ($P < 0.0001$) (Fig. 2F) and suppressed cell proliferation (Fig. 2G). Down-regulation of LGR5[+] (Fig. 2H) and CD133[+] (Fig. 2I) cells in tumors of ALKBH5 cKO mice was confirmed by IF. CRC organoids from cKO mice (Fig. S3F) showed impaired self-renewal (Fig. 2J). Flow cytometry showed that CD133 (Fig. 2K) and LGR5[EGFP] (Fig. 2L) were downregulated in ALKBH5-cKO organoids compared to WT organoids. Hence, LGR5[+] cell-specific ALKBH5 knockout in mice depleted CSCs, thus impairing tumorigenesis.

## ALKBH5 promotes tumorigenicity of patient-derived CSCs and organoids

We next assessed the function of ALKBH5 in colorectal CSCs. Using two patient-derived colorectal CSCs-enriched tumorspheres, POP66 and CSC28[39], we performed loss-of-function and gain-of-function assays. ALKBH5 knockdown in CSCs reduced proliferation (Fig. S4A and S4C) as well as self-renewal ability as determined by in vitro limiting dilution assay (LDA) (Fig. 3A, Fig. S4E and Fig. S4G). In contrast, ALKBH5 overexpression promoted CSC proliferation (Fig. S4B, D) and self-renewal (Fig. 3B, Fig. S4F and Fig. S4H). Concordantly, ALKBH5 silencing down-regulated CSC markers, whereas ALKBH5 overexpression induced their expression (Fig. 3C). To investigate CSC-specific function of ALKBH5 in vivo, we conducted in vivo LDA by serially implanting POP66 or CSC28 cells, with or without ALKBH5, into NOD-SCID mice ($n = 24$). ALKBH5 knockdown markedly reduced the tumorigenicity of POP66 ($P < 0.001$) (Fig. 3D) and CSC28 ($P < 0.001$) (Fig. 3E), with no tumors <200 cells. In contrast, shControl cells were able to initiate tumors with as few as 40 cells. These results collectively indicated that ALKBH5 drives CSC stemness in vitro and in vivo.

Moreover, in PDO 816 and 828, which harbored APC and activated WNT/β-catenin pathway (Supplementary Data 1 and Supplementary

**Fig. 1 | Colon stem cell-specific ALKBH5 knockin accelerates CRC initiation and promotes stemness features. A** Schematic diagram for the construction of colon stem cell-specific *Alkbh5* knockin mice (Rosa26[lsl-Alkbh5]LGR5-Cre[ERT2]). **B** ALKBH5 knockin efficacy in colon crypts by anti-ALKBH5 (anti-ALKBH5 antibody). **C** Ki67 abundance as determined by IF staining (anti-Ki67 antibody, $n = 89$ crypts for WT mice, $n = 89$ crypts for A5-cKI mice. Student's *t*-test, two-sided.) **D** MUC2 abundance as determined by IF staining (anti-MUC2 antibody, $n = 127$ crypts for WT mice, $n = 89$ crypts for A5-cKI mice. Student's *t*-test, two-sided). **E** ChgA abundance as determined by IF staining (anti-ChgA antibody, $n = 100$ crypts for WT mice, $n = 65$ crypts for A5-cKI mice. Student's *t*-test, two-sided). **F** Apoptosis as determined by TUNEL staining (anti-TUNEL probe, $n = 58$ crypts for WT mice, $n = 58$ crypts for A5-cKI mice. Student's *t*-test, two-sided). **G** AOM-DSS-induced CRC study design. The diagram is created in BioRender. Chou, H. (2025) https://BioRender.com/uax7xbx. **H** Representative images of colon tumors after harvesting. **I** Colon tumor number (Student's *t*-test, two-sided) and load (Student's *t*-test, two-sided), ($n = 11$ for WT mice and $n = 12$ for A5-cKI mice). **J** ALKBH5 knock in efficacy in tumor tissues was validated by IF staining (anti-ALKBH5 antibody). **K** Western blot of Alkbh5 and LGR5 and CD133 expression in tumor tissues (upper panel). Quantitative analysis of ALKBH5, LGR5 and CD133 ($n = 5$, each dot represents an independent mouse. Student's *t*-test, two-sided) protein expression, normalized to GAPDH (lower panel). **L** Representative images of H&E staining and pathological grading into LGD, HGD and adenocarcinoma (Chi-square, two-sided). Presented data is representative of 2 independent batches of transgenic mice. **M** Ki67 expression as determined by IF staining (anti-Ki67 antibody, $n = 9$ for WT mice, $n = 7$ for A5-cKI mice. Student's *t*-test, two-sided). **N** Apoptosis as determined by TUNEL staining (anti-TUNEL probe, $n = 8$ for WT mice, $n = 9$ for A5-cKI mice. Student's *t*-test, two-sided). **O** LGR5 (anti-LGR5 mRNA probe) expression as determined by FISH staining ($n = 8$ for WT mice, $n = 8$ for A5-cKI mice. Student's *t*-test, two-sided). **P** CD133 (anti-CD133 antibody) expression as determined by IF staining ($n = 9$ for WT mice, $n = 6$ for A5-cKI mice. Student's *t*-test, two-sided). **Q** Self-renewal capacity of primary CRC organoid (WENR medium) from WT mice and A5-cKI mice under a light microscope (left panel). Surface area of CRC organoids from WT mice and A5-cKI mice (right panel, $n = 107$ organoids for WT mice, $n = 79$ organoids for A5-cKI mice. Student's *t*-test, two-sided). **R** KI67+ proportion of primary CRC organoid from WT mice and A5-cKI mice as determined by flow cytometry ($n = 4$, each dot represents an independent sample. Student's *t*-test, two-sided). **S** Apoptosis cells of primary CRC organoid from WT mice and A5-cKI mice as determined by flow cytometry ($n = 4$, each dot represents an independent sample. Student's *t*-test, two-sided). **T** CD133+ proportion of primary CRC organoid from WT mice and A5-cKI mice as determined by flow cytometry ($n = 4$, each dot represents an independent sample. Student's *t*-test, two-sided). Centers and error bars represent mean and Standard deviation, respectively.

---

Table 5), ALKBH5 knockdown significantly impaired the growth, self-renewal and expression of LGR5, CD133 and CD44 (Fig. 3F). Whereas ectopic expression of ALKBH5 showed opposite phenotypes (Fig. 3G), confirming that ALKBH5 also boosts stemness traits in CRC CSCs and PDOs.

### Integrated MeRIP-seq, bulk RNA-seq, and Ribo-seq identified FAM84A as a downstream target of ALKBH5 in colorectal CSC
ALKHB5 functions as an m6A demethylase. As a demethylase, ALKBH5 overexpression suppressed global m6A levels, while its knockdown increased m6A levels (Fig. 4A, Fig. S5A, B). To dissect ALKBH5-modulated m6A epitranscriptome in colorectal CSCs, integrative MeRIP-seq, bulk RNA-seq and Ribo-seq analyses were performed (Fig. 4B). GSEA of differentially expressed genes (DEGs) in ALKBH5-knockdown CSC28 revealed Wnt signaling pathway among the top depleted KEGG pathways (Fig. 4C). We mapped the m6A methylomes of CSC28 cells by m6A-sequencing. The GGAC consensus motif is highly enriched within m6A sites in shNC, shALKBH5, (Fig. 4D), EV, and ALKBH5-overexpressing CSC28 (Fig. S5C). Other m6A motifs are listed in Supplementary Data 2. MeRIP-seq analysis revealed that in CSC28-shALKBH5 vs. shNC (total 58784 peaks), 12825 (21.82%) peaks were up-regulated whereas 9887 (16.82%) peaks were reduced in shALKBH5 group (Fig. 4D). For ALKBH5-OE vs. EV (total 55322 peaks), 9652 (17.45%) peaks were up-regulated, together with 9855 (17.81%) down-regulated peaks (Fig. 4D). We next performed Ribo-seq to assess translation. Analysis of CDS distribution (Fig. S5D, E), enrichment of transcription initiation sites and tri-nucleotide periodicity (Fig. S5F) confirmed the validity of our Ribo-seq dataset. Pathway analysis based on Ribo-seq dataset also revealed depletion of Wnt signaling pathway among actively translating transcripts in ALKBH5 knockdown CSC28 spheres (Fig. 4E).

Upon ALKBH5 knockdown, m6A-gained genes ($P < 0.05$, |FC| > 2) (1275) demonstrated decreased mRNA expression compared to unaltered genes (9577), without changing the translation efficiency (Fig. S5G). Whereas m6A-loss genes ($P < 0.05$, |FC| > 2) (1681) induced by ALKBH5 overexpression showed up-regulated mRNA levels compared to unaltered genes (9250), but no change in translation efficiency (Fig. S5H). This implies that ALKBH5-mediated m6A demethylation predominantly modulates mRNA expression.

We integrated ALKBH5-m6A targets with genes that were differentially regulated in RNA-seq and Ribo-seq analyses. A total of 2945 m6A-modified genes were identified by ALKBH5 in m6A-seq (log2|FC| = 1, $P < 0.01$). In the RNA-seq data, 319 genes were found to be differentially expressed with a log2|FC| of at least 1.5 and $P < 0.001$ in CSC-shALKBH5 compared to shNC. The ribo-seq analysis revealed 2462 DEGs under the same thresholds in the same comparison. Among the identified genes, six candidates were deregulated by ALKBH5 at m6A, transcriptional, and translational levels (Fig. 4F). Notably, only FAM84A showed a negative correlation with ALKBH5 in both TCGA and GEPIA databases (Fig. 4F), with the observed changes in transcriptional and translational levels are consistent with the correlation coefficient, suggesting that FAM84A may function downstream of ALKBH5 in humans. In two independent cohorts (GSE39582 and GSE17536), we validated the negative correlation between ALKBH5 and FAM84A mRNA (GSE39582: $R = -0.3525$, $P < 0.0001$; GSE17536: $R = -0.3655$, $P < 0.0001$) (Fig. 4F). UCSC Genome Browser snapshots of m6A pulldown versus input revealed that FAM84A m6A was induced after ALKBH5 knockdown (Fig. 4G), whilst being down-regulated by ALKBH5 overexpression (Fig. S5I), implying that ALKBH5 might demethylate FAM84A.

In agreement with our integrative analyses, m6A-immunoprecipitation-qPCR (MeRIP-qPCR) showed that ALKBH5 knockdown increased m6A enrichment in FAM84A mRNA (Fig. 4H). To ask if its demethylase activity is required for stemness-promoting function, we overexpressed wildtype ALKBH5 (ALKBH5[WT]) or catalytic-dead ALKBH5 (ALKBH5[H204A]) in CSCs depleted of endogenous ALKBH5. Overexpression of ALKBH5[WT], but not ALKBH5[H204A] restored self-renewal capacity caused by ALKBH5-knockdown (Fig. S5J), implying that ALKBH5-induced CSC properties in an m6A-dependent manner. Overexpression of ALKBH5[WT], but not ALKBH5[H204A], down-regulated FAM84A m6A modification (Fig. 4H). The direct interaction of ALKBH5 with FAM84A mRNA was confirmed by RNA immunoprecipitation-qPCR (RIP-qPCR) (Fig. 4I), suggesting that ALKBH5 binds and demethylates m6A-modified FAM84A mRNA. Consequently, ALKBH5 knockdown increased FAM84A mRNA expression (Fig. 4J), while ALKBH5[WT] (but not mutant ALKBH5[H204A]) overexpression exerted opposite effects (Fig. 4J). Moreover, using tumor tissues from transgenic mice, we validated binding of ALKBH5 protein to FAM84A mRNA and its consequent effect on FAM84A expression (Fig. 4K). To validate direct function of ALKBH5 on FAM84A m6A modification and mRNA expression, we employed targeted dCas13b-ALKBH5 system[40] (Fig. 4L and Fig. S5K). DCas13b-A5 did not change FAM84A mRNA abundance (Fig. S5L). Homing of dCas13b-ALKBH5 to FAM84A mRNA was induced by co-transfection with three independent gRNAs (Supplementary Table 4). Results showed that co-transfection of dCas13b-ALKBH5 with

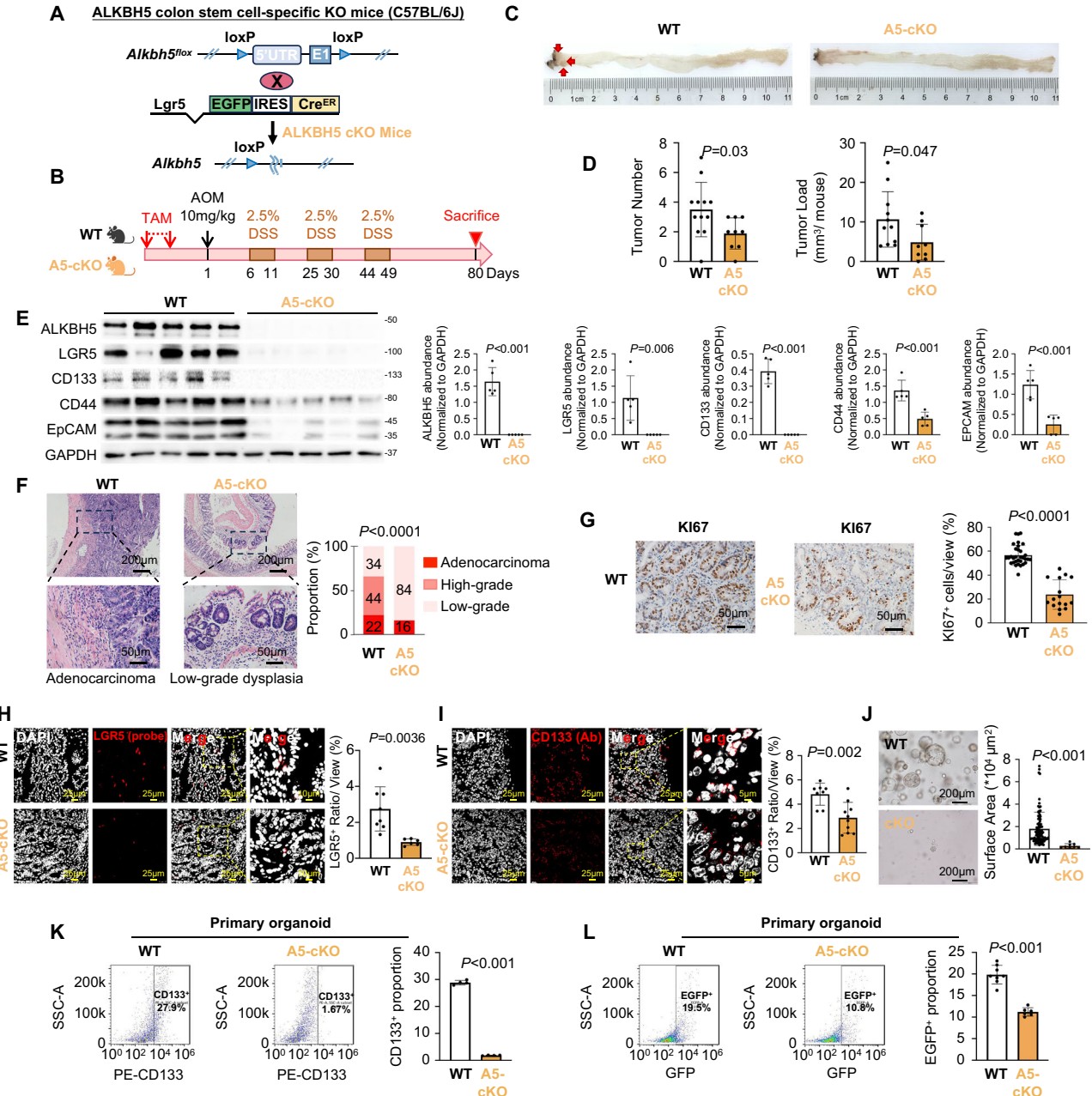

**Fig. 2 | Colon stem cell-specific ALKBH5 knockout inhibits CRC initiation and stemness features. A** Schematic diagram for the construction of colon stem cell-specific *Alkbh5* knockout mice (Alkbh5^flox/flox LGR5-Cre^ERT2). **B** AOM-DSS-induced CRC workflow. The diagram is created in BioRender. Chou, H. (2025) https://BioRender.com/uax7xbx. **C** Representative images of colon tumors after harvesting. **D** Tumor number (Student's *t*-test, two-sided) and load (Student's *t*-test, two-sided). (*n* = 11 for WT mice and *n* = 9 for A5-cKO mice). **E** Western blot validation of Alkbh5 knockout and LGR5, CD133, CD44 and EpCAM in tumor tissues (left panel). Quantitative analysis of ALKBH5, LGR5, CD133, CD44 and EpCAM (*n* = 5, each dot represents an independent mouse. Student's *t*-test, two-sided) protein expression, normalized to GAPDH (right panel). **F** Representative images of H&E staining and pathological analysis of LGD, HGD and adenocarcinoma (Chi-square, two-sided). Presented data is representative of 2 independent batches of transgenic mice. **G** Ki67 abundance as determined by IHC staining (anti-KI67 antibody, *n* = 32 for WT

mice, *n* = 16 for A5-cKO mice. Student's *t*-test, two-sided). **H** LGR5 (anti-LGR5 mRNA probe) expression as assessed by FISH staining (*n* = 8 for WT mice, *n* = 6 for A5-cKO mice. Student's *t*-test, two-sided). **I** CD133 (anti-CD133 antibody) expression as assessed by IF staining (*n* = 8 for WT mice, *n* = 10 for A5-cKO mice. Student's *t*-test, two-sided). **J** Growth of primary organoids (WENR medium) from Alkbh5 knockout mice observed under microscope (left panel). Surface area of CRC organoids from WT mice and A5-cKO mice (right panel, *n* = 92 organoids for WT mice, *n* = 8 organoids for A5-cKO mice. Student's *t*-test, two-sided). **K** CD133^+ proportion of primary CRC organoid from WT mice and A5-cKO mice as determined by flow cytometry (*n* = 4, each dot represents an independent sample. Student's *t*-test, two-sided). **L** LGR5^EGFP+ proportion of primary CRC organoid from WT mice and A5-cKO mice as determined by flow cytometry (*n* = 8 for WT and *n* = 6 for A5-cKO, each dot represents an independent sample. Student's *t*-test, two-sided). Centers and error bars represent mean and Standard deviation, respectively.

gRNAs reduced FAM84A m^6A modification (Fig. 4M) and mRNA expression (Fig. S5M). Collectively, FAM84A is a target of ALKBH5 in CSCs by promoting FAM84A m^6A demethylation and down-regulation of FAM84A expression.

**ALKBH5 decreased the stability of FAM84A mRNA by promoting its dissociation from IGF2PB1**

We investigated how ALKBH5 regulated FAM84A in CRC. In both ALKBH5-deficient CSCs, FAM84A protein was upregulated, while

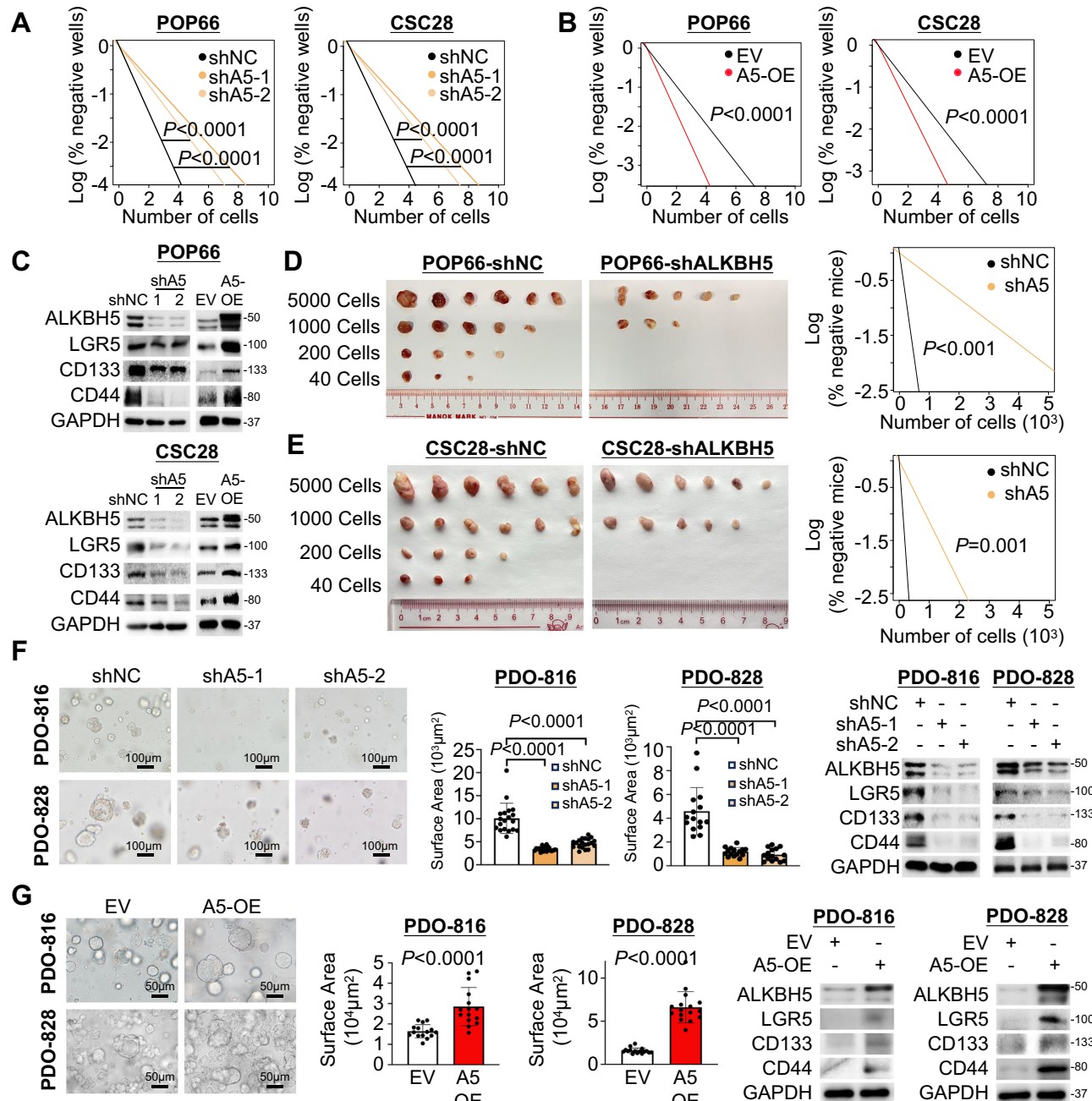

**Fig. 3 | ALKBH5 is essential for the self-renewal of colorectal cancer stem cells (CSCs). A** In vitro LDA on POP66 (Chi-Square, two-sided) and CSC28 (Chi-Square, two-sided) with ALKBH5 knockdown. **B** In vitro LDA on POP66 (Chi-Square, two-sided) and CSC28 (Chi-Square, two-sided) with ALKBH5 overexpression. **C** LGR5, CD133 and CD44 protein expression in POP66 and CSC28 with ALKBH5 knockdown or overexpression. **D** In vivo LDA on POP66 with ALKBH5 knockdown (Chi-Square, two-sided). **E** In vivo LDA on CSC28 after ALKBH5 knockdown (Chi-Square, two-sided). **F** Effect of ALKBH5 knockdown on the growth of CRC organoids PDO-816 (left panel) ($n = 19$ views for shNC organoids, $n = 20$ views for shA5-1 organoids, $n = 20$ views for shA5-2 organoids. one-way ANOVA) and PDO-828 ($n = 15$ views for shNC organoids, $n = 15$ views for shA5-1 organoids, $n = 15$ views for shA5-2 organoids. one-way ANOVA) (EN medium). LGR5, CD133 and CD44 protein expression in PDO816 and PDO828 with ALKBH5 knockdown (right panel). **G** Effect of ALKBH5 overexpression on the growth of CRC organoids PDO-816 (left panel) ($n = 15$ views for EV organoids, $n = 15$ views for A5-OE organoids. Student's $t$-test, two-sided) and PDO-828 ($n = 16$ views for EV organoids, $n = 16$ views for A5-OE organoids. Student's $t$-test, two-sided) (EN medium). LGR5, CD133 and CD44 protein expression in PDO816 and PDO828 with ALKBH5 overexpression (right panel). Centers and error bars represent mean and Standard deviation, respectively.

ALKBH5 overexpression reduced FAM84A protein abundance (Fig. 5A and Fig. S6A, B). Additionally, tumors from ALKBH5 cKO mice showed increased FAM84A expression compared to WT littermates (Fig. 5B and S6C). In human CRC patients, we observed a negative correlation between ALKBH5 and FAM84A ($N = 164$, $P < 0.01$, $X^2 = 6.82$) (Fig. 5C). METTL3 knockdown in CSCs had on significant effect on FAM84A m6A modification (Fig. S6D), mRNA expression (Fig. S6E), and protein

expression (Fig. S6F), inferring that FAM84A m6A levels are primarily regulated by ALKBH5 in CSCs. ALKBH5 knockdown in CSCs did not induce compensatory alterations in METTL3 or FTO (Fig. S6G).

Since ALKBH5 up-regulates FAM84A simultaneously at mRNA and protein levels, there is no significant impact on its translation efficiency (Fig. S6H). We hypothesize that ALKBH5-mediated m6A demethylation regulates FAM84A at mRNA level. Given that m6A

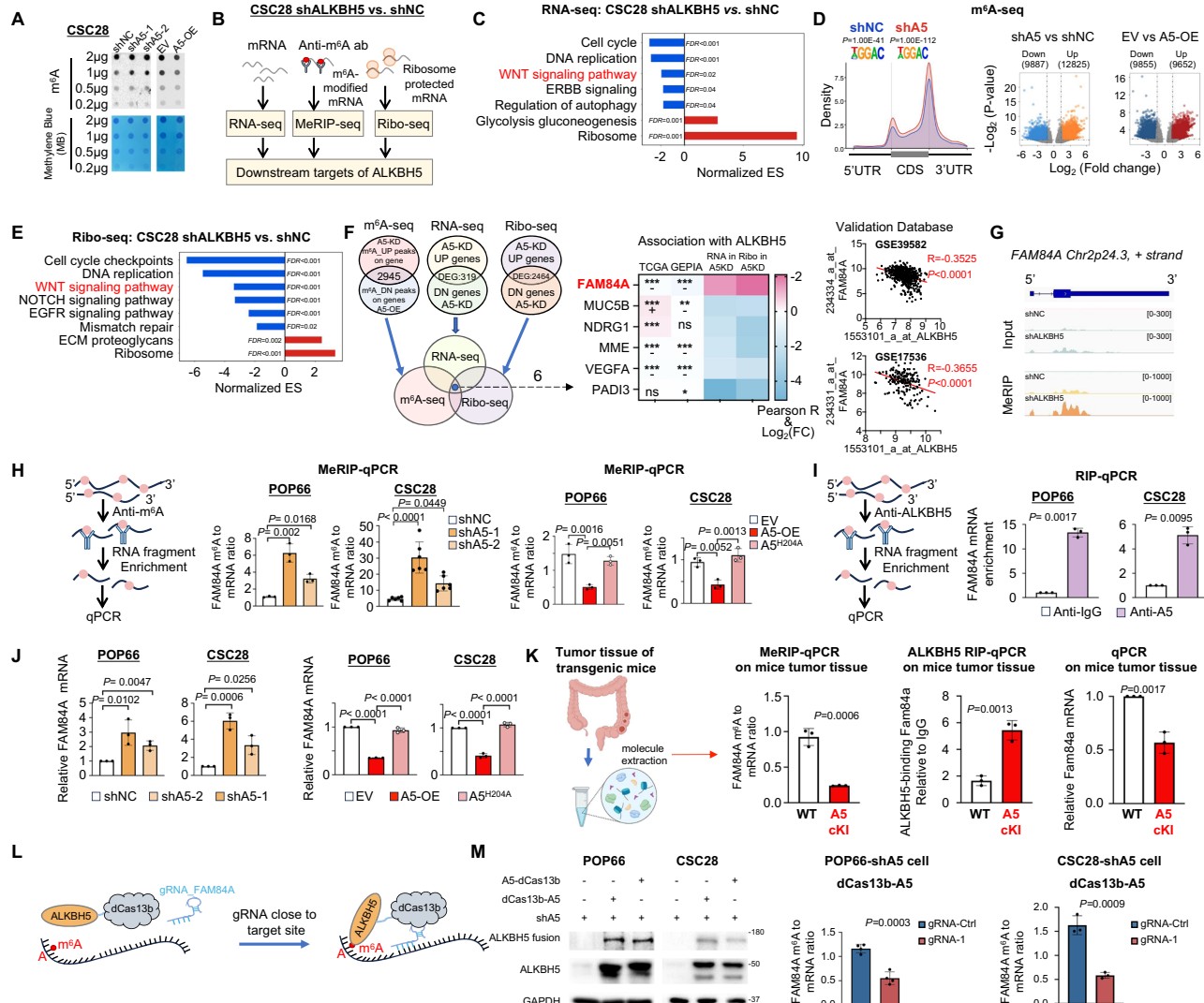

**Fig. 4 | FAM84A is a downstream target of ALKBH5 in colorectal CSCs. A** Dot blot assay of CSC28 with ALKBH5 knockdown and overexpression. **B** Work-flow for systematic identification of ALKBH5 downstream targets. **C** Top enriched differential pathways in ALKBH5-knockdown versus control CSC28 cells by RNA-seq. **D** The top enriched motifs based on m⁶A peaks in ALKBH5-knockdown CSC28 cells and control cells. *P* values for m6A motif were derived using the Fisher's exact test algorithm in HOMER software (upper panel). Differential m⁶A methylated peaks after ALKBH5 knockdown or overexpression, as determined by MeRIP-seq (lower panel). **E** Top differential enriched pathways in ALKBH5-knockdown versus control CSC28 cells by Ribo-seq. **F** Integrated analysis of RNA-seq, MeRIP-seq and Ribo-seq datasets (left panel). The correlation between ALKBH5 and FAM84A mRNA in TCGA (Pearson, two-sided) GEPIA (Pearson, two-sided) datasets and foldchange in RNA-seq (Student's *t*-test, two-sided) and Ribo-seq (Student's *t*-test, two-sided) (middle panel). The correlation between ALKBH5 and FAM84A mRNA in GSE39582 (Pearson, two-sided) and GSE17536 (Pearson, two-sided) (right panel). **G** UCSC snapshots of MeRIP-seq reads of FAM84A. Normalized to RNA input. **H** m⁶A abundance on FAM84A as revealed by MeRIP-qPCR. Schematic design of MeRIP-qPCR assay (left panel). m⁶A modification on FAM84A as determined by MeRIP-qPCR (For POP66, *n* = 3; For CSC28, *n* = 6, each dot represents an independent sample. One-way ANOVA) (middle panel). m⁶A modification on FAM84A after overexpression of WT

ALKBH5 (A5-OE) or mutant ALKBH5 (A5^H204A) in CSC (*n* = 3, each dot represents an independent sample. One-way ANOVA) (right panel). **I** Direct Binding between ALKBH5 and FAM84A mRNA, as determined by RIP-qPCR (*n* = 3, each dot represents an independent sample. Student's *t*-test, two-sided). **J** FAM84A expression as determined by qPCR (*n* = 3, each dot represents an independent sample. One-way ANOVA). **K** m⁶A levels of FAM84A mRNA (left panel), the binding of FAM84A mRNA to ALKBH5 (middle panel) and FAM84A mRNA expression (right panel) in WT and ALKBH5-cKI mice tumor as revealed by MeRIP-qPCR (*n* = 3, each dot represents an independent mouse. Student's *t*-test, two-sided), ALKBH5-RIP-qPCR (Presented data is representative of 3 independent biological replicates. Student's *t*-test, two-sided) and qPCR (*n* = 3, each dot represents an independent mouse. Student's *t*-test, two-sided), respectively. The diagram is created in BioRender. Chou, H. (2025) https://BioRender.com/csvnp3n. **L** Schematic diagram of site-specific RNA targeting using dCas13b-ALKBH5 fusion proteins with gRNA close to the target site. **M** Establishment of dCas13b-ALKBH5 system in colorectal CSCs as determined by western blot (left panel). FAM84Am⁶A modification levels (right panel) in colorectal CSCs co-transfected with dCas13b-ALKBH5 and the gRNA (For POP66, *n* = 4; for CSC28, *n* = 3, each dot represents an independent sample. Student's *t*-test, two-sided). Centers and error bars represent mean and Standard deviation, respectively.

modulation affects mRNA stability, we asked if ALKBH5 regulates FAM84A mRNA decay. ALKBH5 depletion in CSCs stabilized FAM84A mRNA (Fig. 5D and S6I). Reciprocally, ALKBH5 overexpression accelerates FAM84A mRNA decay (Fig. 5E and Fig. S6I). To validate this, we generated luciferase reporters by inserting FAM84A 5'UTR motif (Fig. 5F) or mutated 5'UTR motif (RRACH to TTTCT) (Fig. 5G)

upstream of firefly luciferase. ALKBH5 knockdown in CSCs significantly increased FAM84A luciferase activity (Fig. 5F and Fig. S6J). Mutation of m⁶A sites abrogated effect of ALKBH5 on luciferase activity (Fig. 5G and Fig. S6K). In contrast, ALKBH5^WT (but not ALKBH5^H204A) overexpression reduced FAM84A luciferase activity (Fig. 5F and Fig. S6J), an effect abolished in mutated 5'-UTR motif

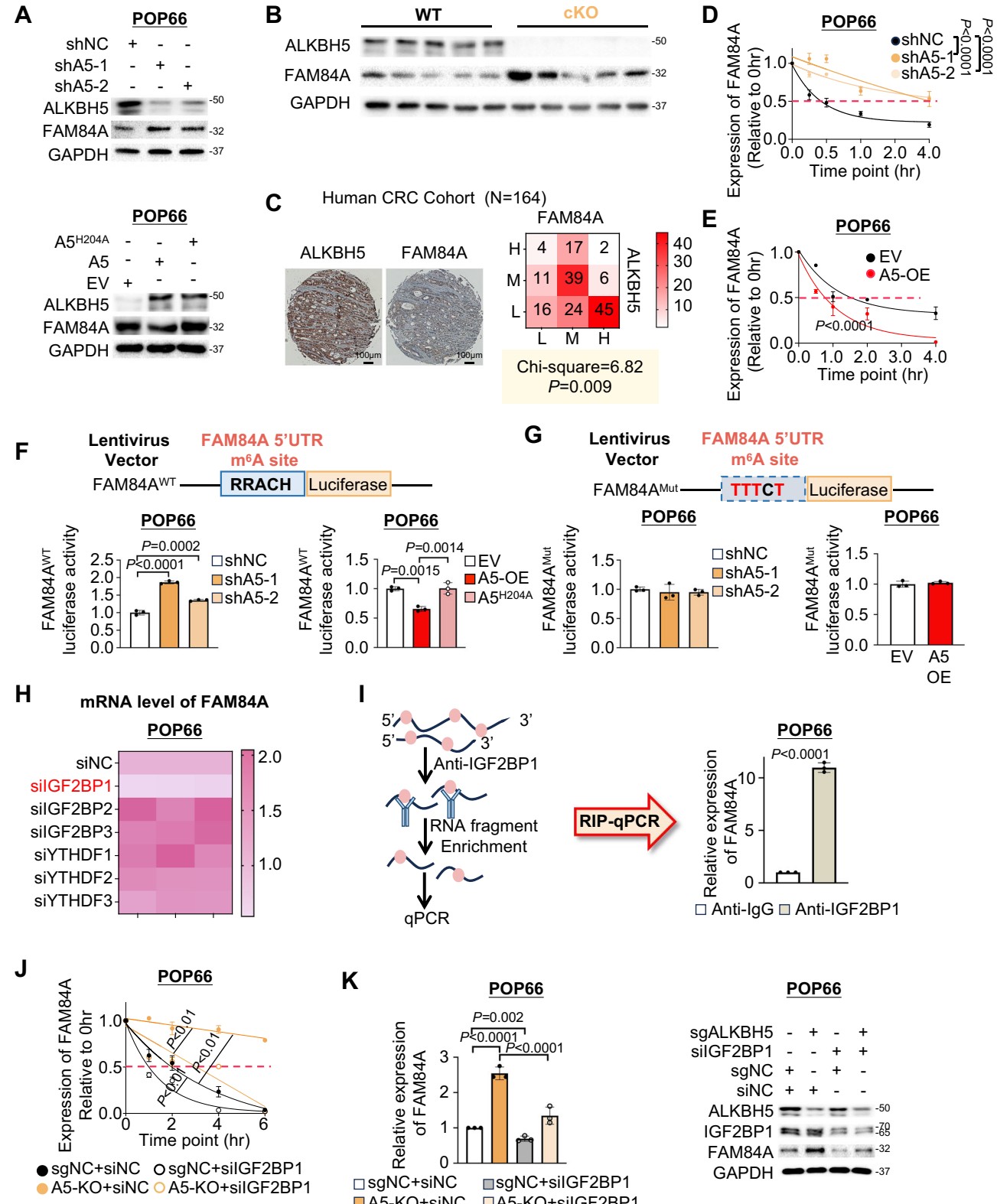

(Fig. 5G and Fig. S6K). These results imply that ALKBH5 impairs FAM84A mRNA stability in a m⁶A-dependent fashion.

The fate of m⁶A-modified transcripts depends on their interplay with m⁶A readers. To unveil m⁶A reader that regulates FAM84A, we knockdown m⁶A readers in ALKBH5-knockout CSCs (Fig. S6L, M). Among m⁶A readers, only IGF2BP1 knockout consistently suppressed FAM84A mRNA in ALKBH5-knockdown CSCs (Fig. 5H and Fig. S6N). We performed RIP assay, which validated the direct binding of IGF2BP1 to FAM84A mRNA (Fig. 5I and Fig. S6O). mRNA decay assay revealed that FAM84A mRNA stability was reduced by IGF2BP1 knockdown in ALKBH5-knockout cells (Fig. 5J and Fig. S6P), confirming IGF2BP1 as a m⁶A reader that stabilizes FAM84A mRNA. Consequently, IGF2BP1 knockdown reversed the induction of FAM84A expression (Fig. 5K and Fig. S6Q) by ALKBH5 knockout. ALKBH5-mediated m⁶A demethylation therefore compromises recognition and stabilization of FAM84A mRNA by

**Fig. 5 | ALKBH5 antagonizes FAM84A mRNA stability by suppressing its interaction with IGF2BP1. A** FAM84A expression as determined by western blot in CSC. **B** FAM84A protein expression as determined by western blot in tumor tissues from transgenic mice. **C** Correlation between ALKBH5 and FAM84A protein expression in the TMA patient cohort ($N = 164$, Chi-square, two-sided). **D** FAM84A mRNA stability as determined by qPCR after ALKBH5 knockdown in POP66. RNA decay rate was normalized to expression at 0 hr. ($n = 3$, each dot represents an independent sample. Two-way ANOVA) **E** FAM84A mRNA stability as determined by qPCR after ALKBH5 overexpression in POP66. RNA decay rate was normalized to expression at 0 hr. ($n = 3$, each dot represents an independent sample. Two-way ANOVA) **F** Plasmid design for the luciferase reporter assay with FAM84A$^{WT}$ 5'UTR (upper panel), luciferase activity of FAM84A$^{WT}$ reporter with ALKBH5 knockdown and FAM84A$^{WT}$ reporter with the overexpression of WT ALKBH5 (A5-OE) or mutant ALKBH5 (A5$^{H204A}$) in POP66 ($n = 3$, each dot represents an independent sample. One-way ANOVA) (lower panel). **G** Plasmid design for luciferase reporter assay with mutated 5'UTR (FAM84A$^{Mut}$) sequences added to the 5' end of luciferase gene (upper panel), luciferase activity of FAM84A$^{Mut}$ reporter with ALKBH5 knockdown and FAM84A$^{Mut}$ reporter with the overexpression of WT ALKBH5 (A5-OE) in POP66 ($n = 3$, each dot represents an independent sample. One-way ANOVA) (lower panel). **H** Heatmap for m6A reader screening. **I** Schematic diagram for RIP-qPCR assay (left panel). Binding between IGF2BP1 protein and FAM84A mRNA was determined by RIP-qPCR in POP66 ($n = 3$, each dot represents an independent sample. Student's t-test, two-sided) (right panel). **J** Effect of IGF2BP1 knockdown on the stability of FAM84A mRNA in POP66 with or without ALKBH5 knockout ($n = 3$, each dot represents an independent sample. Two-way ANOVA). **K** Effect of IGF2BP1 knockdown on FAM84A mRNA ($n = 3$, each dot represents an independent sample. One-way ANOVA) (left panel) and protein (right panel) in POP66 with or without ALKBH5 knockout. Centers and error bars represent mean and Standard deviation, respectively.

IGF2BP1, thus causing accelerated degradation of FAM84A mRNA in CSCs.

## FAM84A impairs tumorigenic potential in colorectal CSCs by mediating β-catenin degradation

Given ALKBH5 negatively regulates FAM84A in colorectal CSCs, we next questioned if FAM84A functions as a stemness repressor. High FAM84A expression is associated with favorable survival and TMA CRC ($N = 193$, $P < 0.05$) (Fig. 6A) and GEPIA ($P = 0.0034$) cohorts (Fig. S7A). Furthermore, FAM84A mRNA negatively correlated with LGR5 and CD133 in cBioportal cohort (Fig. S7B), suggesting that FAM84A might antagonize stemness. To investigate the function of FAM84A in colorectal CSCs, we constructed three FAM84A knockout CSCs. FAM84A knockout in colorectal CSCs enhanced self-renewal capacity (Fig. S7C and Fig. S7D) and increased expression of CSC markers (Fig. S7C and Fig. S7D). Importantly, FAM84A knockout rescued self-renewal capacity in ALKBH5-knockout cells by in vitro LDA (Fig. 6B) and restored LGR5 and CD133 levels (Fig. 6C). Conversely, FAM84A overexpression attenuated viability (Fig. S8), self-renewal (Fig. 6D) and CSC markers (Fig. 6E) in ALKBH5-overexpression cells, validating that FAM84A suppresses stemness traits downstream of ALKBH5 in colorectal CSCs.

To investigate molecular basis of FAM84A, we performed FAM84A pulldown and mass spectrometry to identify potential interacting partners, revealing CTNNB1 as a top protein candidate that interacts with FAM84A (Fig. 6F). Co-immunoprecipitation assay confirmed that FAM84A binds to β-catenin (Fig. 6G). Considering interaction between FAM84A and β-catenin, we asked if FAM84A modulates β-catenin expression. FAM84A knockout increased β-catenin (Fig. 6H and Fig. S9A) and promoted nuclear distribution of active β-catenin (Fig. 6I and Fig. S9B), whilst FAM84A overexpression decreased β-catenin abundance (Fig. 6J and Fig. S9C). Mechanistically, we demonstrated that FAM84A overexpression increased the recruitment of β-catenin destruction complex components GSK-3β and Axin2 to β-catenin (Fig. 6J and Fig. S9C). Hence, FAM84A overexpression induced the ubiquitination of β-catenin (Fig. 6K and Fig. S9D). Supporting involvement of ubiquitin-proteasome system (UPS), proteasome inhibitor MG132 reversed FAM84A-induced down-regulation of β-catenin (Fig. 6L and Fig. S9E). Whereas Chloroquine (CQ) had no rescue effect, suggesting that the lysosomal pathway is not involved in FAM84A-mediated degradation of β-catenin (Fig. S9F). Collectively, we unraveled FAM84A as a CRC stemness suppressor by promoting β-catenin degradation.

## ALKBH5-mediated β-catenin pathway activation is dependent on FAM84A

WNT/β-catenin activation is a hallmark of colorectal CSCs. We reasoned that ALKBH5 might boost β-catenin signaling by down-regulating FAM84A, thereby promoting CSC phenotypes. GSEA based on RNA-seq and ribo-seq consistently revealed that Wnt signaling pathway is downregulated in ALKBH5-deficient CSC28 cells (Fig. S9G). ALKBH5 overexpression up-regulated β-catenin protein expression (Fig. 6M) and decreased β-catenin ubiquitination (Fig. 6N), both of which were partially reversed by FAM84A overexpression. Consistent with in vitro data, tumors from ALKBH5 cKO mice showed decreased β-catenin as compared to WT mice (Fig. 6O). IF validated the proportion of nuclear β-catenin positive cells were increased in ALKBH5 cKI mice compared to WT mice (Fig. 6P). An opposite phenomenon was found in ALKBH5 cKO mice (Fig. 6P).

Cross-linking and mass spectrometry analyses of recombinant FAM84A and β-catenin revealed 2 FAM84A peptides VVQNACGHLGLK (residues 192–203) and NKVHTAR (residues 252–258) interacting with β-catenin (Fig. S9H). Molecular docking showed that ASN-252, VAL-254, LYS-246, PRO-103, and ARG-183 as key amino acid residues involved (Fig. S9I). Contrary to FAM84A$^{WT}$, FAM84A$^{Mut}$ overexpression (residues 252–254; NKV→AAA) failed to down-regulate β-catenin (Fig. S9J) or reduce CSC stemness, as determined by LDA assay (Fig. S9K), validating that residues 252–254 are critical for the ability of FAM84A to bind and inhibit β-catenin function. We thus deciphered an ALKBH5-FAM84A-β-catenin axis for driving CSCs in CRC.

## Targeting ALKBH5 overcomes chemoresistance of colorectal CSCs in vitro

CSCs are the drivers of chemoresistance. ROC curve analysis of ALKBH5 expression in a published cohort (GSE72968) revealed ALKBH5 mRNA predicted nonresponse to chemotherapy with an AUC of 0.71 (Fig. 7A), implying that ALKBH5 might promote chemoresistance. To further ask if ALKBH5 modulates chemotherapy response, we treated CSCs expressing empty vector or ALKBH5 with 5-FU or Oxaliplatin. 5-FU and Oxaliplatin had no effect on ALKBH5 or FAM84A expression (Fig. 7B). IC$_{50}$ showed POP66 and CSC28 with ALKBH5 overexpression were chemoresistant (Fig. 7C). Further, ALKBH5 overexpression abrogated Oxaliplatin and 5-FU-induced apoptosis, as evidenced by flow cytometry (Fig. 7D) and apoptosis markers expression (Fig. 7E). Whereas ALKBH5-depleted CSCs had exacerbated apoptosis after chemotherapy treatment (Fig. 7F, G) with the synergistic induction of cleaved PARP, caspase-7 and caspase-3 (Fig. 7H, I). Oppositely, ALKBH5 knockdown in PDO-816 acted synergistically with Oxaliplatin or 5-FU to suppress tumor growth (Fig. 7J and Fig. S10A) and induce apoptosis (Fig. 7K). Targeting ALKBH5 thus promotes chemosensitivity in colorectal CSCs in vitro.

## Targeting ALKBH5 promotes chemotherapy sensitivity in colorectal CSCs in vivo

To verify our in vitro observations, we assessed Oxaliplatin and 5-FU treatment efficacy in colon stem cell-specific ALKBH5 cKO mice. WT and ALKBH5 cKO were subjected to AOM/DSS-induced CRC, followed by drug treatments (Fig. 8A). Combinations of ALKBH5 cKO plus 5-FU or Oxaliplatin exhibited synergistic suppressive effects on tumor

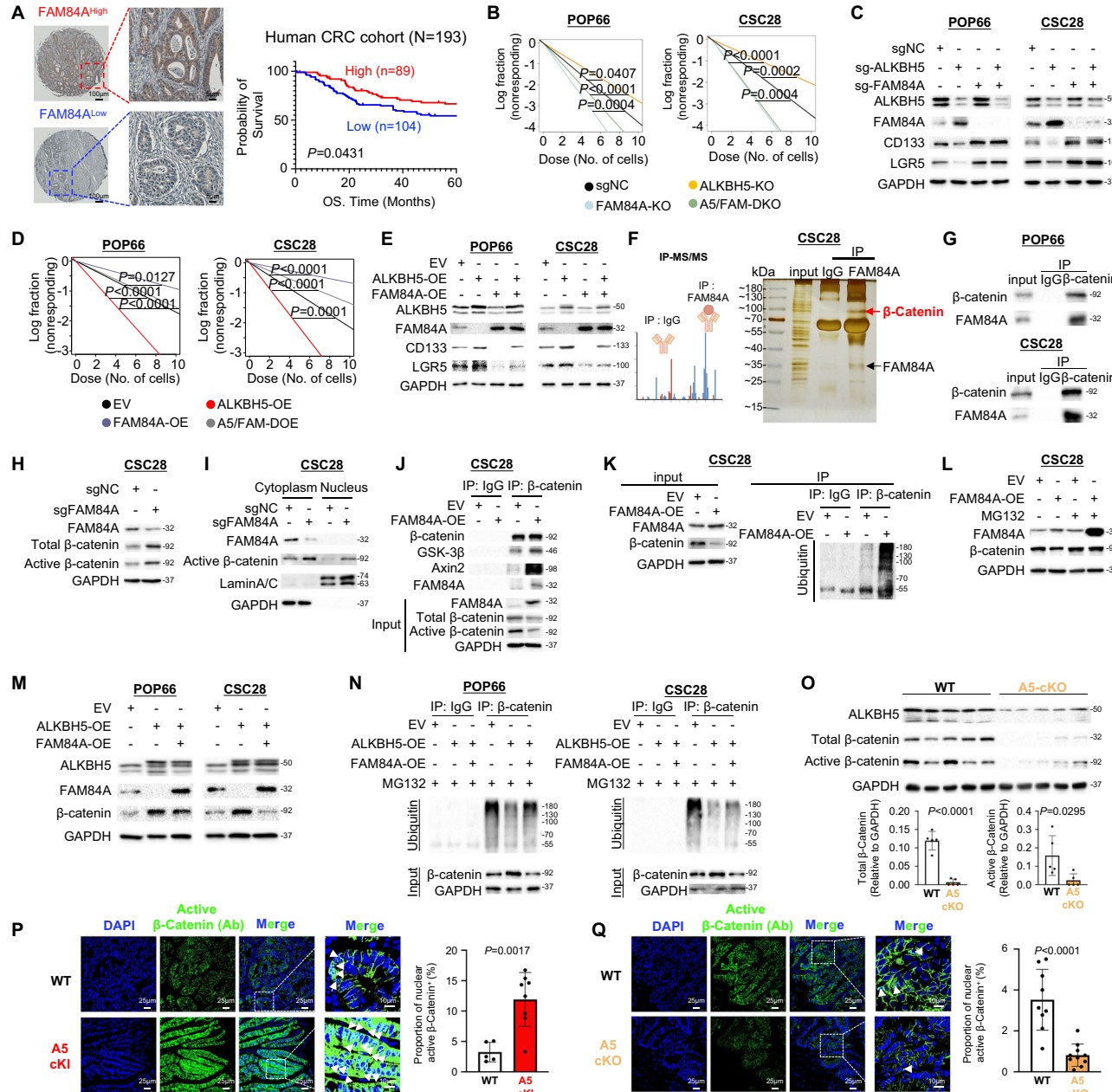

**Fig. 6 | ALKBH5 represses FAM84A-driven β-catenin ubiquitination-mediated degradation in colorectal CSCs. A** Representative images of FAM84A staining in TMA cohort (left panel). Low FAM84A protein expression predicts poor survival of CRC patients (Log-rank test) (right panel). **B** FAM84A knockout rescued the inhibitory effect of ALKBH5 depletion on self-renewal as determined by in vitro LDA in POP66 and CSC28 (Chi-Square, two-sided). **C** FAM84A knockout rescued the inhibitory effect of ALKBH5 depletion on CD133, LGR5 expression as determined by western blot in POP66. **D** ALKBH5-induced self-renewal as determined by in vitro LDA in POP66 and CSC28 (Chi-Square, two-sided). **E** ALKBH5-induced CD133, LGR5 expression was abolished by the ectopic expression of FAM84A in POP66 and CSC28. **F** FAM84A interacting proteins was isolated by co-immunoprecipitation, followed by silver staining and mass spectrometry analysis. The diagram is created in BioRender. Chou, H. (2025) https://BioRender.com/d48babm. **G** The binding between FAM84A and β-catenin as determined by IP. **H** Abundance of total β-catenin and active β-catenin as determined by western blot in CSC28. **I** Abundance of active β-catenin as determined by western blot analysis of cytoplasm and nuclear

protein fractions from CSC28. **J** Interaction between β-catenin with GSK-3β and Axin-2 as revealed by co-immunoprecipitation assay in CSC28. **K** β-catenin expression and ubiquitination in CSC28. **L** Expression of total β-catenin as determined by western blot with or without MG132 treatment in CSC28. **M** β-catenin protein expression in POP66 and CSC28 with ALKBH5 and/or FAM84A overexpression. **N** ubiquitination levels in POP66 and CSC28 with ALKBH5 and/or FAM84A overexpression. **O** β-catenin protein expression in tumor tissues from WT mice and A5-cKO mice (*n* = 5, each dot represents an independent mouse. Student's *t*-test, two-sided). **P** Nuclear active β-catenin levels as determined by IF (anti-active β-catenin antibody) staining in ALKBH5-cKI mice (*n* = 5 for WT mice and *n* = 8 for A5-cKI mice. Student's *t*-test, two-sided) compared with wildtype mice. **Q** Nuclear active β-catenin levels as determined by IF (anti-active β-catenin antibody) staining in A5-cKO mice (*n* = 9 WT mice and *n* = 10 for A5-cKO mice. Student's *t*-test, two-sided) compared with wildtype mice. Centers and error bars represent mean and Standard deviation, respectively.

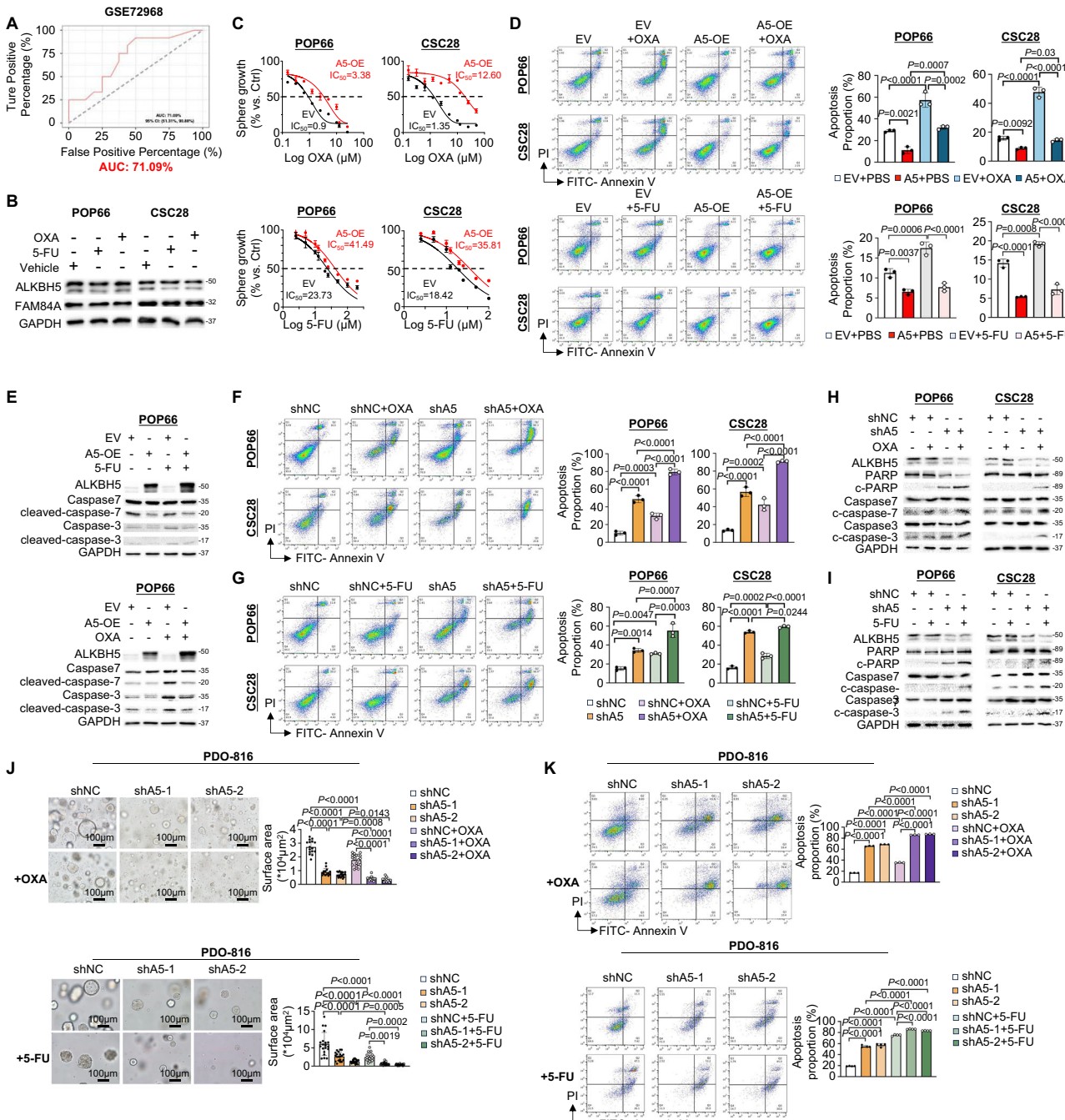

**Fig. 7 | ALKBH5 mediates chemoresistance in colorectal CSCs in vitro. A** ROC curve based on ALKBH5 expression level in chemo-responder and non-responder groups (GSE72968). **B** ALKBH5 and FAM84A protein expression with 5-FU or OXA treatment. **C** IC$_{50}$ of Oxaliplatin (OXA) ($n = 6$, each dot represents an independent sample) (upper panel) and 5-FU ($n = 6$, each dot represents an independent sample for both POP66 and CSC28) (lower panel) as determined by cell viability assay with or without ALKBH5 overexpression in CSC. **D** Flow cytometry of OXA-induced apoptosis in POP66 and CSC28 ($n = 3$, each dot represents an independent sample. One-way ANOVA) with or without ALKBH5 overexpression (upper panel). ALKBH5 overexpression abolished 5-FU-induced apoptosis in POP66 and CSC28 ($n = 3$, each dot represents an independent sample. One-way ANOVA) (lower panel). **E** Expression of apoptosis markers in POP66 with or without ALKBH5-overexpression after 5-FU (upper panel) Oxaliplatin (lower panel) treatment. **F** Flow cytometry of OXA-induced apoptosis in POP66 and CSC28 ($n = 3$, each dot represents an independent sample. One-way ANOVA) with or without ALKBH5

knockdown. **G** Flow cytometry of 5-FU-induced apoptosis in POP66 and CSC28 ($n = 3$, each dot represents an independent sample. One-way ANOVA) with or without ALKBH5 knockdown. **H** Expression of apoptosis markers in POP66 and CSC28 with or without ALKBH5-knockdown after OXA **I** 5-FU treatment. **J** Effect of ALKBH5 knockdown in combination with OXA ($n = 16$ views for shNC, $n = 14$ views for shA5-1, $n = 20$ views for shA5-2, $n = 20$ views for shNC+OXA, $n = 15$ views for shA5-1 + OXA, $n = 12$ views for shA5-2 + OXA. One-way ANOVA) (upper panel) and 5-FU ($n = 20$ views for shNC, $n = 20$ views for shA5-1, $n = 20$ views for shA5-2, $n = 20$ views for shNC+5-FU, $n = 20$ views for shA5-1 + 5-FU, $n = 20$ views for shA5-2 + 5-FU. One-way ANOVA) (lower panel) treatment in PDO-816. **K** Effect of ALKBH5 knockdown in combination of OXA ($n = 3$, each dot represents an independent sample. One-way ANOVA) (upper panel) and 5-FU ($n = 4$, each dot represents an independent sample. One-way ANOVA) (lower panel) by flow cytometry of apoptosis in PDO-816. Centers and error bars represent mean and Standard deviation, respectively.

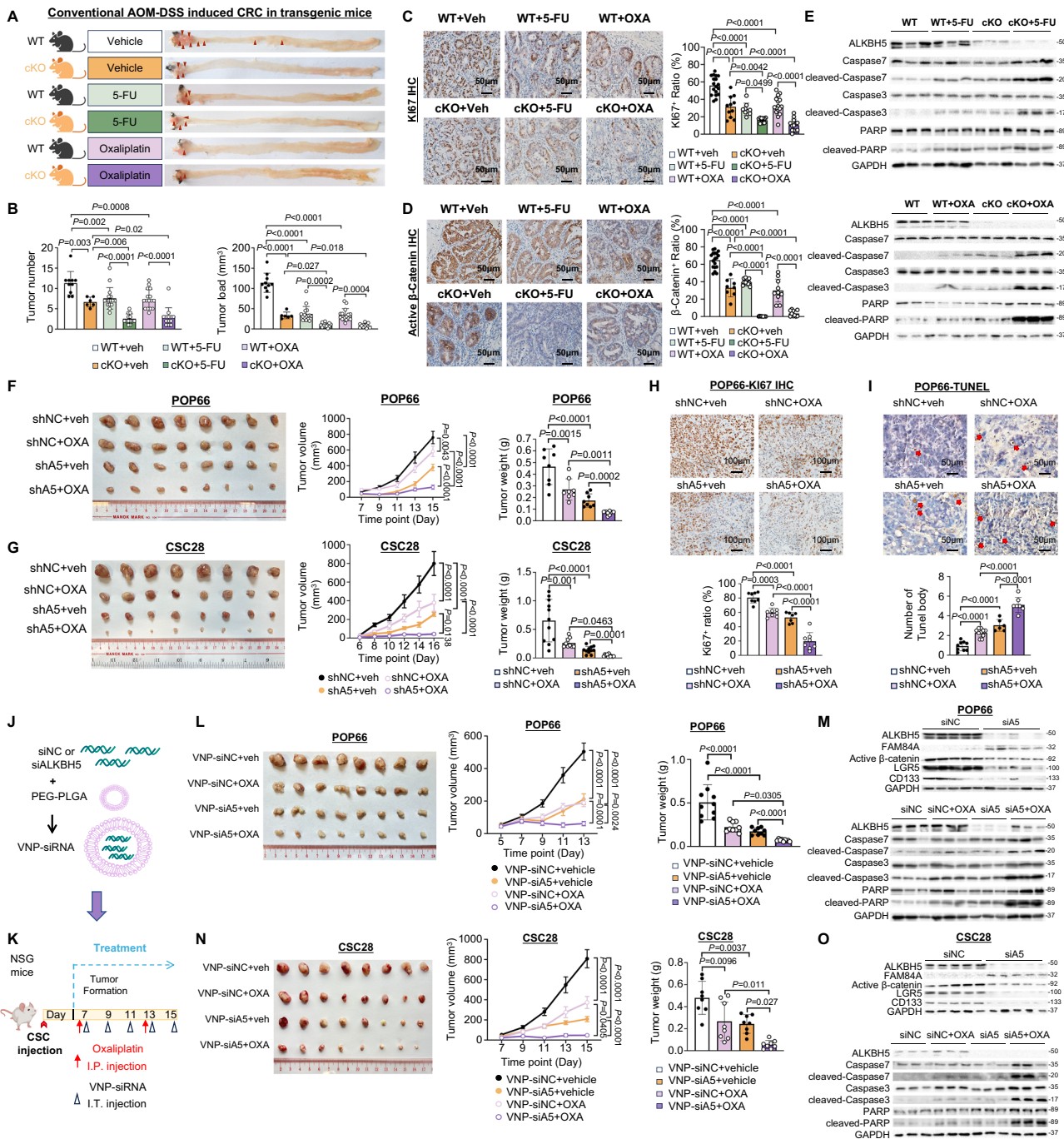

number and load (Fig. 8B). Concordantly, 5-FU or Oxaliplatin in combination with ALKBH5 cKO most effectively inhibited cell proliferation (Fig. 8C), together with drastic down-regulation of β-catenin in tumor tissues (Fig. 8D). Tumor tissues from ALKBH5 cKO mice treated with 5-FU or Oxaliplatin demonstrated synergistic induction of apoptosis markers (Fig. 8E and Fig. S10B, C).

We also implanted CSC spheres with or without ALKBH5 into NOD/SCID-γ mice. Knockdown of ALKBH5 enhanced effect of Oxaliplatin on suppressing tumor growth in POP66 (Fig. 8F and Fig. S11A) and CSC28 (Fig. 8G and Fig. S11B). Consistently, ALKBH5 knockdown plus Oxaliplatin synergistically suppressed cell proliferation (Fig. 8H and Fig. S11C) and promoted apoptosis (Fig. 8I and Fig. S11D). ALKBH5 knockdown also similarly synergized with 5-FU to suppress tumor growth (Fig. S12).

## Nanoparticle-encapsulated siALKBH5 synergizes with chemotherapy to suppress colorectal CSCs in vivo

Since pharmacological inhibitors for ALKBH5 are not available, we therefore utilized a nanoparticle, vesicle-like nanoparticles (VNP), to deliver siALKBH5. VNP based on PLGA is a proven drug carrier, based on biocompatible materials approved by US Food & Drug Administration (FDA)[41]. We constructed VNP-siALKBH5 (Fig. 8J) and validated its knockdown efficiency in vitro (Fig. S13A). To confirm whether the therapeutic efficacy of siALKBH5 depends on altered WNT/β-catenin pathway, we injected CRC organoids from $Apc^{Min/+}Kras^{G12D/+}$ transgenic mice[42] (referred to as AK organoids) into nude mice, followed by treatment with siNC and siA5 nanoparticles (Fig. S13B). As expected, siA5 treatment significantly suppressed tumor growth and tumor weight of AK xenografts compared to siNC (Fig. S13C). Moreover, the

**Fig. 8 | Targeting ALKBH5 enhanced chemotherapy efficacy in vivo.**
**A** Schematic diagram showing the treatment strategy for AOM/DSS-induced CRC in transgenic mice. The diagram is created in BioRender. Chou, H. (2025) https://BioRender.com/uax7xbx. **B** Colon tumor number (One-way ANOVA) and load (One-way ANOVA) at end point. ($n = 10$ for WT, $n = 6$ for A5-cKO, $n = 16$ for WT + 5-FU, $n = 12$ for A5-cKO+5-FU, $n = 15$ for WT + OXA, $n = 13$ for A5-cKO+OXA mice). **C** Determination of Ki67 (anti-KI67 antibody) by IHC stanning of tumor tissues from transgenic mice ($n = 18$ views for WT, $n = 11$ views for A5-cKO, $n = 9$ views for WT + 5-FU, $n = 10$ views for A5-cKO+5-FU, $n = 16$ views for WT + OXA, $n = 13$ views for A5-cKO+OXA mice. One-way ANOVA). **D** Determination of active β-catenin expression (anti- active β-catenin antibody) by IHC stanning of tumor tissues from transgenic mice ($n = 18$ views for WT, $n = 8$ views for A5-cKO, $n = 13$ views for WT + 5-FU, $n = 12$ views for A5-cKO+5-FU, $n = 10$ views for WT + OXA, $n = 10$ views for A5-cKO+OXA mice. One-way ANOVA). **E** Expression of apoptosis markers in WT mice and A5-cKO mice after 5-FU (upper panel) or OXA (lower panel) treatment. ($n = 3$, each dot represents an independent mouse) **F** ALKBH5 knockdown potentiated OXA efficacy in POP66 xenografts as indicated by image of tumor (left panel), tumor growth curve (Two-way ANOVA) (middle panel) and tumor weight (One-way ANOVA) (right panel) ($n = 8$/group). **G** ALKBH5 knockdown potentiated OXA efficacy in CSC28 xenografts as indicated by image of tumor (left panel), tumor growth curve (Two-way ANOVA) (middle panel) and tumor weight (One-way ANOVA) (right panel)

($n = 8$/group). **H** Determination of Ki67 (anti-KI67 antibody) expression by IHC staining of POP66 xenografts ($n = 8$ for shNC, $n = 8$ for shNC+OXA, $n = 8$ for shA5, $n = 8$ for shA5+OXA. One-way ANOVA) **I** Determination of apoptosis by TUNEL staining of POP66 xenografts ($n = 9$ for shNC, $n = 10$ for shNC+OXA, $n = 6$ for shA5, $n = 7$ for shA5+OXA. One-way ANOVA) **J** Construction of VNP-siRNA. **K** The treatment protocol in xenograft model. The diagram is created in BioRender. Chou, H. (2025) https://BioRender.com/4sm73ib. **L** Representative image of POP66 xenografts (left panel), tumor growth curve (Two-way ANOVA) (middle panel) and tumor weight (One-way ANOVA) (right panel) treated with VNP-siALKBH5 and/or OXA ($n = 8$/group). **M** Expression of FAM84A, active β-catenin and stemness markers after VNP-siALKBH5 treatment as determined by western-blot (upper panel) ($n = 5$ independent mouse.) Expression of apoptosis markers (lower panel) in POP66 with or without VNP-siALKBH5 after OXA treatment. ($n = 3$ independent mouse.) **N** Representative image of CSC28 xenografts (left panel), tumor growth curve (Two-way ANOVA) (middle panel) and tumor weight (One-way ANOVA) (right panel) treated with VNP-siALKBH5 and/or OXA ($n = 8$/group). **O** Expression of FAM84A, active β-catenin and stemness markers after VNP-siALKBH5 treatment as determined by western-blot (upper panel). ($n = 5$ independent mouse.) Expression of apoptosis markers (lower panel) in CSC28 with or without VNP-siALKBH5 after OXA treatment. ($n = 3$ independent mouse.). Centers and error bars represent mean and Standard deviation, respectively.

---

addition of WNT inhibitors abrogated the inhibitory effect of siALKBH5 on the growth of primary CRC organoids (Fig. S13D), indicating that effectiveness of targeting ALKBH5 requires activated WNT signaling.

We then treated NSG mice bearing POP66 xenografts with VNP-siNC or VNP-siALKBH5 with or without Oxaliplatin (Fig. 8K). VNP-siALKBH5 plus Oxaliplatin synergistically inhibited POP66 tumor growth and weight (Fig. 8L). ALKBH5 knockdown was validated in VNP-siALKBH5-treated tumors, together with up-regulation of FAM84A and down-regulation of β-catenin, stemness markers (Fig. 8M and Fig. S13E). VNP-siALKBH5 plus Oxaliplatin exacerbated apoptosis, as evidenced by induction of apoptosis markers (Fig. 8M and Fig. S13F). Finally, we validated that VNP-siALKBH5 plus Oxaliplatin also markedly reduced tumor growth (Fig. 8N) along with down-regulation of β-catenin, LGR5 and CD133 (Fig. 8O and Fig. S13G) and induction of cell apoptosis (Fig. 8O and Fig. S13H) in CSC28, indicating in vivo siALKBH5 enhances chemotherapy efficacy in CRC.

## Discussion

Accumulating evidence indicates that m[6]A modification a key determinant in regulating cancer stemness[43]. Here, we identified ALKBH5, an m[6]A eraser, plays a pivotal role in CRC stemness. ALKBH5 is essential for the self-renewal ability of colorectal CSCs in vitro, an observation validated in colon stem cell-specific knockin and knockout mice. We also identified ALKBH5-overexpressed CSCs as drivers of CRC chemoresistance. As a consequence, targeting ALKBH5 sensitized colorectal CSCs to chemotherapeutic drugs, implying ALKBH5 as a potential drug target in CRC treatment.

A series of in vitro and in vivo models clearly demonstrated the pivotal role of ALKBH5 in colorectal CSCs. In CRC patient-derived CSCs and PDOs, ALKBH5 promote self-renewal capacity, together with elevated expression of CSC markers LGR5 and CD133. Thus far, the few reports on the in vitro function of ALKBH5 in CRC cell have been controversial, with studies demonstrating either oncogenic[44,45] or tumor suppressive[46] roles of ALKBH5. Nevertheless, given the multitude of m[6]A-modified transcripts that could be modulated by ALKBH5, the function of ALKBH5 is likely to be highly context-dependent and cell type-specific. To address this, we have established colon stem cell-specific *Alkbh5* knockin and knockout mice. ALKBH5 knockin in colon stem cell lineage exacerbated colorectal tumorigenesis, whereas the opposite phenomenon was observed after the ablation of ALKBH5, suggesting that ALKBH5 functions as an oncogenic factor in a stem cell-selective fashion in CRC tumorigenesis.

To unveil the molecular mechanism through which ALKBH5 boosts stemness in CRC, we performed integrative RNA-seq, MeRIP-seq and Ribo-seq to pin down direct targets of ALKBH5 in CSCs. We revealed a FAM84A mRNA as a direct downstream target of ALKBH5. ALKBH5 binds and demethylates m[6]A-modified FAM84A mRNA, leading to decreased stability of FAM84A mRNA. More specifically, ALKBH5-mediated m[6]A demethylation causes the dissociation of FAM84A mRNA from m[6]A reader IGF2BP1, which would otherwise function to stabilize FAM84A mRNA against degradation. In line with our observations, IGF2BP1 has been shown to bind and stabilize a variety of transcripts in a context-dependent manner[47,48]. As a consequence, the antagonistic impact of ALKBH5 on FAM84A mRNA-IGF2BP1 interplay contributes to the down-regulation of FAM84A in CSCs.

FAM84A, also known as LRATD1, was reported to correlate with morphogenesis and cell motility[49,50]. However, the role of FAM84A in normal physiology and disease remains poorly understood. Here, we unraveled FAM84A as a interaction partner of β-catenin. WNT/β-catenin signaling is a master regulator of self-renewing stem cells in colonic crypts, and its activation is essential for colorectal CSCs[51,52]. We showed that FAM84A behaves as a WNT repressor through the recruitment of β-catenin to the β-Catenin Destruction Complex (GSK3β, Axin-2), resulting in β-catenin ubiquitination and degradation. ALKBH5, by down-regulating FAM84A, thereby reciprocally induces WNT/β-catenin signaling in colorectal CSCs. Reexpression of FAM84A thus abolished ALKBH5-induced active β-catenin and stemness phenotypes. Additionally, analysis of tumors from *Alkbh5* knockin and knockout mice confirmed that ALKBH5 suppressed FAM84A whilst increasing β-catenin and its nuclear translocation in vivo. Collectively, our work deciphered an ALKBH5-m[6]A-FAM84A-β-catenin axis in colorectal CSCs. Consistent with our data, we and others have shown that ALKBH5 positively regulates WNT signaling in CRC[45] and other cancers[53]. ALKBH5 has been shown to regulate WNT signaling in CRC by regulating DKK1[45,54] and WIF-1[55]. Here, we additionally identified FAM84A as a downstream effector for ALKBH5-driven WNT/β-catenin signaling in colorectal CSCs. Besides FAM84A, ALKBH5 was reported to regulate alternative CSC targets, such as FOXM1[28,56], TACC3[34], and NANOG[29], implying that multiple ALKBH5 downstream targets might collaterally contribute to stemness in CRC.

It has long been suggested that CSCs are refractory to chemotherapeutic drugs[57]. We thus reasoned that targeting of ALKBH5 might overcome the chemoresistant phenotype of colorectal CSCs. Indeed, we showed that ALKBH5 depletion enhanced the efficacy of

Oxaliplatin and 5-FU in CSCs, PDOs, and colon stem cell-specific *Alkbh5* knockout mice. Although Oxaliplatin and 5-FU employs different mechanisms to suppress cancer[58,59], both of them culminate in proliferative arrest and apoptosis induction. In this connection, ALKBH5 knockdown act synergistically with both chemotherapeutic drugs to suppress tumor growth and induce apoptosis in CSCs, validating ALKBH5 as a druggable target to reverse chemoresistance in CRC. Moreover, we have designed the VNP-siALKBH5 to directly silence ALKBH5 expression in vivo. VNP-siALKBH5 showed synergistic effects with both 5-FU or Oxaliplatin in inhibiting the growth of colorectal CSCs- and PDOs-derived tumor xenografts, indicating that our VNP-siALKBH5 is a promising adjuvant in the chemotherapy of CRC. While our results provide proof-of-principle results supporting VNP-siALKBH5 as an adjuvant to boost chemotherapy efficacy against colorectal CSCs, several barriers may hamper its clinical application, including the efficient and selective delivery of siRNAs to tumor sites and preventing potential toxic side effects. Further development of small molecules targeting ALKBH5 also represent an alternative approach to target colorectal CSCs.

Although our work in colorectal CSCs, PDOs, and colon stem cell-specific ALKBH5 knockin and knockout mice inferred the role of ALKBH5 in stemness, it has some limitations. Inducible Cre-*lox*P systems, such as LGR5-Cre[ERT2], are known to exhibit variable recombination efficiency. Consistent with this, low-level residual ALKBH5 expression was detected in a subset of Alkbh5[flox/flox]LGR5-Cre[ERT2] mice. The colorectal CSCs and PDOs used in this study harbored activated WNT/β-catenin pathway, a key target of ALKBH5. As a result, these in vitro models are sensitive to ALKBH5 manipulation. Nevertheless, our findings remain representative of human CRC, as 92–97% of CRC tumors exhibit aberrant WNT/β-catenin activation according to the TCGA dataset[60]. Moreover, our assessment of cancer stemness relied on staining of well-defined markers LGR5 and CD133. Given the highly dynamic nature of cellular plasticity, lineage-tracing models and single cell RNA-sequencing would provide a more comprehensive understanding of the stemness-differentiation continuum in tumors[61]. Finally, we did not profile the TME, such as stromal and immune cell populations[62] that might influence CSC behavior. Future studies will be conducted to determine whether CSC-specific ALKBH5 affects stemness through interactions with TME components.

In summary, our study uncovered the critical function of ALKBH5 in colorectal CSCs. Mechanistically, ALKBH5 operates via a m6A-FAM84A-β-catenin signaling axis to activate WNT/β-catenin pathway in colorectal CSCs, thereby boosting tumorigenic potential. ALKBH5 also drives chemoresistance in colorectal CSCs. Targeting ALKBH5 using VNP-siALKBH5 might be a promising adjuvant to improve therapeutic efficacy of chemotherapy in CRC (Fig. 9).

## Methods

### Primary colorectal cancer and adjacent normal tissue samples
A total of 151 paired CRC tumors and adjacent normal tissues for quantitative polymerase chain reaction (PCR) analysis, 12 paired CRC tumors and adjacent normal tissue for western blot. Matched CRC and normal mucosa tissues were obtained from patients with CRC who underwent surgery at Prince of Wales Hospital, The University of Hong Kong. Tissue microarray slides containing CRC cases were kindly provided by Dr. Xiaohong Wang form Peking University Cancer Hospital. The study is approved by human ethics committee of the Chinese University of Hong Kong, and all patients provided informed consent for the clinical samples. The study protocols were approved by the Clinical Research Ethics Committee of Prince of Wales Hospital, the Chinese University of Hong Kong, and Peking University Cancer Hospital. All patients provided written informed consent for obtaining the study specimens. This study was performed in accordance with the declaration of Helsinki of the World Medical Association.

### ALKBH5 knockin and knockout mouse models
All the animal studies were approved by the Animal Experimentation Ethics Committee of the Chinese University of Hong Kong. Conditional *Alkbh5* knockin mice (Rosa26[lsl-Alkbh5]) were generated by Shanghai Model Organisms Company, which were crossed to LGR5-Cre[ERT2] mice to achieve colon stem cell-specific *Alkbh5* knockin mice (Rosa26[lsl-Alkbh5]LGR5-Cre[ERT2], Alkbh5-cKI). To generate the *Alkbh5*[-/-] mice, single-guide RNAs (sgRNAs) targeting exon 1 of Alkbh5 were used, which resulted in Alkbh5 knockout (Alkbh5[flox/flox]). Mice were again crossed to LGR5-Cre[ERT2] mice to achieve colon stem cell-specific *Alkbh5* knockout mice (Alkbh5[flox/flox] LGR5-Cre[ERT2], Alkbh5-cKO). The mice were maintained on a C57BL/6 background. All animal studies are approved by the Animal Experimentation Ethics Committee, CUHK.

### Lipid nanoparticle (LNP) formulation of siRNA drug
siRNA-loaded LNP formulations were formed using microfluidic rapid mixing method as previously reported[63]. For lipid nanoparticle (LNP) formulation of siRNA drug, 1,2-distearoyl-sn-glycero-3-phosphocholine (DSPC), cholesterol and 1,2-dimyristoyl-rac-glycero3-methoxy (poly (ethylene glycol))−2000 (DMG-PEG2000), were purchased from Avanti Polar Lipids, Inc (Alabaster, AL). DLin-MC3-DMA was purchased from Organix, Inc. (Woburn, MA). Lipids were mixed in ethanol at a molar ratio of DLin-MC3DMA/DSPC/Cholesterol/DMG-PEG2000: 50/10/38.5/1.5. 2′-O-Methyl (2′-OMe) modified siRNA was dissolved in a 25 mM acetate buffer (pH = 4.0). The two phases were mixed through herringbone microfluidic chips (microfluidic ChipShop, Germany) at a volumetric flow rate ratio of 3:1 (aqueous to ethanol). The mixed solution was dialyzed against PBS overnight to remove the ethanol and change external pH to 7.4. Afterwards the formulations were passed through a 0.22 μm filter before use. The hydrodynamic size and dispersity of LNP was characterized by a Zetasizer Nano ZS (Malvern Instruments, United Kingdom). siRNA encapsulation efficiency was measured by Ribogreen Assay (Thermo Fisher Scientific).

### AOM/DSS-induced CRC
Mice at 6–8 weeks of age were intraperitoneally injected with tamoxifen (75 mg/kg body weight; 8 days) to activate Alkbh5 knockout or overexpression system. Then, mice were intraperitoneally injected with Azoxymethane (AOM) (10 mg/kg body weight; 5 days) (Sigma-Aldrich). 2.5% dextran sulfate sodium (DSS) (MP Biomedicals) in drinking water were followed for 5 days. DSS treatments were repeated for 3 cycles, and mice were harvested on day 80. Only male mice were used in this study. Randomization was carried out as follows. Animals were assigned a group designation and weighted. Mice were divided into different weight groups. Each animal was assigned a temporary random number within the weight range group. Humane endpoints were triggered by >20% body weight loss. All the chemicals are listed in Supplementary Table 7.

### RNA-sequencing
Colorectal CSC sphere with ALKBH5 knockdown and overexpression were harvested. RNA was isolated using Trizol reagent and treated with DNase I. Quality control was further checked. sequenced using Illumina PE150. RNA sequence reads were mapped on human reference (GRCh38) by HISAT2 (v.2.1.0). Mapped reads were counted by HTSeq (v.0.11.2). HTSeq v0.6.0 was used to count the reads numbers mapped to each gene. And then FPKM of each gene was calculated based on the length of the gene and reads count mapped to this gene. The raw sequencing data generated in this study have been deposited in NCBI Sequence Read Archive under BioProject PRJNA1086204.

### MeRIP-sequencing
Total RNAs were extracted with Trizol reagent and DNA contamination was removed by DNase I. The total RNA was chemically fragmented into ~200-nt-long fragments. The fragmented RNAs were then

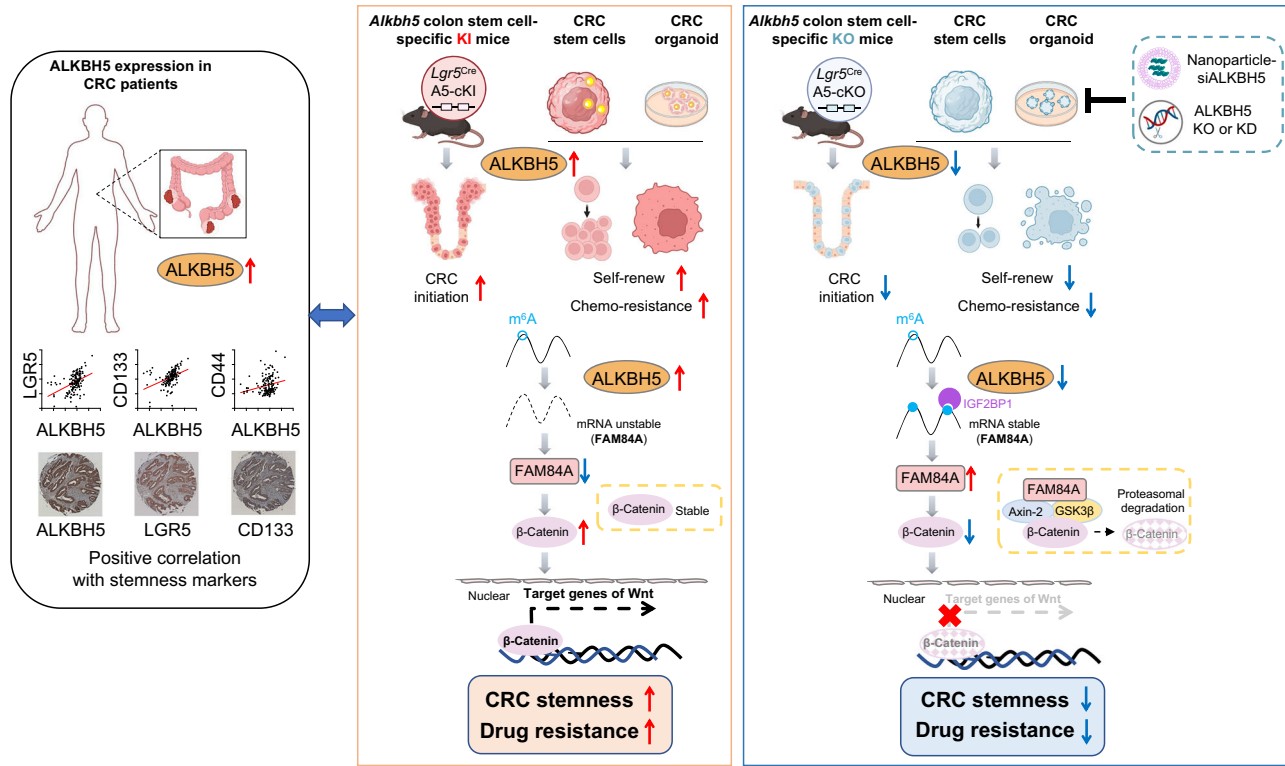

**Fig. 9 | Graphical abstract.** The overall graphical summary of the study (Created in BioRender. Chou, H. (2025) https://BioRender.com/rlno4qj).

incubated with the Protein A/G Magnetic Beads and tumbled with anti-m6A antibody at the presence of RNase inhibitor. Then the eluted RNA was reverse transcribed with High-Capacity cDNA Reverse Transcription Kit. Then the sequencing was performed on an Illumina Novaseq 6000 platform. The raw sequencing data generated in this study have been deposited in NCBI Sequence Read Archive under BioProject PRJNA1086651 and PRJNA1086679.

### Ribo-sequencing
The Ribo-sequencing was followed as described in an established strategy. Briefly, Cycloheximide was added to cell cultures medium to a final concentration of 100 μg/ml. To prepare ribosome footprints (RFs), 10 μL of RNase I and DNase I were added to lysate. RFs were purified using magnet beads. After obtaining the RFs above, Ribo-seq libraries were constructed by NEBNext® Multiple Small RNA Library Prep Set for Illumina® (catalog no. E7300S, E7300L). The 140–160 bp size PCR products were enriched to generate a cDNA library and sequenced using Illumina HiSeqTM X10. The raw sequencing data generated in this study have been deposited in NCBI Sequence Read Archive under BioProject PRJNA1086651.

### Establishment of CRC organoids from primary CRC in transgenic mice
Organoid cultures were established from mice primary CRC tissue. Briefly, when mice were sacrificed, primary tumor tissues from colon were isolated. Tumor tissues were washed by PBS supplemented with 0.1 mg/ml Penicillin-Streptomycin antibiotics. Tumor tissues were then moved to a tube contained with digestion buffer and incubated for 45 min in 37 °C, shaking to release the stem cells into solution. organoids were separated from the whole tumor matrix by filtering the solution using 70μm cell strainers, followed by 2 washes with Advanced DMEM/F12 supplemented with 10 mM HEPES, 0.1 mg/ml Penicillin-Streptomycin antibiotics and 1mM N-Acetylcysteine. Finally, organoids were resuspended in Matrigel (#356231, Corning) plated on 24-well culture plates and overlaid with full medium to initiate

organoid growth. Organoids were grown at 37 °C in 5% $CO_2$ incubator. During the first passage, culture medium was changed once every 2 days. Organoid were passaged at day 7. All the chemicals are listed in Supplementary Table 7.

### Organoid culture for PDO-816 and PDO-828
Organoid medium for PDO-816 and PDO-828 was prepared from 1X Advanced DMEM/F12, 10 mM HEPES, 0.1 mg/ml Penicillin-Streptomycin antibiotics, 2 mM Glutamax, 2% B-27 Supplement (Thermo Fisher Scientific), 1.25mM N-Acetylcysteine, 500 nM A8301 (a potential inhibitor of TGF-β type I receptor) in DMSO, 10 μM Y27632, 50 ng/ml mEGF, 10 nM Gastrin I and 10% v/v Noggin. Aliquots of 50 mL were stored at −80 °C and were thawed prior mixing the medium.

Medium for CRC primary tumor organoid was prepared from 1X Advanced DMEM/F12, 10 mM HEPES, 0.1 mg/ml Penicillin-Streptomycin antibiotics, 100 μg/ml Primocin, 2 mM Glutamax, 2% B-27 Supplement (Thermo Fisher Scientific), 1% N2 supplement, 1.25mM N-Acetylcysteine, 500 nM A8301 (a potential inhibitor of TGF-β type I receptor) in DMSO, 10 μM SB202190 (a selective p38 MAP kinase inhibitor) in DMSO, 10 mM Nicotinamide, 10 μM Y27632, 50 ng/ml mEGF, 100 ng/ml FGF10, 10 nM Gastrin I, 10% v/v mNoggin, 100 ng/ml mNoggin protein, 10% v/v R-spondin and 50% v/v Wnt3a CM. All the chemicals are listed in Supplementary Table 7.

### Subcutaneous colorectal CSCs xenograft assays
For xenograft mouse models, colon CSCs transfected with lentivirus-carrying shNC and shALKBH5-1 were injected subcutaneously into dorsal flank of 3–4 week-old male NOD scid gamma (NSG) mice ($5 \times 10^5$ cells in 100uL phosphate-buffered saline/mouse), respectively. Tumor size was recorded every 2 days by digital caliper. Tumor volume was calculated as follows: volume = (short diameter)$^2$* (long diameter)/2. At the endpoint, tumors were harvested and weighted. To investigate the synergistic effect of 5-FU or oxaliplatin and shALKBH5 on colon CSC growth in vivo, POP66 and CSC28 ($5 \times 10^5$ cells) were subcutaneously

injected into the dorsal flank of 3–4-week-old male NSG mice. Humane endpoints were triggered by: (1) tumor volume≥2000mm³ or (2) > 20% body weight loss. 5-FU (MCE No. HY-90006) was dissolved in 10% DMSO and diluted in 40% PEG, 5% Tween-80 and 45% saline solution. Mice were administered with 5-FU intraperitoneally (50 mg/kg body weight; once every 2 days 5 days) after CSC injection. Oxaliplatin (MCE No. HY-17317) was dissolved in 10% DMSO and diluted in 5% w/v glucose solution. Mice were administered with oxaliplatin intraperitoneally (3.5 mg/kg body weight; once every 2 days 5 days) 5 days after CSC injection. Mice in control group were injected with vehicle. All animal studies were performed in accordance with the guidelines approved by the Animal Experimentation ETHICS Committee of CUHK. Randomization was carried out as follows. Animals were assigned a group designation and weighted. Mice were divided into different weight groups. Each animal was assigned a temporary random number within the weight range group. All the chemicals are listed in Supplementary Table 7.

## Cell lines and organoids

Human colon CSC-enriched spheroid models (POP66 and CSC28) and colon cancer PDOs (816 and 828) were kindly provided by Dr. Catherine Adell O'Brein in Princess Margaret Cancer Center, University of Toronto. POP66 and CSC28 were isolated from liver metastases species from colon adenocarcinoma and cultured in 3D suspension state in previously established serum-free, growth factor-enriched medium which suitable for CSCs. PDO-828 was isolated from a primary colon tumor and PDO-816 was isolated from lung metastases of colon cancer. PDO-816 and 828 were cultured in 3D serum-free Matrigel (Corning, NY, USA) in advanced DMEM/F12 medium (Gibco, CA, USA).

## Plasmids and siRNAs

The full-length FLAG-tagged ALKBH5 cDNA (NM_017758.4) was amplified and cloned into lentiviral vector pLVX-Puro (Takara Bio USA). The full-length FLAG-tagged FAM84A cDNA (NM_001369364.1) was amplified and cloned into lentiviral vector pLVX-Puro (Takara Bio USA). ALKBH5 and negative control (shNC) shRNAs were cloned into lentivirus shRNA expression plasmid pLVshRNA-puro (Inovogen Tech. Co.). The shRNA target sequences were as followed: shALKBH5-1: 5'-AAAGGCTGTTGGCATCAATA-3', and shALKBH5-2: 5'-CCACCCAGC-TATGCTTCAGAT-3'. ALKBH5, FAM84A and negative control (sgNC) sgRNAs were cloned into lentivirus vector Lent-Cas9-Puro. gRNAs are listed in Supplementary Table 3.

For siRNA knockdown, cells were infected with YTHDF1-3 and IGF2BP1-3 siRNAs or control siRNA (siRNAs are listed in Supplementary Table 1).

## Cell viability assay and in vitro Limiting dilution assay (LDA)

For cell viability assay, CSC spheroids were digested into single cell suspension, counted, cells ($1 \times 10^3$/well) were seeded in a 96-well plate and MTT assay (5 mg/mL; Promega) was performed according to the manufacturer's protocol. For in vitro LDA, CSC spheroids were digested into single cell suspension, then cells were cultured in 96-well plates at densities of 10, 5, 2 or 1 cell/well. About 1 week later, wells containing spheres were counted and the results were analyzed by an online tool (ELDA, https://bioinf.wehi.edu.au/software/elda/index.html).

## IHC staining

Paraffin section at 4 μm thickness were dried in a 60 °C oven for 30 min before staining. IHC was performed according to the manufacturer's guidelines. In brief, slides were deparaffinized with xylene and rehydrated by a gradient alcohol series, followed by antigen retrieval with sodium citrate buffer. The slides were blocked by goat serum. The dilutions for ALKBH5 (HPA001796, Sigma-Aldrich) were diluted as

1:500, FAM84A (sc-101207, Santa Cruz Biotechnology) were diluted as 1:200, CD133 (#5860, CST), LGR5 (#TA503316, Origene) were diluted as 1:200 for human tissue microarray. For the slides from CRC patient tissue microarray, sections were evaluated by two pathologists. FAM84A immunohistochemistry staining was scored by a pathologist on a scale from 0 to 3, where 0 = negative; 1 = weak positive; 2 = moderate positive; and 3 = strong positive. In Fig. 6A, scores of 0–1 and 2–3 were defined as FAM84A-low and FAM84A-high, respectively. The staining for ALKBH5 (HPA001796, Sigma-Aldrich) were diluted as 1:500, Ki67 (#9129, CST) were diluted as 1:200, active β-catenin (#19807, CST) were diluted as 1:500 for transgenic mice tissue.

For the slides from xenograft mouse model, Ki67 (#9129, CST) were diluted as 1:200.

The primary antibodies were incubated for 4 overnight. Rabbit on rodent HRP-polymer antibody (#RMR622H, Biocare) was added to slides for 30 min at room temperature. The primary-secondary complex was then visualized with DAB. The slides were counterstained with Harris hematoxylin, dehydrated in graded alcohol, cleared in xylene and coverslipped in Permount. All the positive ratio was recognized and calculated by Qupath software (https://qupath.github.io/).

## IF staining

IF staining was performed according to the manufacturer's guidelines. Briefly, after antigen retrieval, slides were blocked by 1% BSA containing 0.3% Triton X-100 in PBS for 1 h at room temperature and incubated with anti-GFP (#ab6556, Abcam) was diluted as 1:500, anti-Cd133 (#ab271092, Abcam) was diluted as 1:100 at 4 °C overnight. Then the slides were washed 3 times with 0.1% Tween-20 in PBS (PBST). The slides were incubated with Alexa Fluor 488 secondary antibody (1:500, Life Technologies). Then wash the slides by PBST for 3 times. Cover slip slides with DAPI (#8961, CST) and take images with Multiphoton Microscope (Leica TCS SP8).

## TUNEL assay

TUNEL assay were conducted in paraffin-embedded tumor sections using the TUNEL Assay Kit (ab206386, Abcam, UK). Cell apoptosis index was determined by the total number of TUNEL positive cells in tumor tissues.

## MeRIP-qPCR and RIP-qPCR

MeRIP and RIP was performed by EZ-Magna RIP™ RNA-Binding Protein Immunoprecipitation Kit(17–700, Sigma) according to the manufacturer's instructions. In brief, cell pellets were lysed with the RIP Lysis Buffer. The cell lysates were then incubated with magnetic beads bound with anti-m6A (for MeRIP-qPCR), anti-ALKBH5 (for RIP-qPCR), anti-IGF2BP1(for RIP-qPCR) antibody overnight at 4 °C. The beads were then washed with RIP Wash Buffer for six times and the RNAs were released by digesting the antibody with proteinase K in 1% (w/v) SDS at 55 °C for 30 min with gentle shaking. Then the RNAs were isolated by phenol: chloroform: isoamyl alcohol (BP1754I-100, Fisher) and RNA precipitation by ethanol. The resuspended RNAs were further performed RT-qPCR. Primers are listed in Supplementary Table 2.

## MeRIP-qPCR calculation

M⁶A pulldown assay was conducted using the Magna RIP® RNA-Binding Protein Immunoprecipitation Kit (#17-700). Following the kit's protocol, 10% of the RNA was reserved for FAM84A from each sample (shNC, shA5-1, shA5-2; EV, ALKBH5$^{WT}$, ALKBH5$^{H204A}$) as input. Normalized m⁶A abundance was calculated as follows:

$$\Delta CT_{IgG} = CT_{IgG} - CT_{input}$$

$$\Delta CT_{MeRIP} = CT_{MeRIP} - CT_{input}$$

$$\Delta\Delta CT_{Normalized\ MeRIP} = \Delta CT_{MeRIP} - \Delta CT_{IgG}$$

m⁶A abundance for each sample = 2^($-\Delta\Delta CT_{Normalized\ MeRIP}$)

## Luciferase reporter assay

To check the regulation of ALKBH5 on FAM84A, luciferase reporter assay was performed with Dual-Luciferase Reporter Assay System (Promega) following the manufacturer's description. M⁶A-modified fraction of FAM84A and m⁶A-mutant FAM84A was cloned into luciferase reporter plasmid which carries Renilla Luciferase element as internal control, respectively. Empty vector or m⁶A-modified FAM84A and m⁶A-mutant FAM84A were transfected into cells using FuGENE (Promega). The relative luciferase activities were tested a4h post transfection by Dual-L uciferase Reporter Assay System (Promega).

## Translation efficiency calculation

In Fig. S5G, translation efficiency is measured by FPKM value from ribo-seq normalized to FPKM value from bulk RNA-seq for each gene. In Fig. S6H, translation efficiency was measured by simultaneously extracting RNA and proteins from shNC, shA5-1, and shA5-2 cells. qPCR and Western blot were performed to determine mRNA and protein expression of FAM84A and GAPDH, respectively. The expression of FAM84A was normalized to GAPDH, and then the ratio of protein-to-mRNA expression is taken as a measure of translation efficiency.

## RNA stability assay

Colorectal CSC spheres with ALKBH5 knockdown (with or without si-NC/si-m⁶A readers) or ALKBH5 overexpression were cultured in non-treated 6-well plate. Harvesting the cells from one well as the 0 time point. 20 μL of 1 mg/mL Actinomycin D (Sigma, 3A9415) stock to a final concentration of 10 μg/mL in 2 mL of culture media. Then collecting cell pellets at remaining time points. Further RNA extraction and qPCR were performed. Primers are listed in Supplementary Table 2.

## Anti-FLAG pulldown and protein identification

For the screening of FAM84A interact proteins, the extracted FAM84A^FLAG-overexpressing CSC spheres protein lysates were incubated with either anti-FLAG antibody (#2368, CST) or IgG for 4 h at room temperature. Then the antibody-bound proteins were incubated with protein A/G magnetic beads overnight at 4 °C. Then the complex was washed with lysis buffer 5 times on the magnetic rack. The FLAG-interacting proteins were eluted by heating at 100 °C in 50 μl 2X loading buffer. Eluted proteins were subjected to SDS-PAGE and the whole gel was further performed silver staining (Thermo Fisher Scientific) according to the manufacturer's instructions. The corresponding bands ($n = 1$ for each band) detected in silver staining were cut down for mass spectrometry protein identification.

For the in-gel digestion, the entire slab of the SDSPAGE gel was rinsed twice with 1000 μL ddH2O for 1 h. Bands (spots) of interest were excised with a clean scalpel and cut into cubes (about 1 mm). Gel pieces were transferred into a new low binding Eppendorf tube and spined down with microcentrifuge. Add 1000 μL decolorization solution to decolorize until the colloidal particles are colorless (replace the decolorization solution several times), remove the decolorization solution. Gels were then incubated with 800 μL of neat ACN for 10 min until gel pieces shrink (they became opaque and stick together).

After removing of all liquids, 600 μL of 5 mM DTT solution (in 100 mM NH4HCO3) was added to completely cover gel pieces for 30 min at 55 °C. Tubes chilling down to room temperature, incubated with 800 μL of ACN for 10 min. For alkylation, 600 μL of 15 mM IAA solution (in 100 mM NH4HCO3) was incubated with gels for 40 min at room temperature in the dark. Alkylated protein gels were shrinked with ACN and collected with microcentrifuge. Trypsin digestion was preformed by saturating the gels for 30 min with 600 μL trypsin (200 ng in 100 mM NH4HCO3) buffer, then additional trypsin buffer covering the gels for 30 min, then incubating over night at 37 °C. Peptides were extracted with 800 μL of extraction buffer (1:2 (vol/vol) 5% formic acid/ACN) and vaccum dried. Peptides were resuspended with 0.1% formic acid, desalted with C18 tip and stored in 0.1% formic acid for LCMS analysis.

For nanoLC-MS analysis, each sample, 2 μg of total peptides were separated and analyzed with a nanoUPLC (nanoElute2) coupled to a timsTOF Pro2 instrument (Bruker) with a nanoelectrospray ion source. Separation was performed using a reversedphase column (75 μm ID ×25 cm, IonOpticks C18CSI, 1.6 μm, Aurora Ultimate). Mobile phases were H2O with 0.1% FA (phase A) and ACN with 0.1% FA (phase B). Separation of sample was executed with a 30 min gradient at 300 nL/min flow rate. Gradient B: 2% for 0 min, 222% for 20 min, 2237% for 4 min, 3780% for 2 min, 80% for 4 min.

The mass spectrometer adopts DDA PaSEF mode for DDA data acquisition, and the scanning range is from 100 to 1700 m/z for MS1. During PASEF MS/MS scanning, the impact energy increases linearly with ion mobility, from 20 eV (1/K0 = 0.6 Vs/cm2) to 59 eV (1/K0 = 1.6 Vs/cm2).

Vendor's raw MS files were processed using SpectroMine software (4.1.230421.52329) and the built-in Pulsar search engine. MS spectra lists were searched against their specieslevel UniProt FASTA databases (uniprot-Homo sapiens-9606-2022-11.fasta), Carbamidomethyl [C] as a fixed modification, Oxidation (M) and Acetyl (Protein Nterm) as variable modifications. Trypsin was used as proteases. A maximum of 2 missed cleavage(s) was allowed. The false discovery rate (FDR) was set to 0.01 for both PSM and peptide levels. Peptide identification was performed with an initial precursor mass deviation of up to 10 ppm. Unique peptide and Razor peptide were used for protein quantification and total peptide amount for normalization. All the other parameters were reserved as default.

## Co-IP

Protein lysates from colorectal CSC spheres were incubated with either anti-β-catenin (#9582, CST) or IgG for 4 h at room temperature. Then the antibody-bound proteins were incubated with protein A/G magnetic beads overnight at 4 °C. Then the complex was washed with lysis buffer 5 times on the magnetic rack. The FLAG-interacting proteins were eluted by heating at 100 °C in 50 μl 2X loading buffer. Eluted proteins were subjected western blot and test target protein.

## Cross-linking/mass spectrometry analysis

Cross-linking/MS was performed according to the protocol[64]. Briefly, Sample pretreatment began with dissolving the samples in 50 μL of water for later use. Subsequently, the cross-linking reaction was performed by mixing 3 μL of the FAM84A sample with 9 μL of the β-Catenin sample, followed by incubation at room temperature for 15 min. Then, 0.55 μL of 20 mM cross-linker DSS was added, and the mixture was incubated at room temperature with shaking for 1.5 h. The reaction was quenched by adding 1.15 μL of 500 mM ammonium bicarbonate, followed by continued shaking incubation for 20 min. Next, pre-chilled acetone was added to the cross-linked sample, and the mixture was precipitated at −20 °C for 2 h. Afterward, the sample was centrifuged at 4 °C and 12,000 × g for 20 min, and the supernatant was discarded. The protein precipitate was washed twice with pre-chilled acetone and dried. The dried precipitate was redissolved in 12.5 μL of urea lysis buffer (containing 8 M urea and 50 mM Tris-HCl, pH 7.4) and sonicated for 2 min. It was then diluted with 87.5 μL of 50 mM NH4HCO3 solution to reduce the urea concentration to 1 M. Subsequently, DTT was added to a final concentration of 10 mM and incubated at 56 °C for 60 min for reduction. Then, IAM solution was added to a final concentration of 20 mM and incubated at room temperature in the dark for 30 min

for alkylation. The reaction was terminated by adding DTT to a final concentration of 10 mM. Next, Trypsin BE (at a concentration of 0.25 μg/μL) was added and digested at 37 °C for 16 h, followed by a secondary digestion with 5 μL of 0.5 μg/μL Glu-C for an additional 2 h. The digested samples were desalted using a desalting column, dried by vacuum centrifugation at 45 °C, and finally, the peptides were redissolved for LC-MS/MS analysis.

LC-MS analysis was performed using a Vanquish Neo-Orbitrap Fusion Lumos high-resolution LC-MS/MS system. Chromatographic separation was achieved using a capillary column with an inner diameter of 150 μm, length of 170 mm, and packed with 1.9 μm Reprosil-Pur 120 C18-AQ particles. The column temperature was maintained at 60 °C. Mobile phase A consisted of 0.1% formic acid in water, and mobile phase B consisted of 0.1% formic acid in 80% acetonitrile. The flow rate was set at 0.6 μL/min, and the following gradient program was executed: 0–2 min, mobile phase B increased from 4 to 8%; 2–35 min, increased from 8 to 28%; 35–55 min, increased from 28% to 40%; 55–56 min, rapidly increased from 40 to 95%, and maintained at this concentration until 66 min.

Mass spectrometry data acquisition was performed in full scan mode with a resolution of 120,000, a scan range of 300–1800 m/z, using the Profile data format, and a maximum injection time of 20 ms. Data-dependent MS2 fragmentation was performed in dd-MS2 mode with a resolution of 15,000, using a stepped collision energy of 30, and a maximum injection time of 22 ms.

The acquired data were finally analyzed using pLink 2.3.11 software.

## Molecular docking analysis

Molecular docking analysis was performed to determine FAM84A and β-Catenin interactions. The three-dimensional structures of FAM84A was based on Uniprot dataset (ID: Q96KN4) and β-Catenin structure was based on RCSD PDB dataset (ID: 1JDH). Protein was pretreated using PYMOL to remove water molecules, ions and add hydrogens. The treated protein structure was visualized by PYMOL. The interaction diagrams were analyzed HDOCK.

## Cytoplasm/Nuclear fraction protein isolation

Colorectal CSC spheres were collected and centrifuged at $500 \times g$ for 5 min. Washing and suspending the spheres with PBS. NE-PER™ Nuclear and Cytoplasmic Extraction Kit (Thermo Fisher Scientific) was used according to the manufacturer's instructions. All protein extractions were put on ice for use or stored at −80 °C.

## Western blot

Protein was extracted using the CytoBuster protein extraction kit (Novagen, Austin, TX) containing protease inhibitors (Roche) and phosphatase inhibitors (Roche). The antibodies used in this study were listed in Supplementary Table 6_Antibody list.

## ROC curve analysis

The samples from GSE72968 dataset were randomly split into training and validation cohorts at a ratio of 7:3. Based on ALKBH5 mRNA expression, a random forest model was trained to distinguish chemotherapy responders and non-responders on the training cohort and tested on the validation cohort using the caret R package[65]. Receiver operating characteristic curves were generated by pROC R package for performance evaluation[66].

## Single-cell RNA-seq analysis

63689 total cells were identified and labeled as epithelial, mast, myeloid, stromal, B, and T cells using previously described from GSE132465. 17469 individual tumor cells expressed Epcam were extracted for further analysis. Then Epcam$^+$ cancer cells were separated as LGR5-expressed cells (LGR$^{pos}$,14965 cells) and cells that did not express LGR5 (LGR5$^{neg}$, 2504 cells). ALKBH5 expression level were compared between LGR$^{pos}$ and LGR5$^{neg}$ group.

## Statistical analysis

Statistical analysis was performed by Prism Software version 9 (GraphPad Software; San Diego, CA), and the data were shown as mean ± s.d., unless stated otherwise. For each figure, $n$ = the number of independent biological replications. Comparisons between two groups were performed by a two-sides Student's $t$-test. Analysis of variance (ANOVA) was used to compare differences among multiple groups. $P < 0.05$ indicate statistical significance. The statistical details of experiments can be found in each Fig. legends.

## Reporting summary

Further information on research design is available in the Nature Portfolio Reporting Summary linked to this article.

## Data availability

All RNA-seq, MeRIP-seq and ribo-seq data have been deposited in the Sequence Read Archive (SRA) under accession number PRJNA1086651, PRJNA1086679 and PRJNA1086204.

https://www.ncbi.nlm.nih.gov/sra/?term=PRJNA1086651
https://www.ncbi.nlm.nih.gov/sra/?term=PRJNA1086679
https://www.ncbi.nlm.nih.gov/sra/?term=PRJNA1086204

The mass spectrometry proteomics data, Cross-linking/mass spectrometry proteomics data have been deposited to the ProteomeXchange Consortium via the iProX partner repository[67,68] with the dataset identifier PXD071162.

http://proteomecentral.proteomexchange.org/cgi/GetDataset?ID=PXD071162
https://www.iprox.cn/page/project.html?id=IPX0014377000
http://proteomecentral.proteomexchange.org/cgi/GetDataset?ID=PXD071348
https://www.iprox.cn/page/project.html?id=IPX0014431000

The remaining data are available within the Article, Supplementary Information or Source Data file. Source data are provided with this paper.

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

## Acknowledgements

This project was supported by RGC-GRF Hong Kong (14108823, 14111621), RGC-CRF Hong Kong (C4008-23W, C4039-19GF), RGC-ECS Hong Kong (24100520), National Natural Science Foundation of China (NSFC; 81972576), The Kingboard Precision Oncology Program, CUHK.

## Author contributions

Conceptualization: H.M.Z. Methodology: H.M.Z., H.R.C., W.M.L., S.J.C., S.Y.W., W.X.L., C.L., Y.W. Investigation: H.M.Z., C.C.W., H.R.C., W.X.L., H.C. Visualization: H.M.Z., C.C.W., H.R.C., T.H.L., W.X.L., H.S., Y.Q.D., Q.Y.W., K.Y., W.K., Alvin H.K.C. Writing—original draft: H.M.Z., C.C.W., J.Y. Writing—review and editing: H.M.Z., C.C.W., J.Y. Funding acquisition: J.Y. Project administration: J.Y. Supervision: J.Y.

## Competing interests

The authors declare no competing interests.
