## [Transparent Peer Review file · Nature Communications]

RNA N6-methyladenosine eraser ALKBH5 drives stemness and chemoresistance of colorectal cancer and is therapeutic target

Corresponding Author: Professor Jun Yu

Version 0:

Reviewer comments:

Reviewer #1

(Remarks to the Author)

The manuscript by Zhou et al. shows an oncogenic role of ALKBH5 in colorectal cancer. The authors show that ALKBH5 decreases the expression of FAM84A mRNA, a change which affects β -catenin stability. This β -catenin stabilization is crucial, as it promotes cell proliferation and stemness, giving cancer cells a growth advantage and aiding in the development of chemoresistance. Zhou et al. explore ALKBH5's role in these processes using various in vitro models, including colorectal cancer patient-derived organoids, as well as in vivo mouse models to validate the findings in a physiological context. These findings underscore ALKBH5 as a promising therapeutic target. To investigate this potential, the researchers conducted experiments that inhibited ALKBH5 activity using vesicle nanoparticle (VNP)-delivered siRNA and also through genetic knockout approaches. Both methods successfully reduced stem cell-like properties in colorectal cancer cells. Additionally, inhibiting or knocking out ALKBH5 improved responses to chemotherapy, enhancing the cytotoxic effects of standard treatments on these cells.

The role of ALKBH5 in cancer biology has been extensively reported, with particular interest in its function as an m6A RNA demethylase involved in gene expression regulation. However, in colorectal cancer, findings have been contradictory, highlighting that there is a need for more robust studies to clarify its biological role and potential as a therapeutic target. This work clarifies the role of ALKBH5 in colon cancer and the mechanism shown is novel. The authors have performed a huge amount of work to demonstrate that ALKBH5 is indeed involved in tumour initiation, progression and chemoresistance. However, the role of ALKBH5 in human cancer stemness is not so clear: is it the cause or the consequence of affecting Wnt-related stemness? specially in the context of aberrantly activated WNT pathway human tumours what would be the effect? What it is convincing from this work is that ALKBH5 should be taken into consideration for therapeutic options. The methodology and experiments meet the expected standards.

Major points:

- In the manuscript, the role of ALKBH5 is well described in cancer, however ALKBH5 role in stemness is less convincing due to different events as Wnt pathway regulation, cell plasticity and environmental conditions. I'd suggest to explain well the limitations and the assumptions that need to be done.
- PDOs chosen: how similar are in mutation profile? how is Wnt pathway? primary tumour stage? Can this affect the role of ALKBH5 in stemness?
- Mouse models show the role in mouse tumour initiation, what about if ALKBH5 is activated in LGR5-neg cells?
- Mouse models as APC or AKP that have Wnt pathway affected would be a good model to test compounds that target ALKBH5.
- Would therapeutic options that target ALKBH5 be more or less effective depending on WNT pathway status of the primary tumour cancer cells?

Minor points:

- Introduction should included more details about what is known of ALKBH5 role in cancer.
- More details about bioinformatic analyses need to be done. e.g. figure 7a.
- Western blots results should be quantified (N=3).
- Figure 1a, correlation LGR5 and ALKBH5 has a pvalue higher than 0,01.

- Abstract should clarify that is bulk-RNAseq.

Reviewer #2

(Remarks to the Author)

In this study, the authors reconfirmed previous reports that ALKBH5 is highly expressed in colorectal cancer. Using mouse genetics, the authors demonstrated, in several directions, that high expression of ALKBH5 in Lgr5+ ISC increases the stemness of ISCs and promotes carcinogenesis, and that knocking out ALKBH5 in Lgr5+ ISCs reduces stem cell maintenance in ISCs and suppresses carcinogenesis. The authors also discussed that ALKBH5 regulates the Wnt/beta-catenin pathway through the expression of FAM84A and is involved in resistance in cancer therapy. This paper is divided into four main parts: 1. Expression of ALKBH5 in cancer and its contribution to carcinogenesis, 2. ALKBH5 controls the stability of FAM84A mRNA through m6A modification, 3. FAM84A controls the stability of β -catenin protein through ubiquitination by an unknown mechanism, and 4. Attempts to apply ALKBH5 to cancer therapy. Each part is composed of enough data to write an individual paper and could be published as four separate papers. Therefore, while a big story that creates a new concept was presented, the paper is very crowded because a large amount of data equivalent to several papers is packed into one paper. The quality and quantity of the data are fine, but the organization and presentation is not so good, which seems to diminish its merits. There were also some inconsistencies in the paper. If these points were corrected, this would be a good paper.

Overall,

I understand that the authors want to show everything they have considered/examined and to improve the quality of the paper by presenting a lot of data. However, there are too many non-essential figures, making it very busy and difficult to understand. I think it would be better to simply present only the essential data in the main figures and put supporting data in supplements. For example, many panels show data using two types of organoids under two conditions: overexpression and loss of gene expression. I totally agree that this is the most scientifically correct way to do it. However, as a result, the amount of data is too large and the panels in each figure are too small, making it difficult to understand. Please create figures that are easy for readers to understand, not figures that the authors want to show.

The details of the data are often difficult to understand because the figure legend and text are insufficient. The authors tend to include simple results in the figure legend, but please include the results in the text. Please include all the information necessary to interpret the figure in the figure legend. There is too little information, and readers will be tired by the puzzle for deciphering it. Please make sure to draw and write the figures and legends in a way that will lead the reader to the goal smoothly and without any hesitation.

There are many points that need to be improved, so details are provided below.

Major points;

1. It was well known that ALKBH5 is highly expressed in CRCs. Recently, it has been reported that ALKBH5 regulates the Wnt/beta-catenin pathway through mRNA m6A modification-WIF, DKK, and AXIN (doi: 10.1186/s12943-019-1128-6, doi: 10.1186/s12943-019-1128-6, doi: 10.1093/nar/gkaf707, doi: 10.1053/j.gastro.2023.04.032). Other roles of m6A in Wnt signaling are reviewed before (doi: 10.1016/j.biopha.2022.114023). There have also been reports of cancer therapy targeting Wnt signaling using vesicle-like nanoparticle-encapsulated ALKBH5-small interfering RNA (doi: 10.1053/j.gastro.2023.04.032). Please cite similar previous studies correctly in the manuscript. This manuscript has too few references to provide sufficient background.
2. Please accurately redefine the term transgenic mouse as used in this manuscript. The definitions of transgenic, knock-in, and knock-out are vague, and even within this manuscript the definitions are not consistent depending on the situation. Please rewrite them correctly based on the commonly used definitions.
3. Figure 1 is very busy. The first half of this figure uses public database analysis to investigate high expression of ALKBH5 in cancer, but the second half uses an ALKBH5 knock-in mouse model to analyze stemness and carcinogenesis. These two should not be coexisted in the same figure, but should be separated. Also, in Fig S1, the expression of ALKBH5 in cancer and non-cancerous areas is actually examined using actual patient samples. I feel this data is more fundamentally important than Figure 1, which uses database analysis. Alternatively, the data showing high expression of ALKBH5 in CRC can be moved to the supplementary figures, since it has been reported many previously.
4. Regarding Figure 1B, the text states that ALKBH5 expression is higher in LGR5(+) than in LGR5(-), but at first glance it looks the other way around. It might be better to find another way to present it.
5. Regarding Figure 1D, the text states that the correlation between ALKBH5 expression and LGR5 and CD133 expression was examined in two independent CRC cohorts. In fact, the N numbers are different for LGR5 and CD133 expression, but both are listed as cohort 2 in the figure. Indeed, same image of ALKBH5 was shown in both panels of LGR5 and CD133, suggesting they are same cohort. However, even if you read the figure legend, the details are not described, so it is difficult to understand what is actually going on this panel. Why the N number is different in same cohort?
6. In Figure 1F, ALKBH5 expression was observed on tissue sections, but how was it detected? By anti-ALKBH5, anti-FLAG, or EGFP signal? In particular, in Figure S2A, the notation in the figure is LGR5-EGFP, so it seems that EGFP signal is being observed. If so, during knock-in, not only ALKBH5-IRES-EGFP on the knock-in side but also EGFP of EGFP-IRES-CreER on the driver side is present, so we are not only looking at the expression of ALKBH5 knock-in. In this case, it would have to be immunostaining with Anti-FLAG for detecting exogenous ALKBH5.
7. In all figures for IH, IF or WB, please indicate not only the name proteins detected but also the antibodies used in the figures and/or legends.
8. In Figure 1U, the authors observed the expansion of CRC organoids. Usually, normal intestinal organoids develop and expand in WENR conditions, so WENR was used also here in the experiment using CRC organoids. If these conditions could be shown in the figure, it would be easier for readers to understand without going to the Supplementary Methods to

find them. Also, in Fig. 1U, WT organoids did not expand or grow after 4 days of culture despite under WENR, but in Fig. 2J, they expanded significantly as expected under the same conditions. These differences are convenient for showing author's claims, but the lack of consistency between these two experiments is problematic.

9. As seen in Fig 1F-J, the Lgr5-CreER:CAG-ALKBH5 knock-in mouse appears to tend to proliferate intestinal tract, but in Fig 1L, intestinal homeostasis in non-tumorous areas does not seem to be disrupted. If this mouse is not subjected to AOM/DSS-induced carcinogenesis experiments, will there be no problems with long-term intestinal homeostasis? Also, what survival curves do mice show?

10. The knock-in mouse data in Figure 1 and the knock-out mouse data in Figure 2 look excellent when viewed individually, but there are some inconsistencies when looking together and comparing the two. First, regarding the induction of carcinogenesis with AOM/DSS, as seen from Fig. 1K and Fig. 2B, the experiments were conducted under exactly the same conditions in the two. However, when comparing Fig. 1M1 and Fig. 2D1, the number of tumors in the WT control is completely different between the two. The number of tumors in the cKI mice is almost the same as that in the WT control mice in cKO experiment, and the number of tumors in the cKO mice is very close to that in the WT control mice of cKI experiment. If the same results are not obtained in the control mice even when experiments are performed under the same conditions, the reliability of the results is also diminished. This is essentially similar to the CRC organoid experiment mentioned above, and I am afraid that the lack of reproducibility within the same research is a large problem.

11. In Fig 2E, ALKBH5 expression still remains in the cKO mice. Why is this? Authors deleted two exons carrying transcriptional start and translational start respectively, which should completely eliminate ALKBH5 expression. Is it just a leaking by insufficient CreER ability or contamination of healthy tissues?

12. The authors show a significant reduction in stemness and cell proliferation in ALKBH5 cKO mice. However, it is a big surprise that intestinal homeostasis seems to be maintained for 80 days as in cKI mice, as shown in Fig. 2C-D1. If this mouse is not subjected to AOM/DSS-induced carcinogenesis experiments, will there be no problems with long-term intestinal homeostasis? What survival curves do mice show?? (same question with #9)

13. The authors describe for Fig 4A that m6A levels were decreased by overexpression of ALKBH5 and increased by knockdown. However, in my eyes, this does not appear to be the case in the figure. Total m6A level does not seem to be affected by the increase or decrease of ALKBH5. If there is still a clear difference, the density of dots should be quantified and shown.

14. Fig 4H shows that the m6A modification on FAM84A mRNA changes with the increase or decrease of ALKBH5, but this change appears to be almost the same as the change in the amount of FAM84A mRNA shown in Fig 4J-K. Is it possible that this is due to a change in the amount of FAM84A mRNA rather than a change in the level of m6A modification on FAM84A mRNA? It seems necessary to normalize the ratio of m6A modification with the amount of FAM84A mRNA.

15. Although the experiment using Cas9 is sophisticated, I think it is sufficient to show only the model diagram in Fig 4L and the amount of FAM84A mRNA in 4M. The remaining supportive data are sufficient to show in supplemental figures. In this way, please present only essential data throughout the manuscript, and carefully create simple and easy-to-understand figures.

16. In general, Fig 5, I feel that showing only the POP66 data is persuasive enough to simplify the figures. It may be sufficient to show CSC28 in the supplemental figure.

17. For the IP experiment in Fig 6J, please also show an input image of beta-catenin.

18. What does the FITC in the FACS data in Fig 7 represent? There is no explanation in the text or in the figure legend. Also, please adjust the order of the bar graphs samely with the FACS data so as not to give readers a stress. What color are the third and fourth bars of 5FU in D, and the third bars in F and G?

19. In the graph in Fig 7J, OXA appears to reduce proliferation similarly in both control and ALKBH5 KDs. If there is a synergistic effect that accelerates cell death, please also show the surface areas of ALKBH5 KD normalized by OXA(-) control of each.

20. The discussion section of the paper is written like a long summary and does not seem to adequately discuss the results of the study, further questions in this field of research, or problems in applying them to clinical treatment.

Minor points;

1. The transgenic mice model in Methods should be indicated mice models, as mentioned above.
2. The units in the graphs in Figure 1Q and R are incomplete. Ki67+ proportion/what?, Apoptosis/what?
3. Are the endoscopic photographs in Fig 1L and Fig 2C essentially necessary? (The photographs are too small to understand).
4. In Figure 2A, one loxP should remain even after deletion by Cre recombinase.
5. The bar graph in Fig 2F does not show two fractions, high- and low-grade.
6. The table in Fig S3I does not show among which samples have significant differences. The number of tumors can be seen in Fig 3D, so it can be simplified by deleting Fig S3I and denoting the significant differences between each sample in Fig 3D.
7. There is no indication of what MB mean in Fig 4A.
8. The colors of the line graphs in Fig S4A are all the similar, making it difficult to distinguish between groups.
9. Please provide details of the Anti-m6A antibody used for IP.
10. How authors measured the translation efficiency in Fig S4B?
11. The bar graph in Fig. 5K is not colored as stated at the bottom.
12. The graph in Fig 6B makes it difficult to distinguish the lines for each group. In particular, the lines for the two groups (FAM84A KO vs FAM84A;ALKBH5 DKO) that we want to compare in CSC28 are similar in color and overlap (although the data is excellent), and I was finally able to distinguish them by magnifying the PDF. Please find a better way.
13. Please include molecular weight markers in the silver staining and WB photos.
14. Flag-tag should be FLAG-tag.
15. Please include the time/day in the experimental diagram in Fig 8K.
16. In the Method, 50% v/v Wnt3a should be 50% v/v Wnt3a CM.

Reviewer #3

(Remarks to the Author)

In this manuscript, the authors highlighted the critical role of ALKBH5 in regulating self-renewal and drug resistance in colorectal cancer (CRC) stem cells. The authors employed multiple mouse models to thoroughly investigate how ALKBH5 promotes CRC progression through its demethylase function. By integrative analysis of RNA-seq, MeRIP-seq and Ribo-seq, the authors identified FAM48A as a critical downstream target of ALKBH5, which inhibits CSC. Mechanistically, FAM48A promotes β -catenin ubiquitination and degradation by directly interacting with β -catenin. Furthermore, they demonstrated that inhibition of ALKBH5 synergized with chemotherapy to trigger tumor regression in mouse models. Overall, it is a very interesting study, supported by substantial in vitro and in vivo data. In addition, the authors have identified novel mechanisms underlying the oncogenic role of ALKBH5 in CSC. However, several concerns need to be addressed.

Major comments:

1. It would be interesting to know if the authors also utilize the TCGA-CRC dataset to validate their findings by examining the correlation between m6A regulatory factors and ALKBH5.
2. What is the correlation between m6A regulatory factors and specific CRC stem cell marker genes?
3. In Figure 1Q, 1R, the authors showed clear IF staining results. Can flow cytometry be used to quantify the proliferation, apoptosis and differentiation of CRC stem cell populations in control and ALKBH5 knock in mice? Flow cytometry can provide valuable insights into the dynamics of CRC stem cell proliferation and apoptosis in mouse models.
4. In your Alkbh5 cKI mouse model, EGFP is under control of LGR5 promoter, indicating the colon stem cell while ALKBH5 is also co-expressed with EGFP. In this case, we can distinguish the cells expressing only EGFP cells from the cells expressing both ALKBH5 and EGFP in Figure 1F, 1N and 1S.
5. To illustrate the correlation between ALKBH5 and LGR5 or CD133, it would be beneficial to show their co-expression within the same cell by IF.
6. In Figure 2, the frequency of LGR5+ and CD133+ cells in Alkbh5 KO and control mice can be examined by flow cytometry. The deletion efficiency of Alkbh5 in CRC cells needs to be validated by qPCR.
7. While the authors determine CRC stem cells by markers including CD133 and LGR5, some experiments can be confirmed with additional stem cell markers including CD44 and EpCAM.
8. In Figure 3, ALKBH5 overexpression experiment can not address whether the function of ALKBH5 in promoting CSC proliferation and self-renewal is dependent on its enzymatic activity. Does overexpression of ALKBH5 or ALKBH5 mutant in ALKBH5 KD cells rescue these phenotypes?
9. In Figure 3C, you have presented the positive regulation of CD133 and CD44 by ALKBH5. What about LGR5? Also, please show the expression of CD44 in Figure 3F and 3G.
10. The knockdown efficiency of endogenous ALKBH5 in POP66 cells is not significant in Figure 4L. It may not be able to rule out the possibility that m6A is also downregulated by endogenous ALKBH5.
11. Please clarify the difference between A5-dCas13b and dCas13b-A5, as well as the different expression level of these two genes, as shown in Figure 4L.
12. In Figure 6G, which specific domain of FAM84A binds to β -catenin? Does mutating these binding sites disrupt the FAM84A's ability to inhibit β -catenin function and suppress CRC stem cell functions?
13. Have you considered the possibility that FAM84A downregulates β -Catenin through the lysosomal pathway?

Minor comments:

1. For the introduction, please expand on the background of ALKBH5's role in cancer, with a particular focus on its function in cancer stem cells.
2. Please add the x-axis label in Figure 1R.
3. Labeling the ladder and protein sizes in your Western blot results would enhance clarity.
4. In Figure 5K, it's difficult to differentiate between the sgNC+siIFG2BP1 and A5-KO+siIGF2BP1 groups.

We would like to know if the authors performed RNA-seq and Ribo-seq at the same time or separately. It is important to perform RNA-seq and Ribo-seq at the same time as the RNA-seq data was used as an input. It is fine if the authors repeats RNA-seq in separate times. However, we did not find all the data (input). Since we do not have expertise in bioinformatics, we can not evaluate the quality of Raw data from these studies.

Reviewer #4

(Remarks to the Author)

Reviewer #5

(Remarks to the Author)

The authors have established the causal relationships between ALKBH5, m6A, FAM84A, β -catenin, tumor stemness, and chemotherapy resistance in this study. By integrating mouse and organoid models with a human cohort, they have described the clinical relevance of ALKBH5 in CRC chemoresistance.

Based on the motif analysis, the quality of the MeRIP-seq data appears to be suboptimal (refer to Q15), and the basic quality control data for both MeRIP-seq and Ribo-seq are incomplete. I suggest that the authors incorporate further analyses to confirm the quality of the MeRIP-seq and Ribo-seq data (refer to Q16-17). Additionally, it is essential to include data on the relationship between m6A modifications changes and RNA expression/translation efficiency in CSCs in response to ALKBH5 perturbations (refer to Q18).

Overall, this is a well-organized study for Nat Communication. However, there were still many questions that need to be fully explained.

1. Prior studies have indicated that the m6A writer METTL3 promotes CRC stemness (<https://doi.org/10.1186/s12943-019-1038-7>). The role of ALKBH5, which demethylates RNA m6A, in enhancing the stemness of CSCs is intriguing. Did the m6A modification of FAM84A change in response to METTL3 perturbation? Did the authors assess the expression changes of METTL3 and other m6A regulators in response to ALKBH5 knockout or knockdown to prevent compensatory effects?
2. A summary and discussion of the function of other m6A regulators in colonic stem cells and CRC should be included in an appropriate section.
3. There is ongoing debate regarding how ALKBH5 in CRC tumor cells reshapes the surrounding tumor microenvironment. Some researchers have found hyperactivation of WNT signaling in ALKBH5-high patients, leading to the establishment of an immunosuppressive TME (10.1053/j.gastro.2023.04.032). However, other studies show a positive correlation between ALKBH5 and CD8+ T cell infiltration (<https://doi.org/10.1016/j.tranon.2023.101683>). Does the study provide any insights into these debates? Does the data support any mechanisms proposed by these studies?
4. The authors used bulk analysis to establish the correlation between ALKBH5 and stem cell marker expression. Is the tumor cell the primary source of ALKBH5 in CRC TME? Do other major subsets in TME, such as cancer-associated fibroblasts and myeloid subsets, contribute to this correlation?
5. The quality of figure legends needs improvement, with some details missing; for example, the legend of figure 3A.
6. Figure 1B lacks ticks on the y-axis.
7. The quality of the CD133 staining is unconvincing, as it does not overlap with DAPI staining and does not align with cellular morphology (Fig1T and 2I).
8. In Figure 2E, two ALKBH5 knockout samples show comparable expression levels to WT samples but lose LGR5 and CD133 signaling, please explain it.
9. In Figure 2F, the color of the statistical bar plot is difficult to read.
10. In Figure 2G, the title of the y-axis overlaps with the context of y-axis.
11. In Figure 2E, after ALKBH5 knockout, no CD133 can be detected in tumor tissue, while in Figure 2K, the CD133 signal only shows a slight decrease in cKO group, please explain this inconsistency.
12. Figure 2K lacks x-axis ticks and text.
13. Figure 2H-I lacks scale bars in the zoomed-in panels.
14. Did the WNT signaling are significant in GSEA analysis of RNA-seq? The p-values for each signaling pathway need to be added (Figure 4C/E).
15. The motif did not enrich in the GGACU motif; the GGACA also shows a weak p-value, Are there more significant motif in your peaks? And what these peaks talking about?
16. It is necessary to include the percentage of m6A-loss and m6A-gain peaks in response to perturbation.
17. To establish the reliability of Ribo-seq data, the CDS distribution, enrichment of translation initiation sites, and trinucleotide periodicity analysis must be provided.
18. The alterations in RNA expression and translation efficiency of the identified m6A-gain and m6A-loss genes in response to perturbation should be examined, with the remaining genes used as controls.
19. The color bar of Figure 4F is blank.
20. The m6A signaling in Figure 4G needs to be adjusted using RNA-seq input.
21. ALKBH5 overexpression does not show significant m6A level changes in FAM84A in Fig4G.
22. The significance between each group in Figure 5J needs to be added.
23. The color in Figure 5K is missing.
24. Does ALKBH5 expression correlate with survival in your cohort?
25. Explain the strategy used to define FAM84 high and low in your cohort.
26. There are several inconsistent colors in plots and color keys; please check.

Version 1:

Reviewer comments:

Reviewer #1

(Remarks to the Author)

The authors have improved the manuscript. While some sections are more novel and compelling than others, the overall volume of data presented is substantial. So given the depth of the results, in my opinion the manuscript is now suitable for publication in Nature Communications.

Reviewer #2

(Remarks to the Author)

The authors have addressed most of the issues Reviewer #2 pointed out and made efforts to improve the manuscript. As a

result, this revised manuscript has been significantly improved and is now at a scientifically acceptable level for publication. However, a few issues remain.

1. As I have pointed out before, figures are packed due to too much number of panels. In particular, two organoid strains, POP66 and CSC28, are always shown in Fig. 3-4 and 6-8. In most panels, these two organoids show almost the same behavior, so I think it would be fine to follow the previous suggestion as was done in Fig. 5, and show only one of them, and send another to supplemental figures. Since supplemental is already saturated, it may be unavoidable, but having two almost identical figures in same time to show one event causes congestion. This comment is a suggestion to make the manuscript better, not a mandatory instruction, so depends on author's decision.

2. In the answer from authors on major point #11, they discuss that ALKBH5 was knocked out only in the intestinal stem cells with Lgr5-Cre, and it was not knocked out in whole intestine with Cdx2-Cre or Villin-Cre. However, if the samples were taken 80 days after tamoxifen administration as in Fig. 2B, almost everything of the intestinal cells should have been replaced by knockout ISC-derived cells, since the lifespan of intestinal epithelial cells is only a few days, and even Paneth cells, which have a long lifespan, are only 1-2 months. Please indicate how many days after tamoxifen administration the samples were taken. Furthermore, Fig. 1B shows that ALKBH5 is expressed only in the basal stem cell region, but not in the upper part of the crypt, so the authors' answer that small detection of ALKBH5 protein was owing to that they did not knockout ALKBH5 in whole intestine, seems to be inconsistent. Because once ALKBH5 is knocked out in ISCs, no other cells express ALKBH5. Although the current figure has been improved and no additional experiments are likely to be necessary, a logical explanation is still needed as to where the ALKBH5 signal in the previous figure came from.

3. Please increase the signal intensity of ALKBH5 in green in Fig. 1B and S3F and yellow in WT in Fig. S3A-B so that it is easier to see.

4. In figures having graphs, the x-axis is always divided into two parts regardless of the number of experimental groups. When making graphs using general software, the number of divisions on the X-axis is usually optimized depending on the number of experimental groups.

5. Molecular weight markers were included in most of the immunoblotting images, but they were not included in the anti-ubiquitin blots particularly interested in (Fig.6K, 6N). Please show them.

Reviewer #3

(Remarks to the Author)

The authors have provided satisfactory and comprehensive responses to the reviewer's comments.

Reviewer #4

(Remarks to the Author)

Reviewer #5

(Remarks to the Author)

The authors have addressed most of my previous concerns, and the manuscript now shows significant improvement in quality. I only have a few minor suggestions specifically regarding Questions 4, 5, and 14 for further refinement:

1 Q4:

While the p-values are statistically significant ($p < 0.001$), the near-zero Pearson coefficients (e.g., $PCC \approx 0.1$) suggest no meaningful biological association between ALKBH5 and stemness markers in epithelial cells.

2 Q5:

In Figure 3A, the x-axis labels remain misaligned

3 Q14 :

The p-values (e.g., WNT pathway $p = 0.002$) barely meet significance threshold in GSEA analysis, likely reflecting false positives due to testing numerous pathways. Please provide adjusted p-values (FDR/q-values) to confirm the robustness after multiple-test correction.

Version 2:

Reviewer comments:

Reviewer #2

(Remarks to the Author)

In this revised version, the authors have fully addressed all of the issues I pointed out. As a result, the current manuscript involving many essential and solid data is scientifically and formally worthy of publication in Nature Communications. I respectfully congratulate the authors for their efforts and patience.

Reviewer #5

(Remarks to the Author)

The authors have made improvements to the manuscript, and I have no further questions

Response to the comments of referees in relation to the manuscript:

“RNA N⁶-methyladenosine eraser ALKBH5 drives stemness and chemoresistance of colorectal cancer and is a therapeutic target” (NCOMMS-24-52715)

Comments are written in *italics* and responses in plain texts. Modifications made to the manuscript are denoted in **BLUE**.

Response to Reviewer 1:

The manuscript by Zhou et al. shows an oncogenic role of ALKBH5 in colorectal cancer. The authors show that ALKBH5 decreases the expression of FAM84A mRNA, a change which affects β -catenin stability. This β -catenin stabilization is crucial, as it promotes cell proliferation and stemness, giving cancer cells a growth advantage and aiding in the development of chemoresistance. Zhou et al. explore ALKBH5's role in these processes using various in vitro models, including colorectal cancer patient-derived organoids, as well as in vivo mouse models to validate the findings in a physiological context. These findings underscore ALKBH5 as a promising therapeutic target. To investigate this potential, the researchers conducted experiments that inhibited ALKBH5 activity using vesicle nanoparticle (VNP)-delivered siRNA and also through genetic knockout approaches. Both methods successfully reduced stem cell-like properties in colorectal cancer cells. Additionally, inhibiting or knocking out ALKBH5 improved responses to chemotherapy, enhancing the cytotoxic effects of standard treatments on these cells.

The role of ALKBH5 in cancer biology has been extensively reported, with particular interest in its function as an m6A RNA demethylase involved in gene expression regulation. However, in colorectal cancer, findings have been contradictory, highlighting that there is a need for more robust studies to clarify its biological role and potential as a therapeutic target. This work clarifies the role of ALKBH5 in colon cancer and the mechanism shown is novel. The authors have performed a huge amount of work to demonstrate that ALKBH5 is indeed involved in tumour initiation, progression and chemoresistance. However, the role of ALKBH5 in human cancer stemness is not so clear: is it the cause or the consequence of affecting Wnt-related stemness? specially in the context of aberrantly activated WNT pathway human tumours what would be the effect? What is convincing from this work is that ALKBH5 should be taken into consideration for therapeutic options. The methodology and experiments meet the expected standards.

Major points:

1. In the manuscript, the role of ALKBH5 is well described in cancer, however ALKBH5 role in stemness is less convincing due to different events as Wnt pathway regulation, cell plasticity and environmental conditions. I'd suggest to explain well the limitations and the assumptions that need to be done.

Response: We have now added the limitations and assumptions to **Discussion** section, as follows:

Discussion (p. 25, line 529-542):

Although our work in colorectal CSCs, PDOs, and colon stem cell-specific ALKBH5 knockin and knockout mice inferred the role of ALKBH5 in stemness, it has some limitations. The colorectal CSCs and PDOs used in this study harbored activated WNT/ β -catenin pathway, a key target of ALKBH5. As a result, these *in vitro* models are sensitive to ALKBH5 manipulation. Nevertheless, our findings remain representative of human CRC, as 92-97% of CRC tumors exhibit aberrant WNT/ β -catenin activation according to the TCGA dataset [1]. Moreover, our assessment of cancer stemness relied on staining of well-defined markers LGR5 and CD133. Given the highly dynamic nature of cellular plasticity, lineage-tracing models and single cell RNA-sequencing would provide a more comprehensive understanding of the stemness-differentiation continuum in tumors [2]. Finally, we did not profile the tumor microenvironment (TME), such as stromal and immune cell populations [3] that might influence CSC behavior. Future studies will be conducted to determine whether CSC-specific ALKBH5 affects stemness through interactions with TME components.

2. PDOs chosen: how similar are in mutation profile? How is Wnt pathway? Primary tumor stage? Can this affect the role of ALKBH5 in stemness?

Response: To probe the mutational profile of PDOs, we have now sequenced PDO816 and PDO828 using AmpliSeq for Illumina Cancer HotSpot Panel (HOT STOP EXONS) (Table S5). PDO816 is mutated in APC (nonsense) and NRAS (missense); PDO828 is mutated in APC (nonsense), BRAF (missense), PTEN (missense), and TP53 (missense) (Table S6). As both PDOs harbor mutant APC, WNT/ β -catenin pathway is likely to be activated. Consistent with the important role of ALKBH5 in activating WNT/ β -catenin pathway, ALKBH5 promotes stemness in both PDOs. We have now described the new information for PDOs in **Results** section, as follows:

Results (p. 11, line 208-210):

Moreover, in PDO 816 and 828, which harbored APC mutation and activated WNT/ β -catenin pathway (Table S5 and S6), ALKBH5 knockdown significantly impaired the growth, self-renewal and expression of LGR5, CD133 and CD44 (Figure 3F).

3. Mouse models show the role in mouse tumor initiation, what about if ALKBH5 is activated in LGR5-neg cells?

Response: We achieved ALKBH5 knockin in LGR5⁺ cells in mice by crossing with Lgr5-Cre^{ERT2} mice. Other available Cre mouse strains used by us and others, including Cdx2-Cre^{ERT2} and Villin-Cre^{ERT2} mice, could enable colon-specific ALKBH5 knockin. However, both strains also activate ALKBH5 expression in colon stem cells at crypt base [4]. Hence, to the best of our knowledge, specific activation of ALKBH5 in Lgr5⁻ negative cells in mouse colon is technically unfeasible. Moreover, immunofluorescence (IF) staining of WT mouse colon showed that ALKBH5 expression is localized to the crypt base (Figure S2A-S2B), implying that ALKBH5 overexpression in Lgr5⁺ cells better mimics the physiological situation.

4. Mouse models as APC or AKP that have wnt pathway affected would be a good model to test compounds that target ALKBH5.

Response: We have now employed CRC organoids from *Apc*^{Min/+}*Kras*^{G12D/+}*Villin-Cre* mice (AK organoids) [5] to establish a tumor xenograft model in nude mice, followed by treatment with nanoparticle siALKBH5 or siNC (**Figure S13B**). siALKBH5 reduced tumor growth ($P<0.001$) and weight ($P=0.002$), indicating that ALKBH5 is a functional therapeutic target in AK organoids with deregulated WNT pathway. These findings are now described in **Results**, as follows:

Results (p. 21, line 431-436)

To confirm whether the therapeutic efficacy of siALKBH5 depends on altered WNT/ β -catenin pathway, we injected CRC organoids from *Apc*^{Min/+}*Kras*^{G12D/+} transgenic mice [5] (referred to as AK organoids) into nude mice, followed by treatment with siNC and siA5 nanoparticles (**Figure S13B**). As expected, siA5 treatment significantly suppressed tumor growth and tumor weight of AK xenografts compared to siNC (**Figure S13C**).

5. Would therapeutic options that target ALKBH5 be more or less effective depending on WNT pathway status of the primary tumor cancer cells?

Response: As mentioned above, the primary CRC organoids obtained by us expressed mutant *APC* gene, leading to WNT pathway activation. Indeed, 92-97% of CRC tumors harbored aberrant WNT activation according to TCGA dataset [1]. To determine if the effectiveness of ALKBH5 is dependent on WNT status, we treated CRC organoids with WNT inhibitors, with or without siALKBH5 or siNC (**Figure S13D**). WNT inhibitors [6-8] abrogated the inhibitory effect of siALKBH5 on CRC organoids, suggesting that the effectiveness of targeting ALKBH5 depends on activation of WNT pathway. These data are now described in **Results**, as follows:

Results (p. 21, line 436-439)

Moreover, the addition of WNT inhibitors abrogated the inhibitory effect of siALKBH5 on the growth of primary CRC organoids (**Figure S13D**), indicating that effectiveness of targeting ALKBH5 requires activated WNT signaling.

Minor points:

1. Introduction should include more details about what is known of ALKBH5 role in cancer.

Response: We have now included background information on the role of ALKBH5 in cancer in the **Introduction** section, as follows:

Introduction (p. 4, line 74-79):

ALKBH5 has been reported to exert oncogenic [9-12] or tumor suppressive function [13-17] in a context-dependent manner. On the other hand, several studies have suggested that ALKBH5 selectively enhances self-renewal capacity of cancer stem cells (CSCs) in solid tumors and leukemia [18-21] and boosts stemness properties in several

tumor types [11, 22-25] through activation of WNT/ β -catenin signaling. Nevertheless, the role of ALKBH5 in CRC stem cells remains incompletely understood.

2. *More details about bioinformatic analyses need to be done. E.g. Figure 7a.*

Response: We have now included details on the bioinformatic analysis in the **Methods** section, as follows:

Methods (p. 40, line 852-857):

The samples from GSE72968 dataset were randomly split into training and validation cohorts at a ratio of 7:3. Based on ALKBH5 mRNA expression, a random forest model was trained to distinguish chemotherapy responders and non-responders on the training cohort and tested on the validation cohort using the caret R package. Receiver operating characteristic curves were generated by pROC R package for performance evaluation.

3. *Western blots results should be quantified (N=3).*

Response: We have now quantified the western blot in **Figure S1H** (N=12), **Figure 1K** (N=5), **Figure 2E** (N=5), **Figure S6C** (N=5), **Figure 6O** (N=5), **Figure S10B** (N=3), **Figure S10C** (N=3), **Figure S13E** (N=5), **Figure S13F** (N=3), **Figure S13G** (N=5) and **Figure 13H** (N=3).

4. *Figure 1a, correlation LGR5 and A5 has a p value higher than 0.01.*

Response: Correlation between LGR5 and ALKBH5 has a *P*-value of 0.03, which was considered statistically significant ($P < 0.05$).

5. *Abstract should clarify that is bulk-RNA seq.*

Response: Corrected.

Response to Reviewer 2:

In this study, the authors reconfirmed previous reports that ALKBH5 is highly expressed in colorectal cancer. Using mouse genetics, the authors demonstrated, in several directions, that high expression of ALKBH5 in Lgr5+ ISCs increases the stemness of ISCs and promotes carcinogenesis, and that knocking out ALKBH5 in Lgr5+ ISCs reduces stem cell maintenance in ISCs and suppresses carcinogenesis. The authors also discussed that ALKBH5 regulates the Wnt/beta-catenin pathway through the expression of FAM84A and is involved in resistance in cancer therapy. This paper is divided into four main parts: 1. Expression of ALKBH5 in cancer and its contribution to carcinogenesis, 2. ALKBH5 controls the stability of FAM84A mRNA through m6A modification, 3. FAM84A controls the stability of β -catenin protein through ubiquitination by an unknown mechanism, and 4. Attempts to apply ALKBH5 to cancer therapy. Each part is composed of enough data to write an individual paper and could be published as four separate papers. Therefore, while a big story that creates a new concept was presented, the paper is very crowded because a large amount of data equivalent to several papers is packed into one paper. The quality and quantity of the data are fine, but the organization and presentation are not so good, which seems to diminish its merits. There were also some inconsistencies in the paper. If these points were corrected, this would be a good paper.

Overall, I understand that the authors want to show everything they have considered/examined and to improve the quality of the paper by presenting a lot of data. However, there are too many non-essential figures, making it very busy and difficult to understand. I think it would be better to simply present only the essential data in the main figures and put supporting data in supplements. For example, many panels show data using two types of organoids under two conditions: overexpression and loss of gene expression. I totally agree that this is the most scientifically correct way to do it. However, as a result, the amount of data is too large and the panels in each figure are too small, making it difficult to understand. Please create figures that are easy for readers to understand, not figures that the authors want to show.

The details of the data are often difficult to understand because the figure legend and text are insufficient. The authors tend to include simple results in the figure legend, but please include the results in the text. Please include all the information necessary to interpret the figure in the figure legend. There is too little information, and readers will be tired by the puzzle for deciphering it. Please make sure to draw and write the figures and legends in a way that will lead the reader to the goal smoothly and without any hesitation. There are many points that need to be improved, so details are provided below.

Major points;

1. It was well known that ALKBH5 is highly expressed in CRCs. Recently, it has been reported that ALKBH5 regulates the Wnt/beta-catenin pathway through mRNA m6A modification-WIF, DKK, and AXIN (doi: 10.1186/s12943-019-1128-6, doi:

10.1186/s12943-019-1128-6, doi: 10.1093/nar/gkae707, doi: 10.1053/j.gastro.2023.04.032). Other roles of m⁶A in Wnt signaling are reviewed before (doi: 10.1016/j.biopha.2022.114023). There have also been reports of cancer therapy targeting Wnt signaling using vesicle-like nanoparticle-encapsulated ALKBH5-small interfering RNA (doi: 10.1053/j.gastro.2023.04.032). Please cite similar previous studies correctly in the manuscript. This manuscript has too few references to provide sufficient background.

Response: We have now included background information on the role of ALKBH5 and other m⁶A regulators in cancer in the **Introduction** section, as follows:

Introduction (p. 4, line 67-79):

Emerging studies suggest the potential roles of m⁶A regulators in modulating stemness and tumorigenesis in colon. M⁶A writers METTL3 and METTL14 have been reported to sustain self-renewal capacity in the normal colon [26, 27]. METTL3 and m⁶A readers including YTHDF1/2 and IGF2BP1/2 have been implicated in colorectal tumorigenesis, chemoresistance [28-33], and metastasis [33-36] through modulating the translation of m⁶A downstream targets.

ALKBH5 has been reported to exert oncogenic [9-12] or tumor suppressive function [13-17] in a context-dependent manner. On the other hand, several studies have suggested that ALKBH5 selectively enhances self-renewal capacity of cancer stem cells (CSCs) in solid tumors and leukemia [18-21] and also boosts stemness properties in several tumor types [11, 22-25] through activation of WNT/ β -catenin signaling. Nevertheless, the role of ALKBH5 in CRC stem cells remains incompletely understood.

2. Please accurately redefine the term transgenic mouse as used in this manuscript. The definitions of transgenic, knock-in, and knock-out are vague, and even within this manuscript the definitions are not consistent depending on the situation. Please rewrite them correctly based on the commonly used definitions.

Response: Instead of transgenic mouse, we have now defined the mouse strains utilized here as follows: Rosa26^{Isl-*Alkbh5*}LGR5-Cre^{ERT2} (colon stem cell-specific *Alkbh5* knockin mice; *Alkbh5*-cKI) and *Alkbh5*^{flox/flox}LGR5-Cre^{ERT2} (colon stem cell-specific *Alkbh5* knockout mice; *Alkbh5*-cKO).

3. Figure 1 is very busy. The first half of this figure uses public database analysis to investigate high expression of ALKBH5 in cancer, but the second half uses an ALKBH5 knock-in mouse model to analyze stemness and carcinogenesis. These two should not be coexisted in the same figure, but should be separated. Also, in Fig S1, the expression of ALKBH5 in cancer and non-cancerous areas is actually examined using actual patient samples. I feel this data is more fundamentally important than Figure 1, which uses database analysis. Alternatively, the data showing high expression of ALKBH5 in CRC can be moved to the supplementary figures, since it has been reported many previously.

Response: We have now moved data related to ALKBH5 expression in CRC to **Figure**

S1A, S1F-S1K.

4. Regarding Figure 1B, the text states that ALKBH5 expression is higher in LGR5(+) than in LGR5(-), but at first glance it looks the other way around. It might be better to find another way to present it.

Response: We have now updated Figure 1B and relocated this to **Figure S1F**.

5. Regarding Figure 1D, the text states that the correlation between ALKBH5 expression and LGR5 and CD133 expression was examined in two independent CRC cohorts. In fact, the N numbers are different for LGR5 and CD133 expression, but both are listed as cohort 2 in the figure. Indeed, same image of ALKBH5 was shown in both panels of LGR5 and CD133, suggesting they are same cohort. However, even if you read the figure legend, the details are not described, so it is difficult to understand what is actually going on this panel. Why the N number is different in same cohort?

Response: Although staining for LGR5 and CD133 is performed on consecutive slides from the same cohort, cases without tumor tissues or poor-quality tissues were excluded, which differ according to individual TMA slides. Hence, there is a minor difference in the number of cases scored for LGR5 and CD133 staining.

6. In Figure 1F, ALKBH5 expression was observed on tissue sections, but how was it detected? By anti-ALKBH5, anti-FLAG, or EGFP signal? In particular, in Figure S2A, the notation in the figure is LGR5-EGFP, so it seems that EGFP signal is being observed. If so, during knock-in, not only ALKBH5-IRES-EGFP on the knock-in side but also EGFP of EGFP-IRES-CreER on the driver side is present, so we are not only looking at the expression of ALKBH5 knock-in. In this case, it would have to be immunostaining with Anti-FLAG for detecting exogenous ALKBH5.

Response: In **Figure 1B** and **Figure 1J**, ALKBH5 expression was determined using an anti-ALKBH5 antibody, which confirmed the overexpression of ALKBH5 in cKI mice. However, we have initially utilized anti-EGFP to detect LGR5 expression, which was inappropriate as both ALKBH5 and LGR5 transgene carry EGFP. To address this, we have now determined LGR5 expression using a FISH probe for Lgr5 mRNA in revised **Figure 1O** and **Figure 2H**.

7. In all figures for IHC, IF or WB, please indicate not only the name proteins detected but also the antibodies used in the figures and/or legends.

Response: We have now provided detailed clarifications of the antibodies and methods used for each staining in the figure legends.

8. In Figure 1U, the authors observed the expansion of CRC organoids. Usually, normal intestinal organoids develop and expand in WENR conditions, so WENR was used also here in the experiment using CRC organoids. If these conditions could be shown in the figure, it would be easier for readers to understand without going to the Supplementary Methods to find them. Also, in Fig. 1U, WT organoids did not expand or grow after 4 days of culture despite under WENR, but in Fig. 2J, they expanded significantly as

expected under the same conditions. These differences are convenient for showing author's claims, but the lack of consistency between these two experiments is problematic.

Response: We have now described the culture conditions (e.g. WENR medium) in all Figure legends for organoid culture. To ensure consistency between experiments, we have parallelly generated CRC organoids from WT, ALKBH5-cKI, and ALKBH5-cKO mice, and obtained improved results, as shown in revised **Figure 1Q and 2J**.

9. As seen in Fig 1F-J, the Lgr5-CreER:CAG-ALKBH5 knock-in mouse appears to tend to proliferate intestinal tract, but in Fig 1L, intestinal homeostasis in non-tumorous areas does not seem to be disrupted. If this mouse is not subjected to AOM/DSS-induced carcinogenesis experiments, will there be no problems with long-term intestinal homeostasis? Also, what survival curves do mice show?

Response: We have established a new batch of wild-type (WT) mice and ALKBH5-cKI mice without AOM/DSS treatment. To assess long-term intestinal homeostasis, the mice were treated with tamoxifen, and the colon and small intestine were collected after four months. In WT mice, ALKBH5 is primarily expressed in the base of the colon and small intestine (**Figure S2A-S2B**), and the overexpression of ALKBH5 was confirmed in ALKBH5-cKI mice (**Figure S2A-S2B**). However, histological analysis revealed no abnormal histology in the small intestine and colon (**Figure S2C**), and no changes were observed with regards to body weight and colon length compared to WT mice (**Figure S2D**). All ALKBH5-cKI mice were in apparent good health and all mice survived the observation period. These data are now described in **Results**, as follows:

Results (p. 8, line 145-148):

Nevertheless, long-term (4 months) ALKBH5 knockin (**Figure S2A-S2B**) had no effect on intestinal homeostasis, as evidenced by histological analysis of small intestine and colon (**Figure S2C**), and measurement of colon length and body weight (**Figure S2D**).

10. The knock-in mouse data in Figure 1 and the knock-out mouse data in Figure 2 look excellent when viewed individually, but there are some inconsistencies when looking together and comparing the two. First, regarding the induction of carcinogenesis with AOM/DSS, as seen from Fig. 1K and Fig. 2B, the experiments were conducted under exactly the same conditions in the two. However, when comparing Fig. 1M1 and Fig. 2D1, the number of tumors in the WT control is completely different between the two. The number of tumors in the cKI mice is almost the same as that in the WT control mice in cKO experiment, and the number of tumors in the cKO mice is very close to that in the WT control mice of cKI experiment. If the same results are not obtained in the control mice even when experiments are performed under the same conditions, the reliability of the results is also diminished. This is essentially similar to the CRC organoid experiment mentioned above, and I am afraid that the lack of reproducibility within the same research is a large problem.

Response: In our previous results, the WT controls for cKI and cKO experiments were not conducted with the same batch of mice. Together with other experimental variations

(e.g. different batches of AOM, DSS, etc), this could explain the observed difference in tumorigenesis. To validate the reproducibility of our work, we established a new batch of AOM/DSS-induced CRC models, simultaneously evaluating tumorigenesis in WT, ALKBH5-cKI and ALKBH5-cKO mice. As shown in revised **Figure 1I** and **Figure 2D**, we reached the same conclusions as in our previous results, showing that ALKBH5-cKI promotes CRC, whereas ALKBH5-cKO suppresses CRC as compared to WT controls.

11. In Fig 2E, ALKBH5 expression still remains in the cKO mice. Why is this? Authors deleted two exons carrying transcriptional start and translational start respectively, which should completely eliminate ALKBH5 expression. Is it just a leaking by insufficient CreER ability or contamination of healthy tissues?

Response: Incomplete knockout could be expected, since in our mouse models we have deleted only ALKBH5 in Lgr5 +ve lineage (Lgr5-Cre) but not in total colonocytes (e.g., Cdx2-Cre or Villin-Cre). As tumors may arise from Lgr5 -ve cells at a lower frequency, a low level of ALKBH5 can still be detected. During animal harvesting, we have taken great care to isolate tumor tissues to prevent contamination by healthy tissues. We have now repeated the Western blot with improved results in revised **Figure 2E**.

12. The authors show a significant reduction in stemness and cell proliferation in ALKBH5 cKO mice. However, it is a big surprise that intestinal homeostasis seems to be maintained for 80 days as in cKI mice, as shown in Fig. 2C-D1. If this mouse is not subjected to AOM/DSS-induced carcinogenesis experiments, will there be no problems with long-term intestinal homeostasis? What survival curves do mice show?? (same question with #9)

Response: We have established a new batch of wild-type (WT) mice and ALKBH5-cKO mice without AOM/DSS treatment. To assess long-term intestinal homeostasis, the mice were treated with tamoxifen, and the colon and small intestine were collected after four months. We confirmed the knockout of ALKBH5 in intestinal crypts (**Figure S3A-S3B**). However, histological analysis revealed no abnormal histology in the small intestine and colon (**Figure S3C**), and no alterations were observed with regards to body weight and colon length compared to WT mice (**Figure S3D**). All ALKBH5-cKI mice were in apparent good health and all mice survived the observation period. These data are now described in **Results**, as follows:

Results (p. 9, line 176-179):

In the absence of AOM/DSS, long-term (4 months) ALKBH5 knockout (**Figure S3A-S3B**) had no effect on intestinal homeostasis, as evidenced by histological analysis of small intestine and colon (**Figure S3C**), and measurement of colon length and body weight (**Figure S3D**).

13. The authors describe for Fig 4A that m6A levels were decreased by overexpression of ALKBH5 and increased by knockdown. However, in my eyes, this does not appear to be the case in the figure. Total m6A level does not seem to be affected by the increase

or decrease of ALKBH5. If there is still a clear difference, the density of dots should be quantified and shown.

Response: We have now quantified dot blot results, showing that ALKBH5 knockdown significantly increased total m⁶A levels (**Figure S5A**), while ALKBH5 overexpression significantly down-regulated total m⁶A levels (**Figure S5B**).

14. Fig 4H shows that the m⁶A modification on FAM84A mRNA changes with the increase or decrease of ALKBH5, but this change appears to be almost the same as the change in the amount of FAM84A mRNA shown in Fig 4J-K. Is it possible that this is due to a change in the amount of FAM84A mRNA rather than a change in the level of m⁶A modification on FAM84A mRNA? It seems necessary to normalize the ratio of m⁶A modification with the amount of FAM84A mRNA.

Response: The data shown in **Figure 4H** is normalized to mRNA abundance in input sample. We have revised the y-axis label to “FAM84A m⁶A-to-mRNA ratio”, and also provided detailed methodology in **Methods** section, as follows:

Methods (p. 37, line 780-788):

M⁶A pulldown assay was conducted using the Magna RIP® RNA-Binding Protein Immunoprecipitation Kit (#17-700). Following the kit's protocol, 10% of the RNA was reserved for FAM84A from each sample (shNC, shA5-1, shA5-2; EV, ALKBH5^{WT}, ALKBH5^{H204A}) as input. Normalized m⁶A abundance was calculated as follows:

$$\Delta CT_{\text{IgG}} = CT_{\text{IgG}} - CT_{\text{input}}$$

$$\Delta CT_{\text{MeRIP}} = CT_{\text{MeRIP}} - CT_{\text{input}}$$

$$\Delta \Delta CT_{\text{Normalized MeRIP}} = \Delta CT_{\text{MeRIP}} - \Delta CT_{\text{IgG}}$$

$$\text{m}^6\text{A abundance for each sample} = 2^{(-\Delta \Delta CT_{\text{Normalized MeRIP}})}$$

15. Although the experiment using Cas9 is sophisticated, I think it is sufficient to show only the model diagram in Fig 4L and the amount of FAM84A mRNA in 4M. The remaining supportive data are sufficient to show in supplemental figures. In this way, please present only essential data throughout the manuscript, and carefully create simple and easy-to-understand figures.

Response: We have reorganized the data by retaining the model diagram in **Figure 4L** and FAM84A mRNA data in **Figure 4M**. Other data are moved to **Figure S5L-S5M**.

16. In general, figure 5, I feel that showing only the POP66 data is persuasive enough to simplify the figures. It may be sufficient to show CSC28 in the supplemental figure.

Response: We have now moved CSC28 data to **Figure S6A, S6B, S6I, S6J, S6K, S6O, S6P, and S6Q**.

17. For the IP experiment in figure 6J, please also show an input image of beta-catenin.

Response: We have now added input image of β -Catenin in **Figure 6J**.

18. What does the FITC in the FACS data in Fig 7 represent? There is no explanation in the text or in the figure legend. Also, please adjust the order of the bar graphs samely

with the FACS data so as not to give readers a stress. What color are the third and fourth bars of 5FU in D, and the third bars in F and G?

Response: FITC in the FACS data represents Annexin V, a marker for early apoptotic cells. We have also ordered the bar graph in the same order as FACS data and corrected the bar color in revised **Figure 7D, 7E, 7G and 7K**.

19. In the graph in Fig 7J, OXA appears to reduce proliferation similarly in both control and ALKBH5 KDs. If there is a synergistic effect that accelerates cell death, please also show the surface areas of ALKBH5 KD normalized by OXA(-) control of each.

Response: We have now calculated the normalized surface area and the results are now presented in **Figure S10A**.

20. The discussion section of the paper is written like a long summary and does not seem to adequately discuss the results of the study, further questions in this field of research, or problems in applying them to clinical treatment.

Response: We have now added more discussion of our findings, as well as limitations to **Discussion** section, as follows:

Discussion (p. 24, line 501-507):

ALKBH5 has been shown to regulate WNT signaling in CRC by regulating DKK1 [37, 38] and WIF-1 [39]. Here, we additionally identified FAM84A as a downstream effector for ALKBH5-driven WNT/ β -catenin signaling in colorectal CSCs. Besides FAM84A, ALKBH5 was reported to regulate alternative CSC targets, such as FOXM1 [19, 40] and NANOG [20], implying that multiple ALKBH5 downstream targets might collaterally contribute to stemness in CRC.

Discussion (p. 25, line 521-527):

While our results provide proof-of-principle results supporting VNP-siALKBH5 as an adjuvant to boost chemotherapy efficacy against colorectal CSCs, several barriers may hamper its clinical application, including the efficient and selective delivery of siRNAs to tumor sites and preventing potential toxic side effects. Further development of small molecules targeting ALKBH5 also represent an alternative approach to target colorectal CSCs.

Discussion (p. 25, line 529-542):

Although our work in colorectal CSCs, PDOs, and colon stem cell-specific ALKBH5 knockin and knockout mice inferred the role of ALKBH5 in stemness, it has some limitations. The colorectal CSCs and PDOs used in this study harbored activated WNT/ β -catenin pathway, a key target of ALKBH5. As a result, these *in vitro* models are sensitive to ALKBH5 manipulation. Nevertheless, our findings remain representative of human CRC, as 92-97% of CRC tumors exhibit aberrant WNT/ β -catenin activation according to the TCGA dataset [1]. Moreover, our assessment of cancer stemness relied on staining of well-defined markers LGR5 and CD133. Given the highly dynamic nature of cellular plasticity, lineage-tracing models and single cell

RNA-sequencing would provide a more comprehensive understanding of the stemness-differentiation continuum in tumors [2]. Finally, we did not profile the tumor microenvironment (TME), such as stromal and immune cell populations [3] that might influence CSC behavior. Future studies will be conducted to determine whether CSC-specific ALKBH5 affects stemness through interactions with TME components.

Minor points:

1. *The transgenic mice model in Methods should be indicated mice models, as mentioned above.*

Response: Revised in **Methods** section.

2. *The units in the graphs in figure 1Q and R are incomplete. Ki67+ proportion/what?, apoptosis/what?*

Response: These refer to Ki67⁺ proportion per view and apoptosis per view. We have now added these details in **Figures 1M** and **1N**.

3. *Are the endoscopic photographs in Fig 1L and Fig 2C essentially necessary? (The photographs are too small to understand).*

Response: Removed.

4. *In figure 2A, one loxP should remain even after deletion by Cre recombinase.*

Response: Corrected.

5. *The bar graph in Figure 2F does not show two fractions, high- and low- grade.*

Response: Corrected.

6. *The table in figure S3I does not show among which samples have significant differences. The number of tumors can be seen in figure 3D, so it can be simplified by deleting figure S3I and denoting the significant differences between each sample in figure 3D.*

Response: We have removed **Figure S3I-S3J** and denoted differences in **Figure 3D**.

7. *There is no indication of what MB mean in figure 4A.*

Response: In **Figure 4A**, MB refers to methylene blue (MB) staining, and is the loading control for total RNA. We have added this information to **Figure 4A**.

8. *The colors of the line graphs in Figure S4A are all the similar, making it difficult to distinguish between groups.*

Response: We have revised the line graphs in updated **Figure S5J**.

9. *Please provide details of the anti-m6A antibody used for IP.*

Response: We used the anti-m6A antibody from Abcam (ab208577), which is suitable for IP work. This information has been added to **Methods** section.

10. How authors measured the translation efficiency in Fig S4B?

Response: We measured translation efficiency by simultaneously extracting RNA and proteins from shNC, shA5-1, and shA5-2 cells. qPCR and Western blot were performed to determine mRNA and protein expression of FAM84A and GAPDH, respectively. The expression of FAM84A was normalized to GAPDH, and then we calculated the ratio of protein-to-mRNA expression as a measure of translation efficiency. We have now added the description to **Methods**, as follows:

Methods (p. 38, line 802-807):

In **Figure S6H**, translation efficiency was measured by simultaneously extracting RNA and proteins from shNC, shA5-1, and shA5-2 cells. qPCR and Western blot were then performed to determine the mRNA and protein expression of FAM84A and GAPDH, respectively. The expression of FAM84A was normalized to GAPDH, and then the ratio of protein-to-mRNA expression is taken as a measure of translation efficiency.

11. The bar graph in Fig 5K is not colored as stated at the bottom.

Response: Corrected.

12. The graph in Fig 6B makes it difficult to distinguish the lines for each group. In particular, the lines for the two groups (FAM84A KO vs FAM84A; ALKBH5 DKO) that we want to compare in CSC28 are similar in color and overlap (although the data is excellent), and I was finally able to distinguish them by magnifying the PDF. Please find a better way.

Response: We have changed the color scheme to better distinguish the groups.

13. Please include molecular weight markers in the silver staining and WB photos.

Response: Added.

14. Flag-tag should be FLAG-tag.

Response: Revised.

15. Please include the time/day in the experimental diagram in Fig 8K.

Response: Added.

16. In the Method, 50% v/v Wnt3a should be 50% v/v Wnt3a CM.

Response: Revised.

Response to Reviewer 3 and 4:

In this manuscript, the authors highlighted the critical role of ALKBH5 in regulating self-renewal and drug resistance in colorectal cancer (CRC) stem cells. The authors employed multiple mouse models to thoroughly investigate how ALKBH5 promotes CRC progression through its demethylase function. By integrative analysis of RNA-seq, MeRIP-seq and Ribo-seq, the authors identified FAM48A as a critical downstream target of ALKBH5, which inhibits CSC. Mechanistically, FAM48A promotes β -catenin ubiquitination and degradation by directly interacting with β -catenin. Furthermore, they demonstrated that inhibition of ALKBH5 synergized with chemotherapy to trigger tumor regression in mouse models. Overall, it is a very interesting study, supported by substantial in vitro and in vivo data. In addition, the authors have identified novel mechanisms underlying the oncogenic role of ALKBH5 in CSC. However, several concerns need to be addressed.

Major points:

1. It would be interesting to know if the authors also utilized the TCGA-CRC dataset to validate their findings by examining the correlation between m⁶A regulatory factors and ALKBH5.

Response: We have now analyzed correlation between m⁶A regulators and ALKBH5 in TCGA-CRC dataset at mRNA level (N=637). We found that ALKBH5 is negatively correlated with m⁶A readers IGF2BP2, YTHDF1, YTHDF3, YTHDC1, and YTHDC2, with the exception of IGF2BP3 (**Figure S1C**). No significant correlation was observed between ALKBH5 with m⁶A writers (METTL3, METTL14, WTAP) or m⁶A eraser FTO (**Figure S1C**). This suggests that ALKBH5 expression negatively correlates with m⁶A readers, but had no correlation with m⁶A writers and erasers.

Results (p. 6, line 102-104):

*In TCGA cohort, ALKBH5 mRNA negatively correlates with m⁶A readers, but had no correlation with m⁶A writers and erasers (**Figure S1C**).*

2. What is the correlation between m⁶A regulatory factors and specific CRC stem cell marker genes?

Response: We have now analyzed the correlation between m⁶A regulatory factors and CRC stem cell markers CD133 and LGR5 at mRNA level based on TCGA CRC cohort. Although significant associations were found, the majority of them have low correlation coefficient ($R < 0.2$), except the positive correlations between LGR5 and FTO, IGF2BP2, YTHDF3, and YTHDC1 (**Figure S1B**). Nevertheless, at protein level (CPTAC cohort) only ALKBH5 positively correlated with LGR5 (**Figure S1A**). This discrepancy could be due to the post-transcriptional regulation of ALKBH5 expression [41-44]. These data are now described in **Results** section, as follows:

Results (p. 6, line 98-102):

To ask if m⁶A modification plays a role in CRC stemness, we examined the correlation between m⁶A regulators with stemness markers in CPTAC cohort (protein) and TCGA-

CRC dataset (mRNA). At protein level, only ALKBH5 positively correlated with LGR5 (**Figure S1A**) [45], whereas at mRNA level m⁶A regulators showed either weak or no correlation with LGR5 and CD133 (**Figure S1B**).

3. In *Figure 1Q, 1R*, the authors showed clear IF staining results. Can flow cytometry be used to quantify the proliferation, apoptosis and differentiation of CRC stem cell populations in control and ALKBH5 knock in mice? Flow cytometry can provide valuable insights into the dynamics of CRC stem cell proliferation and apoptosis in mouse models.

Response: We established new primary CRC organoids (passage 2) from a new batch of WT mice and ALKBH5 knockin mice, and flow cytometry analysis was conducted. Consistent with IF staining, ALKBH5 knockin promoted cell proliferation (**Figure 1R**), suppressed apoptosis (**Figure 1S**), and induced stemness (CD133⁺, **Figure 1T**). These findings are now described in **Results** section, as follows:

Results (p. 9, line 167-170):

Primary CRC organoids (passage 2) from ALKBH5-cKI mice demonstrated increased proliferation (**Figure 1R**), suppressed apoptosis (**Figure 1S**) and elevated proportion of CD133⁺ cell population (**Figure 1T**) as compared to WT mice.

4. In your *Alkbh5 cKI* mouse model, EGFP is under control of LGR5 promoter, indicating the colon stem cell while ALKBH5 is also co-expressed with EGFP. In this case, we can distinguish the cells expressing only EGFP cells from the cells expressing both ALKBH5 and EGFP in *Figure 1F, 1N and 1S*.

Response: In **Figure 1B** and **1J** the green staining refers to ALKBH5 staining with an anti-ALKBH5 antibody. In **Figure 1O**, we performed FISH for LGR5 mRNA using an LGR5-specific probe. In our mouse model, both ALKBH5 and LGR5 transgenes carry EGFP, thus it is impossible to distinguish cells expressing only EGFP cells from those expressing both ALKBH5 and EGFP. We have now updated **Figure 1 legend** to explain these images.

5. To illustrate the correlation between ALKBH5 and LGR5 or CD133, it would be beneficial to show their co-expression within the same cell by IF.

Response: We have now performed additional co-expression analyses of ALKBH5 (IF) together with LGR5 (FISH) or CD133 (IF) in **Figure S1L** and **S1M**, respectively. These results confirmed co-localization of ALKBH5 with these stemness markers. These data are now described in **Results**, as follows:

Results (p. 7, line 128-130):

Moreover, co-staining of ALKBH5 and LGR5 (**Figure S1L**) and ALKBH5 and CD133 (**Figure S1M**) demonstrated their co-localized expression in murine CRC tumors.

6. In *figure 2*, the frequency of LGR5⁺ and CD133⁺ cells in *Alkbh5 KO* and control mice can be examined by flow cytometry. The deletion efficiency of *Alkbh5* in CRC cells

needs to be validated by qPCR.

Response: We have performed additional flow cytometry of primary tumor organoids (passage 2) from WT and ALKBH5-cKO mice. As shown in **Figure 2K** and **2L**, both CD133⁺ and LGR5-EGFP⁺ cells were depleted in ALKBH5-cKO mice. We have also validated deletion efficiency of ALKBH5 in tumor tissues by qPCR (**Figure S3E**). These findings were added to **Results** section, as follows:

Results (p. 10, line 187-190):

CRC organoids from cKO mice (**Figure S3F**) showed impaired self-renewal (**Figure 2J**). Flow cytometry showed that CD133 (**Figure 2K**) and LGR5^{EGFP} (**Figure 2L**) were downregulated in ALKBH5-cKO organoids compared to WT organoids.

Results (p. 9, line 182-184):

ALKBH5 knockout was confirmed (**Figure S3E and Figure 2E**) along with reduced protein expression of LGR5, CD133, CD44 and EpCAM (**Figure 2E**).

7. While the authors determine CRC stem cells by markers including CD133 and LGR5, some experiments can be confirmed with additional stem cell markers including CD44 and EpCAM.

Response: We have now confirmed our results with CD44 in **Figure 2E, 3F, and 3G**, and EpCAM for **Figure 2E**. These data are described in **Results**, as follows:

Results (p. 9, line 182-184):

ALKBH5 knockout was confirmed (**Figure S3E and Figure 2E**) along with reduced protein expression of LGR5, CD133, CD44 and EpCAM (**Figure 2E**).

Results (p. 11, line 208-212):

Moreover, in PDO 816 and 828, which harbored APC mutation and activated WNT/ β -catenin pathway (**Table S5 and S6**), ALKBH5 knockdown significantly impaired the growth, self-renewal and expression of LGR5, CD133 and CD44 (**Figure 3F**). Whereas ectopic expression of ALKBH5 showed opposite phenotypes (**Figure 3G**), confirming that ALKBH5 also boosts stemness traits in CRC CSCs and PDOs.

8. In Figure 3, ALKBH5 overexpression experiment cannot address whether the function of ALKBH5 in promoting CSC proliferation and self-renewal is dependent on its enzymatic activity. Does overexpression of ALKBH5 or ALKBH5 mutant in ALKBH5 KD cells rescue these phenotypes?

Response: We now performed the overexpression of ALKBH5^{WT} or ALKBH5^{H204A} in ALKBH5-KD cells (**Figure S5J**), showing that overexpression of ALKBH5^{WT}, but not ALKBH5^{H204A}, restored self-renewal in the limiting dilution assay. These findings are now described in **Results** section, as follows:

Results (p. 13, line 268-270):

Overexpression of ALKBH5^{WT}, but not ALKBH5^{H204A} restored self-renewal capacity caused by ALKBH5-knockdown (**Figure S5J**), implying that ALKBH5-induced CSC properties in an m⁶A-dependent manner.

9. In Figure 3C, you have presented the positive regulation of CD133 and CD44 by ALKBH5. What about LGR5? Also, please show the expression of CD44 in Figure 3F and 3G.

Response: We have now added the results for LGR5 in **Figure 3C** and CD44 in **Figure 3F** and **3G**.

10. The knockdown efficiency of endogenous ALKBH5 in POP66 cells is not significant in Figure 4L. It may not be able to rule out the possibility that m6A is also downregulated by endogenous ALKBH5.

Response: We have now repeated all the experiments in **Figure 4M**, **S5L** and **S5M** with improved knockdown of endogenous ALKBH5.

11. Please clarify the difference between A5-dCas13b and dCas13b-A5, as well as the different expression level of these two genes, as shown in Figure 4L.

Response: A5-dCas13b and dCas13b-A5 are both recombinant fusion proteins but with dCas13b added to C-terminus and N-terminus of ALKBH5 sequence, respectively. The plasmid constructs are now shown in **Figure S5K**. The differences in expression levels of the two proteins might be due to variation in transduction efficiencies. We have now repeated the experiments with improved results in **Figure 4M**, **S5L** and **S5M**.

12. In Figure 6G, which specific domain of FAM84A binds to β -catenin? Does mutating these binding sites disrupt the FAM84A's ability to inhibit β -catenin function and suppress CRC stem cell functions?

Response: To identify binding sites of FAM84A to β -catenin, we first performed cross-linking and mass spectrometry using recombinant proteins. This led to the identification of peptides VVQNACGHLGLK (residues 192-203) and NKVHTAR (residues 252-258) in FAM84A interacting with β -catenin (**Figure S9H**). Molecular docking then narrowed down that to ASN-252, VAL-254, LYS-246, PRO-103, and ARG-183 as key amino acid residues involved (**Figure S9I**). Hence, we constructed FAM84A^{Mut} (residues 252-254; NKV→AAA) plasmid. FAM84A^{Mut} overexpression failed to down-regulate β -catenin (**Figure S9J**), contrary to the inhibitory effects of FAM84A^{WT} on both total and active β -catenin expression (**Figure S9J**). Moreover, FAM84A^{Mut} failed to reduce CRC stem cell function, as evidenced by LDA assay (**Figure S9K**). These findings are now added to **Results** section, as follows:

Results (p. 18, line 379-387):

Cross-linking and mass spectrometry analyses of recombinant FAM84A and β -catenin revealed 2 FAM84A peptides VVQNACGHLGLK (residues 192-203) and NKVHTAR (residues 252-258) interacting with β -catenin (**Figure S9H**). Molecular docking showed that ASN-252, VAL-254, LYS-246, PRO-103, and ARG-183 as key amino acid residues involved (**Figure S9I**). Contrary to FAM84A^{WT}, FAM84A^{Mut} overexpression (residues 252-254; NKV→AAA) failed to down-regulate β -catenin (**Figure S9J**) or reduce CSC stemness, as determined by LDA assay (**Figure S9K**), validating that

residues 252-254 are critical for the ability of FAM84A to bind and inhibit β -catenin function.

13. Have you considered the possibility that FAM84A downregulates β -catenin through the lysosomal pathway?

Response: To evaluate this possibility, we have now treated CSCs with overexpression of FAM84A with or without the lysosome inhibitor Chloroquine (CQ, 10 μ M for 24h), and analyzed β -catenin expression. Neither total nor active β -catenin were rescued by CQ (**Figure S9F**), implying that lysosomal pathway is not involved in downregulation of β -catenin by FAM84A. This result is now described in **Results** section, as follows:

Results (p. 17, line 361-363):

Whereas Chloroquine (CQ) had no rescue effect, suggesting that the lysosomal pathway is not involved in FAM84A-mediated degradation of β -catenin (**Figure S9F**).

Minor points:

1. For the introduction, please expand on the background of ALKBH5's role in cancer, with a particular focus on its function in cancer stem cells.

Response: We have now incorporated background information on ALKBH5's role in cancer stem cells in the **Introduction**, as follows:

Introduction (p. 4, line 74-79):

ALKBH5 has been reported to exert oncogenic [9-12] or tumor suppressive function [13-17] in a context-dependent manner. On the other hand, several studies have suggested that ALKBH5 selectively enhances self-renewal capacity of cancer stem cells (CSCs) in solid tumors and leukemia [18-21] and also boosts stemness properties in several tumor types [11, 22-25] through activation of WNT/ β -catenin signaling. Nevertheless, the role of ALKBH5 in CRC stem cells remains incompletely understood.

2. Please add the x-axis label in Figure 1R.

Response: Added.

3. Labeling the ladder and protein sizes in your Western blot results would enhance clarity.

Response: We have now added the ladder and protein sizes for all Western blots.

4. In Figure 5K, it's difficult to differentiate between the sgNC+siIGF2BP1 and A5-KO+siIGF2BP1 groups.

Response: Revised.

5. We would like to know if the authors performed RNA-seq and Ribo-seq at the same time or separately. It is important to perform RNA-seq and Ribo-seq at the same time as the RNA-seq data was used as an input. It is fine if the authors repeats RNA-seq in separate times. However, we did not find all the data (input). Since we do not have

expertise in bioinformatics, we cannot evaluate the quality of Raw data from these studies.

Response: We have conducted parallel RNA-seq and Ribo-seq with the same batch of samples in order to minimize batch effects. Matched RNA-seq data is then used as input for Ribo-seq data analysis. All data has been deposited in Bioproject: PRJNA1086651, PRJNA1086679 and PRJNA1086204. We have now explained this in **Supplementary Methods**, as follows:

Methods (p. 41, line 876-879):

Data availability

Source data are provided with this paper. All RNA-seq, MeRIP-seq and ribo-seq data were uploaded to Sequence Read Archive (SRA): PRJNA1086651, PRJNA1086679 and PRJNA1086204 with Reviewer link listed in **Supplementary Table 7**.

Response to Reviewer 5:

The authors have established the causal relationships between ALKBH5, m6A, FAM84A, β -catenin, tumor stemness, and chemotherapy resistance in this study. By integrating mouse and organoid models with a human cohort, they have described the clinical relevance of ALKBH5 in CRC chemoresistance.

Based on the motif analysis, the quality of the MeRIP-seq data appears to be suboptimal (refer to Q15), and the basic quality control data for both MeRIP-seq and Ribo-seq are incomplete. I suggest that the authors incorporate further analyses to confirm the quality of the MeRIP-seq and Ribo-seq data (refer to Q16-17). Additionally, it is essential to include data on the relationship between m6A modifications changes and RNA expression/translation efficiency in CSCs in response to ALKBH5 perturbations (refer to Q18).

Overall, this is a well-organized study for Nat Communication. However, there were still many questions that need to be fully explained.

1. Prior studies have indicated that the m6A writer METTL3 promotes CRC stemness (<https://doi.org/10.1186/s12943-019-1038-7>). The role of ALKBH5, which demethylates RNA m6A, in enhancing the stemness of CSCs is intriguing. Did the m6A modification of FAM84A change in response to METTL3 perturbation? Did the authors assess the expression changes of METTL3 and other m6A regulators in response to ALKBH5 knockout or knockdown to prevent compensatory effects?

Response: We have now analyzed METTL3-knockdown POP66 and CSC28, revealing that FAM84A m⁶A modification (**Figure S6D**), mRNA expression (**Figure S6E**), and protein expression (**Figure S6F**) were not modulated by shMETTL3.

Moreover, we found that ALKBH5 knockdown did not modulate the expression of m⁶A writer METTL3 or m⁶A eraser FTO, suggesting that there are no compensatory effects for m⁶A modification enzymes (**Figure S6G**). For m⁶A readers, ALKBH5 knockdown down-regulated YTHDF1/2/3 and YTHDC1/2 (**Figure S6G**). These data are now described in **Results** section, as follows:

Results (p. 15, line 296-300):

METTL3 knockdown in CSCs had on significant effect on FAM84A m⁶A modification (**Figure S6D**), mRNA expression (**Figure S6E**), and protein expression (**Figure S6F**), inferring that FAM84A m⁶A levels are primarily regulated by ALKBH5 in CSCs. ALKBH5 knockdown in CSCs did not induce compensatory alterations in METTL3 or FTO (**Figure S6G**).

2. A summary and discussion of the function of other m6A regulators in colonic stem cells and CRC should be included in an appropriate section.

Response: We have now added background information in **Introduction**, as follows:

Introduction (p. 4, line 67-72):

Emerging studies suggest the potential roles of m⁶A regulators in modulating stemness and tumorigenesis in colon. M⁶A writers METTL3 and METTL14 have been reported to sustain self-renewal capacity in the normal colon [26, 27]. METTL3 and m⁶A readers including YTHDF1/2 and IGF2BP1/2 have been implicated in colorectal tumorigenesis, chemoresistance [28-33], and metastasis [33-36] through modulating the translation of m⁶A downstream targets.

3. *There is ongoing debate regarding how ALKBH5 in CRC tumor cells reshapes the surrounding tumor microenvironment. Some researchers have found hyperactivation of WNT signaling in ALKBH5-high patients, leading to the establishment of an immunosuppressive TME (10.1053/j.gastro.2023.04.032). However, other studies show a positive correlation between ALKBH5 and CD8+ T cell infiltration (https://doi.org/10.1016/j.tranon.2023.101683). Does the study provide any insights into these debates? Does the data support any mechanisms proposed by these studies?*

Response: We agree that there are disparities regarding the effect of ALKBH5 in tumor microenvironment (TME) [46, 47]. In this study, we confirmed the role of ALKBH5 in activating WNT/ β -catenin pathway by colon stem cell-specific ALKBH5 knockin and knockout mice, consistent with that reported by Zhai et al. [47]. However, in this work we did not evaluate TME, and future studies will be conducted to elucidate the function of CSC-specific ALKBH5 in modifying TME. We have now added this to **Discussion** section, as follows:

Discussion (p. 24, line 500-501):

Consistent with our data, we and others have shown that ALKBH5 positively regulates WNT signaling in CRC [47] and other cancers [48].

4. *The authors used bulk analysis to establish the correlation between ALKBH5 and stem cell marker expression. Is the tumor cell the primary source of ALKBH5 in CRC TME? Do other major subsets in TME, such as cancer-associated fibroblasts and myeloid subsets, contribute to this correlation?*

Response: To analyze ALKBH5 expression in among different cell types in CRC TME, we analyzed a published scRNA-seq dataset (GSE132465). A substantial proportion of epithelial/tumor cells (n=17469) was found to express ALKBH5 (>30%) as compared to other cell types, including stromal cells, myeloid cells, T cells, and B cells (**Figure S1D**). Further correlation analysis of ALKBH5 with CD133 and LGR5 in individual cell populations demonstrated that in epithelial/tumor cells ALKBH5 most significantly correlated with CD133 (P<0.001) and LGR5 (P<0.001) (**Figure S1E**). In myeloid cells, ALKBH5 and LGR5 showed positive correlation (P<0.01), while no correlation was found in stromal cells, T cells, or B cells (**Figure S1E**). These data are now described in **Results**, as follows:

Results (p. 6, line 106-114):

To ask if ALKBH5 correlates with stemness markers among different cell types in CRC

tumor microenvironment (TME), we analyzed a single-cell RNA sequencing (scRNA-seq) dataset (GSE132465) [49]. A higher proportion of epithelial/tumor cells expressed ALKBH5 (>30%) compared to stromal cells, myeloid cells, T cells, and B cells (**Figure S1D**). Furthermore, ALKBH5 positively correlated with CD133 ($P < 0.001$) and LGR5 ($P < 0.001$) in epithelial/tumor cells (**Figure S1E**). ALKBH5 positively correlated with LGR5 in myeloid cells, whilst no correlation was observed in stromal cells, T cells, or B cells (**Figure S1E**). This suggests that tumor cells are the primary source of ALKBH5 in CRC TME that corresponds to stemness properties.

5. *The quality of figure legends needs improvement, with some details missing; for example, the legend of Figure 3A.*

Response: We have thoroughly revised all the figure legends to include more details.

6. *Figure 1B lacks ticks on the y-axis.*

Response: Revised.

7. *The quality of the CD133 staining is unconvincing, as it does not overlap with DAPI staining and does not align with cellular morphology (Fig 1T and 2I)*

Response: We have now optimized the protocol for CD133 immunofluorescence, and updated results are now presented in **Figure 1P** and **Figure 2I**.

8. *In figure 2E, two ALKBH5 knockout samples show comparable expression levels to WT samples but loss LGR5 and CD133 signaling, please explain it.*

Response: Since we only targeted ALKBH5 deletion in intestine stem cell lineage, it is probable that ALKBH5 is still expressed in non-stem cell lineages. Thus, the apparent reductions in ALKBH5 might be less dramatic compared to LGR5 and CD133. We have now repeated western blot analysis with improved results (revised **Figure 2E**).

9. *In figure 2F, the color of the statistical bar plot is difficult to read.*

Response: Revised.

10. *In figure 2G, the title of the y-axis overlaps with the context of y-axis.*

Response: Revised.

11. *In Figure 2E, after ALKBH5 knockout, no CD133 can be detected in tumor tissue, while in Figure 2K, the CD133 signal only shows a slight decrease in cKO group, please explain this inconsistency.*

Response: The data displayed in **Figure 2K** was performed after >5 passages. However, as most cKO organoids failed to survive, subsequent outgrowth might select for clones with high stemness/CD133 expression. We have now repeated flow cytometry on cKO organoids with minimal passaging (2nd passage), and demonstrated dramatic reductions in surface CD133 expression (revised **Figure 2K**), consistent with our results in **Figure 2E**.

12. Figure 2K lacks x-axis ticks and text.

Response: Added.

13. Figure 2H-I lacks scale bars in the zoomed-in panels.

Response: Added.

14. Did the WNT signaling are significant in GSEA analysis of RNA-seq? the p-values for each signaling pathway need to be added (Figure 4C/E).

Response: All the enriched pathways displayed in **Figure 4C and 4E** were statistically significant, and we have now added p-values to all panels.

15. The motif did not enrich in the GGACU motif; the GGACA also shows a weak p-value. Are there more significant motif in your peaks? And what these peaks talking about?

Response: We have now optimized bioinformatic pipeline and re-analyzed m⁶A peak calling using MeTPeak, and performed motif analysis based on the top 5,000 peaks in each sample. In CSC-shNC and CSC-shALKBH5 cells, the motif “WGGAC” is top enriched with $P=1.0e^{-41}$ and $P=1.0e^{-112}$, respectively (**Figure 4D**). In CSC-EV cells, “TGGAC” is a top enriched motif with $P=1.0e^{-138}$, whereas “AGGAC” is enriched in CSC-ALKBH5 cells with $P=1.0e^{-74}$ (**Figure S5C**). These results are consistent with the m⁶A consensus sequence “RRACH” with much higher confidence previously [50, 51], thus confirming the validity of m⁶A pulldown and sequencing assay. These data are now described in **Results**, as follows:

Results (p. 11, line 222-225):

We mapped the m⁶A methylomes of CSC28 cells by m⁶A-sequencing. The GGAC consensus motif is highly enriched within m⁶A sites in shNC, shALKBH5, (**Figure 4D**), EV, and ALKBH5-overexpressing CSC28 (**Figure S5C**). Other m⁶A motifs are listed in **Supplementary Table 8**.

16. It is necessary to include the percentage of m⁶A-loss and m⁶A-gain peaks in response to perturbation.

Response: We have added the percentage of m⁶A-loss and m⁶A-gain peaks in response to ALKBH5 knockdown and overexpression in **Results**, as follows:

Results (p. 11, line 225-229):

MeRIP-seq analysis revealed that in CSC28-shALKBH5 vs. shNC (total 58784 peaks), 12825 (21.82%) peaks were up-regulated whereas 9887 (16.82%) peaks were reduced in shALKBH5 group (**Figure 4D**). For ALKBH5-OE vs. EV (total 55322 peaks), 9652 (17.45%) peaks were up-regulated, together with 9855 (17.81%) down-regulated peaks (**Figure 4D**).

17. To establish the reliability of Ribo-seq, the CDS distribution, the enrichment of translation initiation sites, and tri-nucleotide periodicity analysis must be provided.

Response: We have now analyzed the reads length of Ribo-seq, CDS distribution and the enrichment of translation initiation sites with 3-nt periodicity. Ribosome-protected fractions (RPFs) predominantly map with read length distribution of 28~30nt (**Figure S5D**). 85.06% P-sites are mapped in CDS in CSC28-shNC cells, 78.39% P-sites are mapped in CDS in CSC28-shALKBH5 cells, 84.27% P-sites are mapped in CDS in CSC28-EV cells, and 82.64% P-sites are mapped in CDS in CSC28-ALKBH5-OE cells, indicating the majority of the P-sites are mapped in CDS (**Figure S5E**). In addition, RPFs from all samples are enriched at the start codon (**Figure S5F**) and displayed a 3-nt periodicity (**Figure S5F**). These are explained in **Results** section, as follows:

Results (p. 12, line 230-232):

Analysis of CDS distribution (**Figure S5D-S5E**), enrichment of transcription initiation sites and tri-nucleotide periodicity (**Figure S5F**) confirmed the validity of our Ribo-seq dataset.

18. The alterations in RNA expression and translation efficiency of the identified m⁶A-gain and m⁶A-loss genes in response to perturbation should be examined, with the remaining genes used as controls.

Response: We have now performed additional analysis by first defining m⁶A-gain and m⁶A-loss genes as those exhibiting consistent increase or decrease of m⁶A levels across all sites, respectively. Since ALKBH5 functions as an m⁶A eraser, we compared m⁶A-gained genes (1275) in shALKBH5 as compared to shNC ($P < 0.05$, $|FC| > 2$), using other genes (9577) as controls. mRNA expression of m⁶A-gained genes significantly reduced compared to control genes (**Figure S5G**), while translation efficiency demonstrated no change (**Figure S5G**). Next, we assessed m⁶A-loss genes (1681) in ALKBH5-OE cells compared to EV cells ($P < 0.05$, $|FC| > 2$) with 9250 genes as control. As shown in **Figure S5H**, mRNA expression of m⁶A-loss genes significantly increased compared to control genes, with no change in translation efficiency. These data are now described in **Results**, as follows:

Results (p. 12, line 236-242):

Upon ALKBH5 knockdown, m⁶A-gained genes ($P < 0.05$, $|FC| > 2$) (1275) demonstrated decreased mRNA expression compared to unaltered genes (9577), without changing the translation efficiency (**Figure S5G**). Whereas m⁶A-loss genes ($P < 0.05$, $|FC| > 2$) (1681) induced by ALKBH5 overexpression showed up-regulated mRNA levels compared to unaltered genes (9250), but no change in translation efficiency (**Figure S5H**). This implies that ALKBH5-mediated m⁶A demethylation predominantly modulates mRNA expression.

19. The color bar of Figure 4F is blank.

Response: Revised.

20. The m⁶A signaling in Figure 4G needs to be adjusted using RNA-seq input.

Response: The data shown has already been adjusted for RNA input. We have added

this description to **Figure 4G and legend**.

21. *ALKBH5 overexpression does not show significant m⁶A level changes in FAM84A in Figure 4G.*

Response: As there are multiple m⁶A peaks on FAM84A, we focused on m⁶A sites with FC>2, P< 0.01. In **Figure 4G**, only the m⁶A peaks at 5'UTR were down-regulated by ALKBH5 overexpression, and we focused on 5'UTR region of FAM84A in subsequent validation. We have now showed an enlarged version of **Figure 4G** in **Figure S5I** for more easy visualization.

22. *The significance between each group in Figure 5J needs to be added.*

Response: We have now added p-values between each group in **Figure 5J**.

23. *The color in Figure 5K is missing.*

Response: Revised.

24. *Does ALKBH5 expression correlate with survival in your cohort?*

Response: With the same CRC cohort, we have demonstrated that ALKBH5 expression significantly correlated with poor survival in our previous publication [37].

25. *Explain the strategy used to define FAM84A high and low in your cohort.*

Response: We have now provided the strategy to define FAM84A-high and -low based in **Supplementary Methods** section, as follows:

Methods (p. 35, line 738-741):

FAM84A immunohistochemistry staining was scored by a pathologist on a scale from 0-3, where 0=negative; 1=weak positive; 2=moderate positive; and 3=strong positive. In **Figure 6A**, scores of 0-1 and 2-3 were defined as FAM84A-low and FAM84A-high, respectively.

26. *There are several inconsistent colors in plots and color keys; please check.*

Response: Revised.

References:

1. Cancer Genome Atlas, N., *Comprehensive molecular characterization of human colon and rectal cancer*. Nature, 2012. **487**(7407): p. 330-7.
2. Becker, W.R., et al., *Single-cell analyses define a continuum of cell state and composition changes in the malignant transformation of polyps to colorectal cancer*. Nat Genet, 2022. **54**(7): p. 985-995.
3. Qin, X., et al., *An oncogenic phenoscape of colonic stem cell polarization*. Cell, 2023. **186**(25): p. 5554-5568 e18.
4. Hinoi, T., et al., *Mouse model of colonic adenoma-carcinoma progression based on somatic Apc inactivation*. Cancer Res, 2007. **67**(20): p. 9721-30.
5. Wong, C.C., et al., *The cholesterol uptake regulator PCSK9 promotes and is a therapeutic*

- target in APC/KRAS-mutant colorectal cancer*. Nature Communications, 2022. **13**(1): p. 3971.
6. Hwang, S.-Y., et al., *Direct Targeting of β -Catenin by a Small Molecule Stimulates Proteasomal Degradation and Suppresses Oncogenic Wnt/ β -Catenin Signaling*. Cell Reports, 2016. **16**(1): p. 28-36.
 7. Lu, D., et al., *Salinomycin inhibits Wnt signaling and selectively induces apoptosis in chronic lymphocytic leukemia cells*. Proceedings of the National Academy of Sciences of the United States of America, 2011. **108**(32): p. 13253-13257.
 8. Emami, K.H., et al., *A small molecule inhibitor of beta-catenin/CREB-binding protein transcription [corrected]*. Proceedings of the National Academy of Sciences of the United States of America, 2004. **101**(34): p. 12682-12687.
 9. Yu, J., et al., *ALKBH5 activates CEP55 transcription through m6A demethylation in FOXP2 mRNA and expedites cell cycle entry and EMT in ovarian cancer*. Biology Direct, 2024. **19**(1): p. 105.
 10. Hua, X., et al., *ALKBH5 promotes non-small cell lung cancer progression and susceptibility to anti-PD-L1 therapy by modulating interactions between tumor and macrophages*. Journal of Experimental & Clinical Cancer Research : CR, 2024. **43**(1): p. 164.
 11. Wang, Q., et al., *The demethylase ALKBH5 mediates ZKSCAN3 expression through the m6A modification to activate VEGFA transcription and thus participates in MNNG-induced gastric cancer progression*. Journal of Hazardous Materials, 2024. **473**: p. 134690.
 12. Xu, X., et al., *FSH induces EMT in ovarian cancer via ALKBH5-regulated Snail m6A demethylation*. Theranostics, 2024. **14**(5): p. 2151-2166.
 13. Zheng, Z., et al., *ALKBH5 suppresses gastric cancer tumorigenesis and metastasis by inhibiting the translation of uncapped WRAP53 RNA isoforms in an m6A-dependent manner*. Molecular Cancer, 2025. **24**(1): p. 19.
 14. Ge, J., et al., *RNA demethylase ALKBH5 suppresses tumorigenesis via inhibiting proliferation and invasion and promoting CD8+ T cell infiltration in colorectal cancer*. Translational Oncology, 2023. **34**: p. 101683.
 15. Hu, Y., et al., *Demethylase ALKBH5 suppresses invasion of gastric cancer via PKMYT1 m6A modification*. Molecular Cancer, 2022. **21**(1): p. 34.
 16. Li, W., et al., *ALKBH5 inhibits thyroid cancer progression by promoting ferroptosis through TIAM1-Nrf2/HO-1 axis*. Molecular and Cellular Biochemistry, 2023. **478**(4): p. 729-741.
 17. He, Y., et al., *ALKBH5-mediated m6A demethylation of KCNK15-AS1 inhibits pancreatic cancer progression via regulating KCNK15 and PTEN/AKT signaling*. Cell Death & Disease, 2021. **12**(12): p. 1121.
 18. Cao, L., et al., *A LATS2 and ALKBH5 positive feedback loop supports their oncogenic roles*. Cell Reports, 2024. **43**(4): p. 114032.
 19. Zhang, S., et al., *m6A Demethylase ALKBH5 Maintains Tumorigenicity of Glioblastoma Stem-like Cells by Sustaining FOXM1 Expression and Cell Proliferation Program*. Cancer Cell, 2017. **31**(4).
 20. Zhang, C., et al., *Hypoxia induces the breast cancer stem cell phenotype by HIF-dependent and ALKBH5-mediated m6A-demethylation of NANOG mRNA*. Proceedings of the National Academy of Sciences of the United States of America, 2016. **113**(14): p. E2047-E2056.

21. Gao, Y., et al., *ALKBH5 modulates hematopoietic stem and progenitor cell energy metabolism through m6A modification-mediated RNA stability control*. Cell Reports, 2023. **42**(10): p. 113163.
22. Meng, Y., et al., *Hepatitis B Virus-Mediated m6A Demethylation Increases Hepatocellular Carcinoma Stemness and Immune Escape*. Molecular Cancer Research : MCR, 2024. **22**(7): p. 642-655.
23. Chen, G., et al., *Hypoxia induces an endometrial cancer stem-like cell phenotype via HIF-dependent demethylation of SOX2 mRNA*. Oncogenesis, 2020. **9**(9): p. 81.
24. Liu, X., et al., *M6A demethylase ALKBH5 regulates FOXO1 mRNA stability and chemoresistance in triple-negative breast cancer*. Redox Biology, 2024. **69**: p. 102993.
25. Shen, C., et al., *RNA Demethylase ALKBH5 Selectively Promotes Tumorigenesis and Cancer Stem Cell Self-Renewal in Acute Myeloid Leukemia*. Cell Stem Cell, 2020. **27**(1).
26. Zhang, T., et al., *m6A mRNA modification maintains colonic epithelial cell homeostasis via NF-κB-mediated antiapoptotic pathway*. Science Advances, 2022. **8**(12): p. eabl5723.
27. Danan, C.H., et al., *Intestinal transit-amplifying cells require METTL3 for growth factor signaling and cell survival*. JCI Insight, 2023. **8**(23).
28. Xu, C., et al., *FTO facilitates colorectal cancer chemoresistance via regulation of NUPR1-dependent iron homeostasis*. Redox Biology, 2025. **83**: p. 103647.
29. Tan, Y.-T., et al., *WTAP weakens oxaliplatin chemosensitivity of colorectal cancer by preventing PANoptosis*. Cancer Letters, 2024. **604**: p. 217254.
30. Lin, X., et al., *Gegen Qinlian Decoction reverses oxaliplatin resistance in colorectal cancer by inhibiting YTHDF1-regulated m6A modification of GLS1*. Phytomedicine : International Journal of Phytotherapy and Phytopharmacology, 2024. **133**: p. 155906.
31. Chen, L., et al., *LIMK1 m6A-RNA methylation recognized by YTHDC2 induces 5-FU chemoresistance in colorectal cancer via endoplasmic reticulum stress and stress granule formation*. Cancer Letters, 2023. **576**: p. 216420.
32. Chen, L.-J., et al., *IGF2BP3 promotes the progression of colorectal cancer and mediates cetuximab resistance by stabilizing EGFR mRNA in an m6A-dependent manner*. Cell Death & Disease, 2023. **14**(9): p. 581.
33. Lin, Z., et al., *N6-methyladenosine demethylase FTO enhances chemo-resistance in colorectal cancer through SIVA1-mediated apoptosis*. Molecular Therapy : the Journal of the American Society of Gene Therapy, 2023. **31**(2): p. 517-534.
34. Wu, J., et al., *METTL16 Promotes Stability of SYNPO2L mRNA and leading to Cancer Cell Lung Metastasis by Secretion of COL10A1 and attract the Cancer-Associated Fibroblasts*. International Journal of Biological Sciences, 2024. **20**(11): p. 4128-4145.
35. Ouyang, P., et al., *METTL3 recruiting M2-type immunosuppressed macrophages by targeting m6A-SNAIL-CXCL2 axis to promote colorectal cancer pulmonary metastasis*. Journal of Experimental & Clinical Cancer Research : CR, 2024. **43**(1): p. 111.
36. Wang, X., et al., *m6A modified BACE1-AS contributes to liver metastasis and stemness-like properties in colorectal cancer through TUFT1 dependent activation of Wnt signaling*. Journal of Experimental & Clinical Cancer Research : CR, 2023. **42**(1): p. 306.
37. Zhai, J., et al., *ALKBH5 Drives Immune Suppression Via Targeting AXIN2 to Promote Colorectal Cancer and Is a Target for Boosting Immunotherapy*. Gastroenterology, 2023. **165**(2): p. 445-462.

38. Liang, Z., et al., *ALKBH5 governs human endoderm fate by regulating the DKK1/4-mediated Wnt/ β -catenin activation*. Nucleic Acids Research, 2024. **52**(18): p. 10879-10896.
39. Tang, B., et al., *m6A demethylase ALKBH5 inhibits pancreatic cancer tumorigenesis by decreasing WIF-1 RNA methylation and mediating Wnt signaling*. Molecular Cancer, 2020. **19**(1): p. 3.
40. Shriwas, O., et al., *DDX3 modulates cisplatin resistance in OSCC through ALKBH5-mediated m(6)A-demethylation of FOXM1 and NANOG*. Apoptosis, 2020. **25**(3-4): p. 233-246.
41. Yu, F., et al., *Post-translational modification of RNA m6A demethylase ALKBH5 regulates ROS-induced DNA damage response*. Nucleic Acids Research, 2021. **49**(10): p. 5779-5797.
42. Yu, F., et al., *KRAS mutants confer platinum resistance by regulating ALKBH5 posttranslational modifications in lung cancer*. The Journal of Clinical Investigation, 2025. **135**(6).
43. Meng, S., et al., *PRMT5-Mediated ALKBH5 Methylation Promotes Colorectal Cancer Immune Evasion via Increasing CD276 Expression*. Research (Washington, D.C.), 2025. **8**: p. 0549.
44. Ding, K., et al., *Liver ALKBH5 regulates glucose and lipid homeostasis independently through GCGR and mTORC1 signaling*. Science (New York, N.Y.), 2025. **387**(6737): p. eadp4120.
45. Vasaikar, S., et al., *Proteogenomic Analysis of Human Colon Cancer Reveals New Therapeutic Opportunities*. Cell, 2019. **177**(4).
46. Ge, J., et al., *RNA demethylase ALKBH5 suppresses tumorigenesis via inhibiting proliferation and invasion and promoting CD8(+) T cell infiltration in colorectal cancer*. Transl Oncol, 2023. **34**: p. 101683.
47. Zhai, J., et al., *ALKBH5 Drives Immune Suppression Via Targeting AXIN2 to Promote Colorectal Cancer and Is a Target for Boosting Immunotherapy*. Gastroenterology, 2023. **165**(2): p. 445-462.
48. Wang, C., et al., *ALKBH5 facilitates tumor progression via an m6A-YTHDC1-dependent mechanism in glioma*. Cancer Letters, 2025. **612**: p. 217439.
49. Lee, H.-O., et al., *Lineage-dependent gene expression programs influence the immune landscape of colorectal cancer*. Nature Genetics, 2020. **52**(6): p. 594-603.
50. Wei, C.M. and B. Moss, *Nucleotide sequences at the N6-methyladenosine sites of HeLa cell messenger ribonucleic acid*. Biochemistry, 1977. **16**(8): p. 1672-1676.
51. Dominissini, D., et al., *Topology of the human and mouse m6A RNA methylomes revealed by m6A-seq*. Nature, 2012. **485**(7397): p. 201-206.

Response to the comments of referees in relation to the manuscript:

“RNA N⁶-methyladenosine eraser ALKBH5 drives stemness and chemoresistance of colorectal cancer and is a therapeutic target” (NCOMMS-24-52715A)

Comments are written in *italics* and responses in plain texts. Modifications made to the manuscript are denoted in **BLUE**.

Response to Reviewer 1:

The authors have improved the manuscript. While some sections are more novel and compelling than others, the overall volume of data presented is substantial. So given the depth of the results, in my opinion the manuscript is now suitable for publication in Nature Communications.

Response: We thank the reviewer for the positive comments.

Response to Reviewer 2:

The authors have addressed most of the issues Reviewer #2 pointed out and made efforts to improve the manuscript. As a result, this revised manuscript has been significantly improved and is now at a scientifically acceptable level for publication. However, a few issues remain.

1. As I have pointed out before, figures are packed due to too much number of panels. In particular, two organoid strains, POP66 and CSC28, are always shown in Fig. 3-4 and 6-8. In most panels, these two organoids show almost the same behaviour, so I think it would be fine to follow the previous suggestion as was done in Fig. 5, and show only one of them, and send another to supplemental figures. Since supplemental is already saturated, it may be unavoidable, but having two almost identical figures in same time to show one event causes congestion. This comment is a suggestion to make the manuscript better; not a mandatory instruction, so depends on author's decision.

Response: We thank the reviewer for their suggestion. Considering the importance of demonstrating experimental reproducibility for our key results, we have chosen to keep the current presentation format in the main figures.

2. In the answer from authors on major point #11, they discuss that ALKBH5 was knocked out only in the intestinal stem cells with Lgr5-Cre, and it was not knocked out in whole intestine with Cdx2-Cre or Villin-Cre. However, if the samples were taken 80 days after tamoxifen administration as in Fig. 2B, almost everything of the intestinal cells should have been replaced by knockout ISC-derived cells, since the lifespan of intestinal epithelial cells is only a few days, and even Paneth cells, which have a long lifespan, are only 1-2 months. Please indicate how many days after tamoxifen administration the samples were taken. Furthermore, Fig. 1B shows that ALKBH5 is expressed only in the basal stem cell region, but not in the upper part of the crypt, so the authors' answer that small detection of ALKBH5 protein was owing to that they did not knockout ALKBH5 in whole intestine, seems to be inconsistent. Because once ALKBH5 is knocked out in ISCs, no other cells express ALKBH5.

Although the current figure has been improved and no additional experiments are likely

to be necessary, a logical explanation is still needed as to where the ALKBH5 signal in the previous figure came from.

Response: We agree that, in principle, complete ALKBH5 ablation in Lgr5-expressing lineage can lead to the loss of ALKBH5 across the entire intestinal tract. However, the efficiency of Cre-loxP recombination with inducible systems like LGR5-Cre^{ERT2} [1] can be variable and incomplete. Consequently, this inherent variability likely explains the low-level residual ALKBH5 expression detected in a subset of mice. We have added this explanation to **Discussion** section, as follows:

Discussion (p. 25, line 532-534):

Inducible Cre-loxP systems, such as LGR5-Cre^{ERT2}, are known to exhibit variable recombination efficiency. Consistent with this, low-level residual ALKBH5 expression was detected in a subset of Alkbh5^{flox/flox}LGR5-Cre^{ERT2} mice.

3. Please increase the signal intensity of ALKBH5 in green in Fig. 1B and S3F and yellow in WT in Fig. S3A-B so that it is easier to see.

Response: We have now updated **Figure 1B**, **Figure S3F** (ALKBH5 staining, green), and **Figure S3A-3B** (ALKBH5 staining, yellow) with enhanced signal intensity.

4. In figures having graphs, the x-axis is always divided into two parts regardless of the number of experimental groups. When making graphs using general software, the number of divisions on the X-axis is usually optimized depending on the number of experimental groups.

Response: We have carefully revised the number of divisions on the X-axis according to the number of experimental groups.

5. Molecular weight markers were included in most of the immunoblotting images, but they were not included in the anti-ubiquitin blots particularly interested in (Fig. 6K, 6N). Please show them.

Response: We have now added molecular weight markers to **Figure 6K**, **6N**, and **S9D** as suggested.

Response to Reviewer 3:

The authors have provided satisfactory and comprehensive responses to the reviewer's comments.

Response: We thank the reviewer for the positive comments.

Response to Reviewer 4:

Response: We thank the reviewer for the positive comments.

Response to Reviewer 5:

The authors have addressed most of my previous concerns, and the manuscript now shows significant improvement in quality. I only have a few minor suggestions specifically regarding Questions 4, 5, and 14 for further refinement:

1 Q4: While the p-values are statistically significant ($p < 0.001$), the near-zero Pearson coefficients (e.g., $PCC \approx 0.1$) suggest no meaningful biological association between ALKBH5 and stemness markers in epithelial cells.

Response: The low Pearson coefficients is due to the heterogeneity intrinsic to single-cell RNA sequencing. To address this issue, we now confined our analysis to cells that express detectable levels of both ALKBH5 and stemness markers (LGR5 or CD133). As shown in **Figure S1E**, we observed significant positive correlations with improved Pearson coefficients in the tumor cell subset between ALKBH5 and LGR5 ($R=0.39$, $P < 0.0001$, $n=1295$ cells), and ALKBH5 and CD133 ($R=0.25$, $P < 0.0001$, $n=2867$). But for other cells in the TME (including stromal, myeloid, T cell, and B cell subsets), there are few numbers of cells (< 30 cells) meeting this criterion, precluding robust correlation analysis. Our re-analysis robustly demonstrates that 1) tumor cells, but not other cell subsets, constitute the primary cellular source of ALKBH5 within the CRC TME; and 2) only tumor cells have high co-expression of ALKBH5 and stemness markers LGR5 and CD133, thereby reinforcing the link between ALKBH5 and stemness in tumor cells. These data are now described in **Results**, as follows:

Results (p.6, line 110-115):

Furthermore, within tumor cell subsets, co-expression of ALKBH5 with LGR5 ($R=0.39$, $P < 0.0001$; $n=1295$ cells) and CD133 ($R=0.25$, $P < 0.0001$; $n=2867$ cells) were observed (**Figure S1E**). In contrast, few cells showed co-expression of ALKBH5 with stemness markers in myeloid, stromal, T, or B cells (**Figure S1E**). This suggests that tumor cells are the predominant source of ALKBH5 in the CRC TME that corresponds to stemness properties.

2 Q5: In Figure 3A, the x-axis labels remain misaligned.

Response: revised.

3. Q14: The p-values (e.g., WNT pathway $p=0.002$) barely meet significance threshold in GSEA analysis, likely reflecting false positives due to testing numerous pathways. Please provide adjusted p-values (FDR/q-values) to confirm the robustness after multiple-test correction.

Response: We have now updated GSEA analysis in **Figure 4C, 4E and S9G** using q-values. The WNT signaling pathway remains significantly enriched (RNA-seq: $q=0.02$; Ribo-seq: $q < 0.001$).

References:

1. Schmidt-Supprian, M. and K. Rajewsky, *Vagaries of conditional gene targeting*. Nature Immunology, 2007. 8(7): p. 665-668.

Response to the comments of referees in relation to the manuscript:

“RNA N⁶-methyladenosine eraser ALKBH5 drives stemness and chemoresistance of colorectal cancer and is a therapeutic target” (NCOMMS-24-52715B)

Comments are written in *italics* and responses in plain texts. Modifications made to the manuscript are denoted in **BLUE**.

REVIEWER COMMENTS

Response to Reviewer 2:

Reviewer #2 (Remarks to the Author):

In this revised version, the authors have fully addressed all of the issues I pointed out.

As a result, the current manuscript involving many essential and solid data is scientifically and formally worthy of publication in Nature Communications. I respectfully congratulate the authors for their efforts and patience.

Response: We thank the reviewer for the positive comments.

Response to Reviewer 5:

Reviewer #5 (Remarks to the Author):

The authors have made improvements to the manuscript, and I have no further questions.

Response: We thank the reviewer for the positive comments.